# FEDERATED LEARNING, LESSONS FROM GENERALIZATION STUDY: YOU MAY NEED TO COMMUNICATE LESS OFTEN

## ABSTRACT

We investigate the generalization error of statistical learning models in a Federated Learning (FL) setting. Specifically, we study the evolution of the generalization error with the number of communication rounds between the clients and the parameter server, i.e., the effect on the generalization error of how often the local models as computed by the clients are aggregated at the parameter server. In our setup, the more the clients communicate with the server the less data they use for local training at each round. We establish PAC-Bayes and rate-distortion theoretic bounds on the generalization error that account explicitly for the effect of the number of rounds, say $R \in \mathbb{N}^*$, in addition to the number of participating devices $K$ and individual datasets size $n$. The bounds, which apply in their generality for a large class of loss functions and learning algorithms, appear to be the first of their kind for the FL setting. Furthermore, we apply our bounds to FL-type Support Vector Machines (FSVM); and we derive (more) explicit bounds on the generalization error in this case. In particular, we show that the generalization bound of FSVM increases with $R$, suggesting that more frequent communication with the parameter server diminishes the generalization power of such learning algorithms. This implies that comparatively with the empirical risk, the population risk decreases less faster with $R$. Moreover, our bound *suggests* that for every $R$, the generalization error of the FSVM setting decreases faster than that of centralized learning by a factor of $\mathcal{O}(\sqrt{\log(K)/K})$, thereby generalizing recent findings in this direction for $R = 1$ (known as "one-shot" FL) to arbitrary number of rounds. Furthermore, we also provide results of experiments that are obtained using neural networks (ResNet-56) which show evidence that not only may our observations for FSVM hold more generally, but also that the population risk may even start to increase beyond some value of $R$.

## 1 INTRODUCTION

A major focus of machine learning research over recent years has been the development of statistical learning algorithms that can be applied to spatially distributed data. In part, this is due to the emergence of new applications for which it is either not possible (due to lack of resources (Zinkevich et al., 2010; Kairouz et al., 2021)) or not desired (due to privacy concerns (Truex et al., 2019; Wei et al., 2020; Mothukuri et al., 2021)) to collect all the data at one point prior to applying a suitable machine learning algorithm to it (Verbraeken et al., 2020; McMahan et al., 2017; Konečný et al., 2016; Kasiviswanathan et al., 2011). One popular such algorithm is Federated-Learning (FL) (McMahan et al., 2017; Yang et al., 2018; Kairouz et al., 2019; Li et al., 2020; Reddi et al., 2020; Karimireddy et al., 2020; Yuan et al., 2021). In FL, there are $K$ clients or devices that hold each their own dataset and collaborate to collectively train a *(global) model* without sharing their data. The distributions of the clients' data can be identical (*homogeneous*) or different (*heterogeneous*). The model is depicted in Figure 1 and described in Section 2. An FL algorithm typically proceeds in $R \in \mathbb{N}^*$ *communication rounds*: at each round, every device by using some potentially different local algorithms, *e.g.,* Stochastic Gradient Descent (SGD), produces a local model. Then, all these local models of all clients are sent to the *parameter server* (PS) which aggregates them into a (global) model. This aggregated model is sent back to clients and used *typically* as an initialization point for the local algorithms in the next round, although more general forms of statistical dependencies are allowed.

The multi-round interactions between clients and PS are critical to the FL algorithm. Despite its importance, however, little is known about its effect on the performance of the algorithm. In fact, it was shown theoretically (Stich, 2019; Haddadpour et al., 2019; Qin et al., 2022), and also observed experimentally therein, that in FL-type algorithms the empirical risk generally decreases with the number of rounds. This

observation is sometimes over-interpreted and it is believed that more frequent communication of the devices with the PS is generally beneficial for the performance of FL-type algorithms during inference or test phases. This belief was partially refuted in a very recent work (Chor et al., 2023) where it was shown that the generalization error may increase with the number of rounds. Their result, which is obtained by studying the evolution of a bound on the generalization error that they developed for their setup, relies strongly, however, on the assumed assumptions, namely (i) linearity of the con-

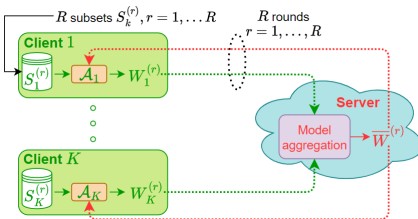

Figure 1: Multi-round Federated Learning

sidered Bregman divergence loss with respect to the hypothesis, (ii) all the devices are constrained to run an SGD with mini-batch size one at every iteration and (iii) linearity of the aggregated model with respect to the devices' individual models. Also, even for the restricted setting considered therein, their bound on the generalization error (Chor et al., 2023, Theorem 1), which is essentially algebraic in nature, does not exploit fully the distributed architecture of the learning problem. Moreover, the dependence on the number of rounds is somewhat buried, at least in part, by decomposing the problem into several virtual one-client SGDs which are inter-dependent among clients and rounds through their initializations.

The effect of the multi-round interactions on the performance of FL algorithms remains highly unexplored, however, for general loss functions and algorithms. For example, with the very few exceptions that we discuss in this paper (most of which pertain to the specific case $R = 1$ (Yagli et al., 2020; Barnes et al., 2022a;b; Sefidgaran et al., 2022a), and with relatively strong assumptions on the loss function and algorithm) the existing literature crucially lacks true bounds on the generalization error for FL-type algorithms, i.e., ones that bring the analysis of the distributed architecture and rounds into the bound, and even less that show some form of explicit dependency on $R$. One central mathematical difficulty in studying the behavior of the expected generalization error is caused by the interactions with the PS inducing statistical correlations among the devices' models which become stronger with $R$ and are not easy to handle. For example, common tools that are generally applied in similar settings, such as the Leave-one-out Expansion Lemma of Shalev-Shwartz et al. (2010), do not apply easily in this case.

**Contributions.** In this paper, we study the problem of how the generalization error of FL-type algorithms evolves with the number of rounds $R$. Unless otherwise specified (for the specialization to Support Vector Machines in Section 4), we assume no particular form for the devices' individual algorithms or deterministic aggregation function at the PS. For this general setting:

- We establish PAC-Bayes bounds (Theorems 1 and 2) and rate-distortion theoretic bounds (Theorem 3 and Theorem 6 in the appendix) on the generalization error that account explicitly for the effect of the number of rounds $R$, in addition to the number of participating devices $K$ and the size of the dataset $n$. These bounds appear to be the first of their kind for the problem that we study.

  The established bounds reflect the structure of the distributed interactive learning algorithm in particular by capturing the contribution of each client at each round to the generalization error of the final model. Our bounds are in terms of *averages* of such contributions among the clients and rounds. In a sense, this validates the intuition that, for a desired generalization error of the final model, some devices may be allowed to overfit during some or all of the rounds as long as other devices compensate for that overfitting. That is, the targeted generalization error level of the final model is suitably split among the devices and rounds. This intuition is also captured, but in a different way, by our *lossy* bounds of Theorems 2, 3, and 6, in the form of a trade-off between the amounts of "lossy compressions" (or "distortion levels") of all clients across all rounds. Finally, we notice that the Kullback–Leibler divergence terms in our PAC-Bayes bounds have the advantage of involving priors that are possibly distinct across devices and rounds. This may be beneficial when these terms are used as regularizers during training. This direction, which can be seen as an extension of the centralized online setup of Haddouche & Guedj (2022) is left for future works.

- We apply our bounds to Federated Support Vector Machines (FSVM); and derive (more) explicit bounds on the generalization error in this case (Theorem 4). Interestingly, we show that the margin generalization bound of FSVM decreases with $K$ and increases with $R$. In particular, this *suggests* that more frequent communication with the PS diminishes the generalization power of FSVM algorithms. As a consequence, comparatively with the empirical risk, the population risk decreases less faster with $R$. Besides, our bound *suggests* that for any $R$, the generalization error of the FSVM setting decreases faster than that of centralized learning by a factor of $\mathcal{O}(\sqrt{\log(K)/K})$, thereby generalizing recent findings in this direction (Sefidgaran et al., 2022a) for $R = 1$ to any arbitrary number of rounds.

- We validate our theoretical findings for FSVM through experiments. Moreover, we perform similar experiments using neural networks (ResNet-56) and observe that our findings obtained for FSVM also hold in this case. That is: (i) the generalization error increases with the number of rounds $R$, and (ii) due to the tradeoff between the empirical risk and generalization error, there exists potentially an optimal number of rounds $R^*$ that minimizes the population risk. We hasten to mention that the total number of training data points and SGD iterations are kept fixed regardless of the value of $R$; hence, the observed increase in the generalization error *cannot* be attributed to the classical "overfitting" phenomenon.

We remark that in the particular case of FL-based SGD, at a high level, there exists a connection between our setup and *LocalSGD* (Stich, 2019; Haddadpour et al., 2019; Qin et al., 2022; Gu et al., 2023), which focuses on the problem of *parallel computing*. The *LocalSGD* literature, however, mostly reports improvements in convergence rates; and their proof techniques do not seem to be applicable for the study of the generalization error. Our findings *suggest* that even in a centralized setup one may still achieve some performance gains, from the viewpoint of generalizability and population risk, by splitting the available dataset into smaller subsets, learning from each separately, aggregating the learned models, and then iterating.

**Notations.** We denote random variables (r.v.), their realizations, and their domains by upper-case, lower-case, and calligraphy fonts, *e.g.,* $X$, $x$, and $\mathcal{X}$. We denote the distribution of $X$ by $P_X$ and its support by supp $P_X$. A r.v. $X$ is called $\sigma$-subgaussian, if for all $t \in \mathbb{R}$, $\log \mathbb{E}[\exp(t(X - \mathbb{E}[X]))] \leq \sigma^2 t^2/2$, where $\mathbb{E}$ denote the expectation. As an example, if $X \in [a, b]$, then $X$ is $\frac{b-a}{2}$-subgaussian. For two distributions $P$ and $Q$ with the Radon-Nikodym derivative $\mathrm{d}Q/\mathrm{d}P$ of $Q$ with respect to $P$, the Kullback–Leibler (KL) divergence is defined as $D_{KL}(Q\|P) := \mathbb{E}_Q[\log(\mathrm{d}Q/\mathrm{d}P)]$ if $Q \ll P$ and $\infty$ otherwise. The mutual information between two r.v. $(X, Y)$ with distribution $P_{X,Y}$ and marginals $P_X$ and $P_Y$ is defined as $I(X; Y) := D_{KL}(P_{X,Y}\|P_X P_Y)$. The notation $\{x_i\}_{i \in [m]}$ is used to denote a collection of $m$ real numbers. The integer ranges $\{1, \ldots, K\} \subset \mathbb{N}^*$ and $\{K_1, \ldots, K_2\} \subset \mathbb{N}^*$ are denoted by $[K]$ and $[K_1 : K_2]$, respectively. Finally, for $k \in [K]$, we use the shorthand notation $[K]\backslash k := \{1, \ldots, K\}\backslash\{k\}$.

## 2 FORMAL PROBLEM SETUP

Consider the $K$-client federated learning model shown in Figure 1.

**Datasets.** For $k \in [K]$, let $Z_k$ be some input data for client or device $k$ distributed according to an unknown distribution $\mu_k$ over some data space $\mathcal{Z}_k = \mathcal{Z}$. For example, in supervised learning settings $Z_k := (X_k, Y_k)$ where $X_k$ stands for a data sample at device $k$ and $Y_k$ its associated label. The distributions $\{\mu_k\}$ are allowed to be distinct across devices. Each client is equipped with its own training dataset $S_k := \{Z_{k,1}, \ldots, Z_{k,n}\} \subseteq \mathcal{Z}^n$, consisting of $n$ independent and identically distributed (i.i.d.) data points drawn according to the unknown distribution $\mu_k$. We consider an $R$-round learning framework, $R \in \mathbb{N}^*$, where every sample of a training dataset can be used during only one round by the device that holds it, but possibly multiple times during that round. Accordingly, it is assumed that every device partitions its data $S_k$ into $R$ disjoint subsets[1] such that $S_k = \bigcup_{r \in [R]} S_k^{(r)}$ where $S_k^{(r)}$ is the dataset used by client $k \in [K]$ during round $r \in [R]$. This is a reasonable assumption that encompasses many practical situations in which at each round every client has access to new data. For ease of the exposition, we assume that $R$ divides $n$ and let $n_R := n/R$ and $S_k^{(r)} := \{Z_{k,1}^{(r)}, \ldots, Z_{k,n_R}^{(r)}\}$. Also, throughout we will often find it convenient to use the handy notation $P_{S_k} := \mu_k^{\otimes n}$, $P_{S_k^{(r)}} := \mu_k^{\otimes n_R}$ and, for $k \in [K]$ and $r \in [R]$,

$$S_{[K]}^{(r)} = S_1^{(r)}, \ldots, S_K^{(r)}, \qquad S_{[K]}^{[r]} = S_1^{[r]}, \ldots, S_k^{[r]} = S_{[K]}^{(1)}, \ldots, S_{[K]}^{(r)},$$
$$S_k^{[r]} = S_k^{(1)}, \ldots, S_k^{(r)}, \qquad \mathbf{S} = S_{[K]}^{[R]}. \tag{1}$$

Similar notations will be used for other variables, e.g., $W_{[K]}^{(r)} = W_1^{(r)}, \ldots, W_K^{(r)}$ and $\overline{W}^{[R]} = \overline{W}^1, \ldots, \overline{W}^{(R)}$.

**Overall algorithm.** The devices collaboratively train a (global) model by performing both local computations and updates based on $R$-round interactions with the parameter server (PS). Let $\mathcal{A}_k$ denote the algorithm used by device $k \in [K]$. An example is the popular SGD where in round $r$ every device $k$ takes one or more gradient steps with respect to samples from the part $S_k^{(r)}$ of its local dataset $S_k$. It should be emphasized that the algorithms $\{\mathcal{A}_k\}$ may be identical or not. During round $r \in [R]$ the algorithm $\mathcal{A}_k$ produces a local model $W_k^{(r)}$. At the end of every round $r$, all the devices send their individual models $W_k^{(r)}$ to the PS which aggregates them into a (global) model $\overline{W}^{(r)} \in \mathcal{W}$ and sends it back to them. The

---

[1] The reader is referred to Appendix B for some extensions of this setup.

aggregated model is used by every device in the next round $(r+1)$, together with the part $S_k^{(r+1)}$ of its dataset $S_k$ in order to obtain a new local model $W_k^{(r+1)}$.

**Local training at devices.** Formally, the algorithm $\mathcal{A}_k$ is a possibly stochastic mapping $\mathcal{A}_k \colon \mathcal{Z}^{n_R} \times \mathcal{W} \to \mathcal{W}$; and, for $r \in [R]$, we have $W_k^{(r)} := \mathcal{A}_k(S_k^{(r)}, \overline{W}^{(r-1)})$ – for convenience, we set $\overline{W}^{(0)} = \varnothing$ or some default value. We denote the conditional distribution induced by $\mathcal{A}_k$ over $\mathcal{W}$ at round $r$ by $P_{W_k^{(r)}|S_k^{(r)},\overline{W}^{(r-1)}}$.

**Model aggregation.** The aggregation function at the PS is set to be deterministic and arbitrary, such that at round $r$ this aggregation can be represented equivalently by a degenerate conditional distribution $P_{\overline{W}^{(r)}|W_{[K]}^{(r)}} = P_{\overline{W}^{(r)}|W_1^{(r)},\ldots,W_K^{(r)}}$. A common choice is the simple averaging $\overline{W}^{(r)} = \left(\sum_{k=1}^K W_k^{(r)}\right)/K$.

The above process repeats until all $R$ rounds are completed, and yields a final (global) model $\overline{W}^{(R)}$. Let $\mathbf{W} = \left(W_{[K]}^{[R]}, \overline{W}^{[R]}\right)$ where the notation used here and throughout is similar to (1).

**Induced probability distributions.** The above-described algorithm, summarized in Algorithm 1 in the appendices, induces the conditional distribution

$$P_{\mathbf{W}|\mathbf{S}} = \bigotimes_{r \in [R]} \left\{ \bigotimes_{k \in [K]} \left( P_{W_k^{(r)}|S_k^{(r)},\overline{W}^{(r-1)}} \right) P_{\overline{W}^{(r)}|W_{[K]}^{(r)}} \right\}, \tag{2}$$

of models, whose joint distribution with $\mathbf{S}$ is

$$P_{\mathbf{S},\mathbf{W}} = \bigotimes_{r \in [R]} \left\{ \bigotimes_{k \in [K]} \left( P_{S_k^{(r)}} P_{W_k^{(r)}|S_k^{(r)},\overline{W}^{(r-1)}} \right) P_{\overline{W}^{(r)}|W_{[K]}^{(r)}} \right\}. \tag{3}$$

Hereafter, we will refer to the aforementioned algorithm for short as being an $(P_{\mathbf{W}|\mathbf{S}}, K, R, n)$-*FL model*.

**Generalization error.** Let $\ell \colon \mathcal{Z} \times \mathcal{W} \to \mathbb{R}^+$ be a given loss function. For a (global) model or hypothesis $\overline{w}^{(R)} \in \mathcal{W}$, its associated empirical and population risks are defined respectively as

$$\hat{\mathcal{L}}\left(\mathbf{s}, \overline{w}^{(R)}\right) = \frac{1}{nK} \sum_{k=1}^K \sum_{i=1}^n \ell\left(z_{k,i}, \overline{w}^{(R)}\right), \quad \mathcal{L}\left(\overline{w}^{(R)}\right) = \frac{1}{K} \sum_{k=1}^K \mathbb{E}_{Z_k \sim \mu_k}\left[\ell\left(Z_k, \overline{w}^{(R)}\right)\right]. \tag{4}$$

Note that by letting $\hat{\mathcal{L}}\left(s_k^{(r)}, \overline{w}^{(R)}\right) = \frac{1}{n_R} \sum_{i=1}^{n_R} \ell\left(z_{k,i}^{(r)}, \overline{w}^{(R)}\right)$, the empirical risk can be re-written as

$$\hat{\mathcal{L}}\left(\mathbf{s}, \overline{w}^{(R)}\right) = \frac{1}{KR} \sum_{r=1}^R \sum_{k=1}^K \hat{\mathcal{L}}\left(s_k^{(r)}, \overline{w}^{(R)}\right). \tag{5}$$

The *generalization error* of the model $\overline{w}^{(R)}$ for dataset $\mathbf{s} = (s_1, \ldots, s_K)$, $s_k = \bigcup_{r=1}^R s_k^{(r)}$, is evaluated as

$$\text{gen}\left(\mathbf{s}, \overline{w}^{(R)}\right) = \mathcal{L}\left(\overline{w}^{(R)}\right) - \hat{\mathcal{L}}\left(\mathbf{s}, \overline{w}^{(R)}\right) = \frac{1}{KR} \sum_{r=1}^R \sum_{k=1}^K \text{gen}\left(s_k^{(r)}, \overline{w}^{(R)}\right), \tag{6}$$

where $\text{gen}\left(s_k^{(r)}, \overline{w}^{(R)}\right) = E_{Z_k \sim \mu_k}\left[\ell\left(Z_k, \overline{w}^{(R)}\right)\right] - \hat{\mathcal{L}}\left(s_k^{(r)}, \overline{w}^{(R)}\right)$.

**Example (FL-SGD).** An important example is one in which every device runs *Stochastic Gradient Decent* (SGD) or variants of it, such as *mini-batch* SGD. In this latter case, denoting by $e$ the number of *epochs* and by $b$ the *mini-batch* size, at iteration $t \in [e\,n_R/b]$ client $k$ updates its model as

$$W_{k,t}^{(r)} = \text{Proj}\left(W_{k,t-1}^{(r)} + \frac{\eta_{r,t}}{b} \sum_{z \in \mathcal{B}_{k,r,t}} \nabla \tilde{\ell}\left(z, W_{k,t-1}^{(r)}\right)\right), \tag{7}$$

where $\tilde{\ell} \colon \mathcal{Z} \times \mathcal{W} \to \mathbb{R}^+$ is some *differentiable surrogate* loss function used for optimization, $\eta_{r,t} > 0$ is the learning rate at iteration $t$ of round $r$, $\mathcal{B}_{k,r,t} \in S_k^{(r)}$ is the mini-batch with size $b$ chosen at iteration $t$, and $\text{Proj}(w') = \arg\min_{w \in \mathcal{W}} \|w - w'\|$. Also, in this case we let $W_k^{(r)} := W_{k,\tau}^{(r)}$, where $\tau := e\,n_R/b$. Besides, here the aggregation function at the PS is typically set to be the arithmetic average $\overline{W}^{(r)} = \left(\sum_{k \in [K]} W_k^{(r)}\right)/K$. This example will be analyzed in the context of Support Vector Machines in Section 4.

## 3 GENERALIZATION BOUNDS FOR FEDERATED LEARNING ALGORITHMS

In this section, we consider a (general) $(P_{\mathbf{W}|\mathbf{S}}, K, R, n)$-*FL algorithm*, as defined formally in Section 2; and study the generalization error of the final (global) hypothesis $\overline{W}^{(R)}$ as measured by (6). Note that the statistical properties of $\overline{W}^{(R)}$ are described by the induced distributions (2) and (3). We establish several

bounds on the generalization error (6). The bounds, which are of PAC-Bayes type and rate-distortion theoretic, have the advantage of taking the structure of the studied multi-round interactive learning problem into account. Also, they account explicitly for the effect of the number of communication rounds $R$ with the PS, in addition to the number of devices $K$ and the size $n$ of each individual dataset. To the best of our knowledge, they are the first of their kind for this problem.

## 3.1 PAC-BAYES BOUNDS

For convenience, we start with two *lossless* bounds, which can be seen as distributed versions of those of McAllester (1998; 1999); Maurer (2004); Catoni (2003) tailored specifically for the multi-round interactive FL problem at hand.

**Theorem 1.** *Assume that the loss $\ell(Z_k, w)$ is $\sigma$-subgaussian for every $w \in \mathcal{W}$ and any $k \in [K]$. Also, let for every $k \in [K]$ and $r \in [R]$, $\mathsf{P}_{k,r}$ denote a conditional prior on $W_k^{(r)}$ given $\overline{W}^{(r-1)}$. Then we have:*

**(i)** *With probability at least $(1 - \delta)$ over $\mathbf{S}$, for all $P_{\mathbf{W}|\mathbf{S}}$, $\mathbb{E}_{\mathbf{W} \sim P_{\mathbf{W}|\mathbf{S}}}\big[\mathrm{gen}(\mathbf{S}, \overline{W}^{(R)})\big]$ is bounded by*

$$\sqrt{\frac{\frac{1}{KR}\sum_{k\in[K],r\in[R]}\mathbb{E}_{\overline{W}^{(r-1)}\sim P_{\overline{W}^{(r-1)}|S_{[K]}^{[r-1]}}}\Big[D_{KL}\Big(P_{W_k^{(r)}|S_k^{(r)},\overline{W}^{(r-1)}}\|\mathsf{P}_{k,r}\Big)\Big] + \log(\frac{\sqrt{2n}}{\sqrt{R}\delta})}{(2n/R - 1)/(4\sigma^2)}}.$$

**(ii)** *For any FL-model $P_{\mathbf{W}|\mathbf{S}}$, with probability at least $(1 - \delta)$ over $(\mathbf{S}, \mathbf{W}) \sim P_{\mathbf{S}, \mathbf{W}}$,*

$$\mathrm{gen}(\mathbf{S}, \overline{W}^{(R)}) \leqslant \sqrt{\frac{\frac{1}{KR}\sum_{k\in[K],r\in[R]}\log\Big(\frac{\mathrm{d}P_{W_k^{(r)}|S_k^{(r)},\overline{W}^{(r-1)}}}{\mathrm{d}\mathsf{P}_{k,r}}\Big) + \log(\frac{\sqrt{2n}}{\sqrt{R}\delta})}{(2n/R - 1)/(4\sigma^2)}}.$$

The proof of Theorem 1, stated in Appendix F.1, judiciously extends the technique of a *variable-size* compressibility approach that was proposed recently in Sefidgaran & Zaidi (2023) in the context of establishing data-dependent PAC-Bayes bounds for a centralized learning setting, i.e., $K = 1$ and $R = 1$, to the more involved setting of FL. This extension is not trivial, however. For example, while the bound of the part (i) of Theorem 1 involves KL-divergence terms that may seem typical of classic PAC-Bayes bounds, the utility of the result is in expressing the bound in terms of average (over clients and rounds) of expected KL-divergence terms, where for every $r \in [R]$ the expectation is over $\overline{W}^{(r-1)}$. This is important, and non-intuitive, as the FL algorithm is *not* composed of $K \times R$ independent centralized algorithms. Indeed, a major technical difficulty in the analysis is to account properly for the problem's distributed nature as well as the statistical "couplings" among the devices' models, induced by the multi-round interactions. In part, these couplings are accounted for in the bound through the conditioning on $\overline{W}^{(r-1)}$. We refer the reader to a discussion after Theorem 2, which is a "lossy" version of the result of Theorem 1, on how the proof proceeds to break down the overall FL-algorithm into $K \times R$ *inter-dependent* "centralized"-like algorithms.

It should also be noted that, in fact, one could still consider the $R$-round FL problem end-to-end and view the entire system as a (virtual) centralized learning system with input the collection $\mathbf{S} = (S_1, \dots, S_K)$ of all devices' datasets and output the final aggregated model $\overline{W}^{(R)}$ and apply the results of Sefidgaran & Zaidi (2023) (or those of McAllester (1998; 1999); Maurer (2004); Catoni (2003); Seeger (2002); Tolstikhin & Seldin (2013)). The results obtained that way, however, do not account for the interactive and distributed structure of the problem. In contrast, note that for example the first bound of Theorem 1 involves, for each device $k$ and round $r$ the KL-divergence term $\mathbb{E}_{\overline{W}^{(r-1)}}\Big[D_{KL}\Big(P_{W_k^{(r)}|S_k^{(r)},\overline{W}^{(r-1)}}\|\mathsf{P}_{k,r}\Big)\Big]$ which can be seen as accounting for (a bound on) the contribution of the model of that client at that round $r$ to the overall generalization error. In a sense, the theorem says that only the average of those KL divergence terms matters for the bound, which could be interpreted as validating the intuition that some clients may be allowed to "overfit" during some or all of the rounds as long as the overall generalization error budget is controlled suitably by other devices. Also, as it can be seen from the result, the choice of priors is specifically tailored for the multi-round multi-client setting in the sense that the prior of client $k$ at round $r$ could depend on all past aggregated models $\overline{w}^{(r-1)}$ and $\mathsf{P}_k^{(r)}$ is allowed to depend on all local models and datasets till round $(r - 1)$. The result also has implications on the design of practical learning FL systems: for example, on one aspect it suggests that the aforementioned KL-divergence term can be used as a round-dependent regularizer which account better for the variation of the quality of the training datasets across devices and rounds.

Finally, by viewing our studied learning setup with disjoint datasets along the clients and rounds as some form of distributed *semi-online* process, Theorem 1 may be seen as a suitable distributed version of the

PAC-Bayes bound of Haddouche & Guedj (2022) established therein for a centralized *online-learning* setup. Note that a direct extension of that result to our distributed setup would imply considering the generalization error of client $k$ at round $r$ with respect to the intermediate aggregated model $\overline{w}_k^{(r)}$, not the final $\overline{w}_k^{(R)}$ as we do. In fact, in that case, the problem reduces to an easier virtual one-round setup that was also studied in (Barnes et al., 2022a;b) for Bregman divergences losses and linear and local Gaussian models; but at the expense of analyzing alternate quantity in place of the true generalization error (6) that we study.

We now present a more general, *lossy*, version of the bound of Theorem 1. The improvement is allowed by introducing a proper *lossy compression* that is defined formally below into the bound. This prevents the new bound from taking large values for deterministic algorithms with continuous hypothesis spaces.

**Lossy compression.** Consider a *quantization set* $\hat{\mathcal{W}} \subseteq \mathcal{W}$ and let $\{\hat{W}_k^{(r)}\}_{k\in[K],r\in[R]}$, $\hat{W}_k^{(r)} \subseteq \hat{\mathcal{W}}$, be a set of *lossy* versions of $\{W_k^{(r)}\}_{k\in[K],r\in[R]}$, defined via some conditional Markov kernels $p_{\hat{W}_k^{(r)}|S_k^{(r)},\overline{W}^{(r-1)}}$, *i.e.,* we consider *lossy compression* of $W_k^{(r)}$ using only $S_k^{(r)}$ and the round-$(r-1)$ aggregated model $\overline{W}^{(r-1)}$. For a given *distortion level* $\epsilon \in \mathbb{R}^+$, $\{p_{\hat{W}_k^{(r)}|S_k^{(r)},\overline{W}^{(r-1)}}\}_{k\in[K],r\in[R]}$ is said to satisfy the $\epsilon$-distortion criterion if following holds: for every $\mathbf{s} \in \mathcal{Z}^{nK}$,

$$\left| \mathbb{E}_{P_{\mathbf{W}|\mathbf{s}}}\Big[ \text{gen}(\mathbf{s},\overline{W}^{(R)}) \Big] - \frac{1}{KR}\sum_{k\in[K],r\in[R]} \mathbb{E}_{(Pp)_{k,r}}\Big[ \text{gen}(s_k^{(r)},\hat{\overline{W}}^{(R)}) \Big] \right| \leqslant \epsilon/2, \tag{8}$$

where $\quad (Pp)_{k,r} = P_{\overline{W}^{(r-1)},W_{[K]\backslash k}^{(r)}|s_{[K]}^{[r-1]},s_{[K]\backslash k}^{(r)}}\, p_{\hat{W}_k^{(r)}|s_k^{(r)},\overline{W}^{(r-1)}}\, P_{\hat{\overline{W}}^{(R)}|W_{[K]\backslash k}^{(r)},\hat{W}_k^{(r)},s_{[K]}^{[r+1:R]}}. \tag{9}$

This condition can be simplified for Lipschitz losses, *i.e.,* when $\forall w,w' \in \mathcal{W},\ |\ell(z,w) - \ell(z,w')| \leqslant \mathfrak{L}\rho(w,w')$, for some distortion function $\rho\colon \mathcal{W}\times\mathcal{W}\to\mathbb{R}^+$. Then, a sufficient condition for (8) is

$$\sum_{k\in[K],r\in[R]} \mathbb{E}_{(Pp)_{k,r}P_{\mathbf{W}|\mathbf{s},\overline{W}^{(r-1)},W_{[K]\backslash k}^{(r)}}}\Big[ \rho(\overline{W}^{(R)},\hat{\overline{W}}^{(R)}) \Big] \leqslant KR\epsilon/(4\mathfrak{L}). \tag{10}$$

**Theorem 2.** *Suppose that* $\ell(z,w) \in [0,C] \subset \mathbb{R}^+$. *Let for every* $k \in [K]$ *and* $r \in [R]$, $\mathsf{P}_{k,r}$ *be a conditional prior on* $\hat{W}_k^{(r)}$ *given* $\overline{W}^{(r-1)}$. *Fix any* distortion level $\epsilon \in \mathbb{R}^+$. *Consider any* $(P_{\mathbf{W}|\mathbf{S}},K,R,n)$-*FL model and any choices of* $\{p_{\hat{W}_k^{(r)}|S_k^{(r)},\overline{W}^{(r-1)}}\}_{k\in[K],r\in[R]}$ *that satisfy the* $\epsilon$-*distortion criterion. Then, with probability at least* $(1-\delta)$ *over* $\mathbf{S}\sim P_{\mathbf{S}}$, *we have that* $\mathbb{E}_{\mathbf{W}\sim P_{\mathbf{W}|\mathbf{S}}}\Big[ \text{gen}(\mathbf{S},\overline{W}^{(R)}) \Big]$ *is upper bounded by*

$$\sqrt{\frac{\frac{1}{KR}\sum_{k\in[K],r\in[R]}\mathbb{E}_{\overline{W}^{(r-1)}\sim P_{\overline{W}^{(r-1)}|S_{[K]}^{[r-1]}}}\Big[ D_{KL}\Big(p_{\hat{W}_k^{(r)}|S_k^{(r)},\overline{W}^{(r-1)}}\|\mathsf{P}_{k,r}\Big) \Big] + \log(\frac{\sqrt{2n}}{\sqrt{R}\delta})}{(2n/R-1)/C^2}} + \epsilon.$$

A trivial extension of the lossy PAC-Bayes bounds for centralized algorithms could be done by considering the *quantization* of the final aggregated model $W^{(R)}$. Here, instead, for every $k \in [K]$ and round $r \in [R]$, we quantize the local model $W_k^{(r)}$ separately while keeping the other devices' local models at that round *i.e.,* the vector $W_{[K]\backslash k}^{(r)}$, fixed. This allows us to study the amount of the "propagated" distortion till round $R$. Note that by quantizing $W_k^{(r)}$ for a distortion constraint on the generalization error that is at most $\nu$, the immediate aggregated model $\overline{\overline{W}}_k^{(r)}$ is guaranteed to have a generalization error within a distortion level of at most $\nu/K$ from the true value. Also, interestingly, the distortion criterion (8) allows to "allocate" suitably the targeted total distortion $KR\epsilon/2$ into smaller constituent levels among all clients and across all rounds.

**Proof outline:** Theorem 2 is proved in Appendix F.2 by breaking down the overall FL algorithm into $KR$ "centralized"-like algorithms, using the following high-level steps:

i. For every pair $(k,r)$, we define a virtual FL algorithm that is equivalent to the original one except for round $r$ of client $k$ for which the "local quantized algorithm" $p_{\hat{W}_k^{(r)}|S_k^{(r)},\overline{W}^{(r-1)}}$ is considered instead of $P_{W_k^{(r)}|S_k^{(r)},\overline{W}^{(r-1)}}$. This overall algorithm is denoted by $(Pp)_{k,r}$ as given by (9). Also, we define the overall algorithm with respect to the prior $\mathsf{P}_{k,r}$. These choices allow us to define $2KR$ 'virtual' algorithms, each of them differing in one distinct (client, round)-local algorithm from the original.

ii. Next, by Lemma 3 we relate the probability that the generalization error exceeds a given threshold to the supremum of the expected average generalization error of all $KR$-virtual algorithms $(PP_{k,r})_{k,r}$. The supremum is taken with respect to all joint distributions that are in the vicinity of $\mu^{\otimes nK}$.

iii. Finally, we apply a change of measure argument by two successive applications of Donsker-Varadhan's inequality, from $(PP_{k,r})_{k,r}$ to $(PP_{k,r})_{k,r}$ and from $\nu_{\mathbf{S}}$ to $\mu^{\otimes nK}$. Using the special form of $(PP_{k,r})_{k,r}$, the KL-divergences $D_{KL}\big((PP_{k,r})_{k,r}\|(PP_{k,r})_{k,r}\big)$ lead to the desired KL-divergence terms. This allows in the last step of the proof to bound the cumulant generating function as needed.

## 3.2 RATE-DISTORTION THEORETIC BOUNDS

Define for $k \in [K]$, $r \in [R]$ and $\epsilon \in \mathbb{R}$, the rate-distortion function

$$\mathfrak{RD}(P_{\mathbf{S},\mathbf{W}}, k, r, \epsilon) = \inf_{p_{\hat{W}_k^{(r)}|S_k^{(r)}, \overline{W}^{(r-1)}}} I(S_k^{(r)}; \hat{W}_k^{(r)}|\overline{W}^{(r-1)}), \tag{11}$$

where the mutual information is evaluated with respect to $P_{S_k^{(r)}} P_{\overline{W}^{(r-1)}} p_{\hat{W}_k^{(r)}|S_k^{(r)}, \overline{W}^{(r-1)}}$ and the infimum is over all conditional Markov kernels $p_{\hat{W}_k^{(r)}|S_k^{(r)}, \overline{W}^{(r-1)}}$ that satisfy

$$\mathbb{E}_{\mathbf{S},\mathbf{W},\hat{\overline{W}}^{(R)}}\Big[\operatorname{gen}(S_k^{(r)}, \overline{W}^{(R)}) - \operatorname{gen}(S_k^{(r)}, \hat{\overline{W}}^{(R)})\Big] \leqslant \epsilon, \tag{12}$$

where the joint distribution of $\mathbf{S}, \mathbf{W}, \hat{\overline{W}}^{(R)}$ factorizes as $P_{\mathbf{S},\mathbf{W}} \, p_{\hat{W}_k^{(r)}|S_k^{(r)}, \overline{W}^{(r-1)}} P_{\hat{\overline{W}}^{(R)}|W_{[K]\backslash k}^{(r)}, \hat{W}_k^{(r)}}$.

**Theorem 3.** *For any $(P_{\mathbf{W}|\mathbf{S}}, K, R, n)$-FL model with distributed dataset $\mathbf{S} \sim P_{\mathbf{S}}$, if the loss $\ell(Z_k, w)$ is $\sigma$-subgaussian for every $w \in \mathcal{W}$ and any $k \in [K]$, then for every $\epsilon \in \mathbb{R}$ it holds that*

$$\mathbb{E}_{\mathbf{S},\mathbf{W} \sim P_{\mathbf{S},\mathbf{W}}}\Big[\operatorname{gen}(\mathbf{S}, \overline{W}^{(R)})\Big] \leqslant \sqrt{2\sigma^2 \sum_{k \in [K], r \in [R]} \mathfrak{RD}(P_{\mathbf{S},\mathbf{W}}, k, r, \epsilon_{k,r})/(nK)} + \epsilon,$$

*for any set of parameters $\{\epsilon_{k,r}\}_{k \in [K], r \in [R]} \subset \mathbb{R}$ which satisfy $\frac{1}{KR} \sum_{k \in [K]} \sum_{r \in [R]} \epsilon_{k,r} \leqslant \epsilon$.*

Similar to the PAC-Bayes type bounds of Theorem 1 and Theorem 2, the bound of Theorem 3 also shows the "contribution" of each client's local model during each round to (a bound on) the generalization error as measured by (6). we refer the reader to Appendix F.3 for its proof, due to lack of space and the considerable needed technicality details. As it will become clearer from the below, an extended version of this theorem, stated in Appendix D.2, is particularly useful to study the *Federated Support Vector Machines* (FSVM) of the next section. For instance, an application of that result will yield a (more) explicit bound on the generalization error of FSVM in terms of the parameters $K$, $R$, and $n$.

Finally, a tail rate-distortion theoretic bound is derived in supplements. This result states loosely that having a good generalization performance with high probability requires the algorithm to be compressible not only under the true distribution $P_{\mathbf{S},\mathbf{W}}$, but also for all those distributions $\nu_{\mathbf{S},\mathbf{W}}$ that are in the vicinity of $P_{\mathbf{S},\mathbf{W}}$.

## 4 FEDERATED SUPPORT VECTOR MACHINES (FSVM)

In this section, we study the generalization behavior of Support Vector Machines (SVM) (Cortes & Vapnik, 1995; Vapnik, 2006) when optimized in the FL setup using SGD. SVM is a popular model, mainly used for binary classification, and is particularly powerful when used with high-dimensional kernels. For ease of exposition, however, we only developed results for linear SVMs that can be extended easily to any kernels.

Consider a binary classification problem in which $\mathcal{Z} = \mathcal{X} \times \mathcal{Y}$, $\mathcal{X} \subseteq \mathbb{R}^d$ with $\mathcal{Y} = \{-1, +1\}$. Using SVM for this problem consists of finding a suitable hyperplane, represented by $w \in \mathbb{R}^d$, that properly separates the data according to their labels. For convenience, we only consider the case with zero bias. In this case, the label of an input $x \in \mathbb{R}^d$ is estimated using the sign of $\langle x, w \rangle$. The 0-1 loss $\ell_0: \mathcal{Z} \times \mathcal{W} \to \mathbb{R}^+$ is then evaluated as $\ell_0(z, w) := \mathbb{1}_{\{y\langle x, w\rangle > 0\}}$. For the specific cases of centralized ($K = R = 1$) and "one-shot" FL ($K \geqslant 2$, $R = 1$) settings (Grønlund et al., 2020; Sefidgaran et al., 2022a), it was observed that if the algorithm finds a hyperplane that separates the data with some *margin* then it generalizes well. Motivated by this, we consider the 0-1 margin loss function $\ell_\theta: \mathcal{Z} \times \mathcal{W} \to \mathbb{R}^+$ for $\theta \in \mathbb{R}^+$ as $\ell_\theta(z, w) := \mathbb{1}_{\{y\langle x, w\rangle > \theta\}}$. Similar to previous studies, we consider the empirical risk with respect to the margin loss, *i.e.,* $\hat{\mathcal{L}}_\theta(\mathbf{s}, \overline{w}^{(R)}) = \frac{1}{nK} \sum_{k \in [K], i \in [n]} \ell_\theta(z_{k,i}, \overline{w}^{(R)})$ which is equal to $\frac{1}{KR} \sum_{k \in [K], r \in [R]} \hat{\mathcal{L}}_\theta(\mathbf{s}_k^{(r)}, \overline{w}^{(R)})$. The population risk is considered with respect to 0-1 loss function $\ell_0$, *i.e.,* $\mathcal{L}(\overline{w}^{(R)}) = \frac{1}{K} \sum_{k \in [K]} \mathbb{E}_{Z_k \sim \mu_k}\big[\ell_0(Z_k, \overline{w}^{(R)})\big]$. The margin generalization error is then defined as $\operatorname{gen}_\theta(\mathbf{s}, \overline{w}^{(R)}) = \mathcal{L}(\overline{w}^{(R)}) - \hat{\mathcal{L}}_\theta(\mathbf{s}, \overline{w}^{(R)})$.

For the statement of the result that will follow, we make three assumptions on SGD. Due to lack of space, these assumptions are attentively described and discussed in Appendix C, and here we only state them informally. Consider an $(K, R, n, e, b)$-FL-SGD model. Let $\mathcal{W} = \mathfrak{B}_d(1)$, where $\mathfrak{B}_d(\nu)$ denotes the $d$-dimensional ball with origin center and radius $\nu > 0$.

- **Assumption 1:** There exists some $q_{e,b} \in \mathbb{R}^+$ such that for each client $k \in [K]$ at each round $r \in [2:R]$, the local models $w_k^{(r)}$ and $w_k'^{(r)}$ attained by some initializations $\overline{w}^{(r-1)}$ and $\overline{w}'^{(r-1)}$, respectively, satisfy $\|w_k^{(r)} - w_k'^{(r)}\| \leqslant q_{e,b} \|\overline{w}^{(r-1)} - \overline{w}'^{(r-1)}\|$. If $q_{e,b} < 1$, SGD is called to be *contractive*.

- **Assumption 2:** There exists some $\alpha \in \mathbb{R}^+$ such that for each client $k \in [K]$ at each round $r \in [2:R]$, the local models $w_k^{(r)}$ and $w_k'^{(r)}$ attained by some initializations $\overline{w}^{(r-1)}$ and $\overline{w}'^{(r-1)} := \overline{w}^{(r-1)} + \frac{w_\varepsilon}{K}$, satisfy $\|w_k'^{(r)} - \left(w_k^{(r)} + \frac{1}{K} \mathsf{D}_{k,r}\, w_\varepsilon\right)\| \leqslant \frac{\alpha}{K^2} \|w_\varepsilon\|^2$ for some matrix $\mathsf{D}_{k,r} \in \mathbb{R}^{d \times d}$, possibly dependent on $S_k^{(r)}$ and $\overline{w}^{(r-1)}$, whose spectral norm is bounded by $q_{e,b}$. Intuitively, this is an assumption on the first-order approximation error of the local steps of SGD for one client at one round.

- **Assumption 3:** The number of clients $K$ is sufficiently large, *i.e.,* $K \geqslant f(q_{e,b}, \alpha, R)$ for some function $f$ which is precised in Appendix C.

Now, we are ready to state our bound on the generalization error of FSVM.

**Theorem 4.** *For FSVM optimized using $(K, R, n, e, b)$-FL-SGD with $\mathcal{W} = \mathfrak{B}_d(1)$, $\mathcal{X} = \mathfrak{B}_d(B)$ and $\theta \in \mathbb{R}^+$, if Assumptions 1, 2, and 3 hold for some constants $q_{e,b}$ and $\alpha$, then,*

$$\mathbb{E}_{\mathbf{S}, \mathbf{W} \sim P_{\mathbf{S}, \mathbf{W}}}\left[\mathrm{gen}_\theta\left(\mathbf{S}, \overline{W}^{(R)}\right)\right] = \mathcal{O}\left(\sqrt{\frac{B^2 \log(nK\sqrt{K}) \sum_{r \in [R]} L_r}{nK^2 \theta^2}}\right), \tag{13}$$

$$\textit{where} \qquad L_r = \inf_{t \geqslant q^{(R-r)}}\left\{t \log \max\left(\frac{K\theta}{Bt}, 2\right)\right\} \leqslant q_{e,b}^{2(R-r)} \log \max\left(\frac{K\theta}{B q_{e,b}^{(R-r)}}, 2\right). \tag{14}$$

To the best of our knowledge, this result is the first of its kind for FSVM. We pause to state a few remarks that are in order, before discussing some key elements of its proof technique. First, the bound of Theorem 4 shows an explicit dependence on the number of communication rounds $R$, in addition to the number of participating devices $K$ and the size of individual datasets $n$. In particular, the bound increases with $R$ for fixed $(n, K)$. This *suggests* that the generalization power of FSVM may diminish by more frequent communication with the PS, as illustrated also numerically in Section 5. As a consequence, the population risk decreases less faster with $R$ than the empirical risk. By taking into account the extra communication cost of larger $R$, this means that during the training phase of such systems, one might choose *deliberately* to stop before convergence (at some appropriate round $R^\star \leqslant R$), accounting for the fact that while interactions that are beyond $R^\star$ indeed generally contribute to diminishing the empirical risk further their net effect on the true measure of performance, which is the population risk, may be negligible. In the experiment section, we show such net effect when local models are ResNet-56 could be even negative. Second, for fixed $(n, R)$ the bound improves (i.e., gets smaller) with $K$ with a factor of $\sqrt{\log(K)/K}$. This behavior was previously observed in different contexts and under different assumptions in Sefidgaran et al. (2022a) and Barnes et al. (2022a;b), but in both works only for the "one-shot" FL setting, i.e., $R = 1$. In fact, it is easily seen that for the specific case $R = 1$, the bound recovers the result of Sefidgaran et al. (2022a, Theorem 5).

**Proof outline:** Theorem 4 is proved in Appendix F.4 using an extended version of Theorem 3, *i.e.,* Proposition 1 (in Appendix D.2), and by bounding the appearing rate-distortion terms therein. Intuitively, rate-distortion terms are equal to the number of bits needed to represent an (optimal) quantized model given certain quantization precision. To establish an appropriate upper bound that is independent of the dimension $d$, we apply a quantization on a smaller $d$-independent dimension, using the Johnson-Lindenstrauss (JL) dimension-reduction transformation (Johnson & Lindenstrauss, 1984), which is inspired by Grønlund et al. (2020); Sefidgaran et al. (2022a); but with substantial extensions needed for the considered multiple-client multi-round setup. In particular, the main difficulty here is that while in the rate-distortion terms of Proposition 1, the mutual information term (11) does not depend on $\overline{W}^{(R)}$, the distortion criterion (12) does. Thus, one needs to study the propagation of the distortion, induced by quantizing a local model, until the last round. The distortion propagation differs depending on the round $r$ where the local model is located. Hence, for a fixed propagated distortion, the amount of dimension reduction should depend on $r$.

More precisely, for each $k \in [K]$ and $r \in [R]$ we first map the local models $W_k^{(r)}$ to a space with a smaller dimension of order $m_r = \mathcal{O}\left(\frac{B^2 \log(nK\sqrt{K}) \log(K/t)}{K^2 \theta^2 t^2}\right)$, for some $t \geqslant q^{(R-r)}$, using JL transformation. As can be observed, we allow possibly different values of the dimensions for different values of $r$. For a contractive SGD, $m_r$ decreases with $r$. Then, we define the quantized model subtly using this locally mapped model, as defined in equation (63). This quantization together with a newly defined loss functionin

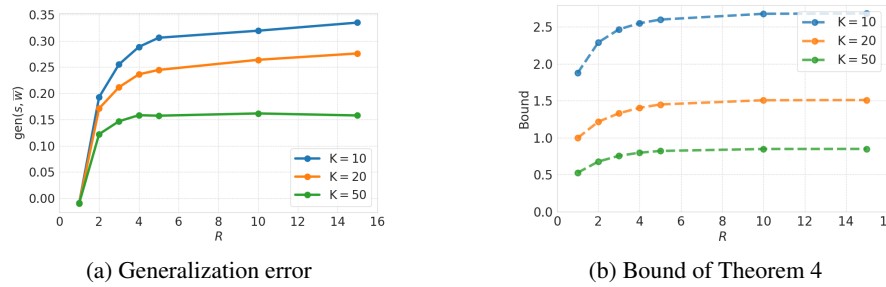

(a) Generalization error            (b) Bound of Theorem 4

Figure 2: Generalization error of FSVM and bound of Theorem 4 as functions of $R$, for $n = 100$

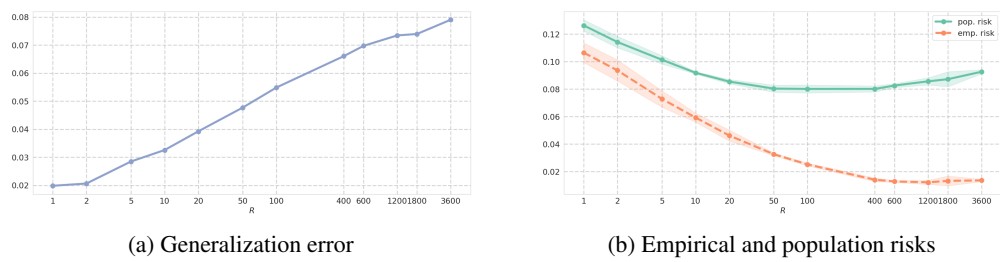

(a) Generalization error            (b) Empirical and population risks

Figure 3: Performance of FL-SGD with ResNet-56 local models as a function of $R$

equation (65) allow to study the propagation of such quantization distortion in two steps: first, we show that the immediate aggregated model has $K$ times smaller distortion level than that of client $k$. Next, we study the evolution of distortion along SGD iterations till the end of the round $R$, using induction on the rounds (see "Bounding (73)" in the proof) and by exploiting the properties of the JL transformation (see "Bounding (74)" & "Bounding (75)" in the proof). Intuitively, for a contractive SGD, the distortion *decreases*. This is why our $m_r$ decreases with $r$. For a non-contractive SGD, the opposite holds.

## 5 EXPERIMENTS

**FSVM.** We start by verifying the increasing behavior of the generalization error of FSVM with respect to $R$, as suggested by Theorem 4. To do so, we consider a binary classification problem, with two extracted classes (digits 1 and 6) of MNIST dataset. Details and further experiments can be found in Appendix E.

Fig. 2 shows the (estimated) expected generalization error $\mathbb{E}_{\mathbf{S},\mathbf{W}}[\text{gen}(\mathbf{S}, \overline{W}^{(R)}]$ and the computed bound of Theorem 4 versus the number of communication rounds $R$, for fixed $n = 100$ and for $K \in \{10, 20, 50\}$. The expectation is estimated over $M = 100$ Monte-Carlo simulations. As can be observed, for any value of $K$, the generalization error increases with $R$, as predicted by the bound of Theorem 4. We emphasize that for Fig. 2 the total number of training data points and SGD iterations are kept fixed regardless of the number of running communication rounds $R$ (for every value of $R$ the devices perform $\tau = en/R$ local SGD iterations per-round); and, hence, the increase in the generalization error *cannot* be attributed to the classical "overfitting" phenomenon. Appendix E.4 also reports similar behavior in the context of *heterogeneous* data.

**Additional experiments on generalization of FL.** To numerically verify the validity of our findings beyond FSVM, we conducted additional experiments using ResNet-56 as local models, and with CIFAR-10 dataset. Fig. 3a shows the generalization error of the global model as a function of $R$ while Fig. 3b shows the corresponding empirical and population risks. We provide average values over 5 runs and the shaded areas correspond to the standard deviation values. Experimental details are given in Appendix E.

One can observe in Fig. 3a that the generalization error is increasing with $R$, showing that the behavior suggested by Theorem 4, and observed in the above FSVM experiments, remains valid in a different setup. The empirical risk in Fig. 3b is decreasing with $R$ as expected. More importantly, the population risk in Fig. 3b can be observed to have a global minimum for $R^* \simeq 100$, while the maximum number of communication rounds is $R = 3600$, thus showing that one can minimize the "true" objective *i.e.,* the population risk, while reducing communication from a large amount.[2]

---

[2]The question of estimating, *prior to training*, the optimal value $R^*$ of $R$ is an important one, with far-reaching consequences in practice, e.g., for the design of simultaneously communication-efficient and generalizable FL algorithms. This is left for future works.

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

# Appendices

The appendices are organized as follows:

- Appendix A contains the algorithmic representation of the considered federated learning setup.

- Appendix B contains the discussion and some preliminary results on the case where the datasets used by each client during different rounds of the federated learning algorithm are not disjoint.

- Appendix C contains the rigorous statement and discussion of the Assumptions made for establishing the generalization bound of FSVM in Theorem 4.

## A    FEDERATED LEARNING ALGORITHM

Denote the deterministic aggregation function equivalent to the degenerate conditional distribution $P_{\overline{W}^{(r)}|W_{[K]}^{(r)}}$, $r \in [R]$, by the mapping $\mathcal{M}\colon \mathcal{W}^K \mapsto \mathcal{W}$, such that

$$\overline{W}^{(r)} = \mathcal{M}\left(W_{[K]}^{(r)}\right) \sim P_{\overline{W}^{(r)}|W_{[K]}^{(r)}}.$$

Then, the federated learning setup, described in Section 2 can be stated alternatively as in Algorithm 1.

---

**Algorithm 1** The federated learning algorithm

---

1: **Inputs**: number of rounds $R$, local algorithms $\mathcal{A}_k$ and dataset $S_k = \bigcup_{r\in[R]} S_k^{(r)}$ for $k \in [K]$, and the aggregation function $\mathcal{M}$

2: **Output**: $\overline{W}^{(R)}$

3: $\overline{W}^{(0)} = \varnothing$

4: **for** $r = 1, 2 \ldots, R$ **do**

5:     $\forall k \in [K]\colon W_k^{(r)} = \mathcal{A}_k(S_k^{(r)}, \overline{W}^{(r-1)})$

6:     $\overline{W}^{(r)} = \mathcal{M}\left(W_{[K]}^{(r)}\right)$

7: **end for**

---

# B    EXTENSION OF THE CONSIDERED FEDERATED LEARNING SETUP

In this paper, as a first step towards understanding the generalization behavior of Federated Learning algorithms, we considered a setup where in each round multiple "local epochs" $e \in \mathbb{N}^*$ can occur for each client's data but the used datapoints are not reused in other rounds. That is, at round $r \in [R]$, client $k \in [K]$ computes updates using $S_k^{(r)} \subseteq S_k$ in order to minimize the empirical risk. When $e = 1$, our setup is then a "distributed online learning" setup. Therefore, we have defined a variant of the online learning framework or the "without replacement" setup with one epoch, that has been used before *e.g.,* in (Haddouche & Guedj, 2022) for a similar one-client multi-round setup.

Here, we show that our framework and proof techniques can be extended to include multiple epochs of the without replacement setup, *i.e.,* when one data can be used in multiple rounds, as well. More precisely, assume $\bigcup_{r \in [R]} S_k^{(r)} \subseteq S_k$ where $S_k^{(r)}$ is the dataset used by client $k \in [K]$ during round $r \in [R]$ with size $n_{k,r}$. Here, in contrast to the rest of the paper, we allow the datasets $\{S_k^{(r)}\}_{r \in [R]}$ to have intersections for each $k \in [K]$, *i.e.,* a given $Z_{k,j} \in \mathcal{S}_k$, $j \in [n]$, may be used in multiple rounds and therefore belongs to multiple $S_k^{(r)}$, $r \in [R]$. We denote the elements of $S_k^{(r)}$ by $S_k^{(r)} = \{Z_{k,r,1}, \ldots, Z_{k,r,n_{k,r}}\}$. Note that

$$\bigcup_{r \in [R]} \bigcup_{j \in [n_{k,r}]} \{Z_{k,r,j}\} \subseteq \{Z_{k,1}, \ldots, Z_{k,n}\}.$$

The local training algorithm and the model aggregation are the same as before. In other words, $P_{\mathbf{W}|\mathbf{S}}$ is similar to (2) defined as

$$P_{\mathbf{W}|\mathbf{S}} = \bigotimes_{r \in [R]} \left\{ \bigotimes_{k \in [K]} \left( P_{W_k^{(r)}|S_k^{(r)}, \overline{W}^{(r-1)}} \right) P_{\overline{W}^{(r)}|W_{[K]}^{(r)}} \right\}. \tag{15}$$

**Generalization error.**    For a (global) model or hypothesis $\overline{w}^{(R)} \in \mathcal{W}$, the population risk is defined as before:

$$\mathcal{L}\left(\overline{w}^{(R)}\right) = \frac{1}{K} \sum_{k=1}^{K} \mathbb{E}_{Z_k \sim \mu_k}\left[\ell\left(Z_k, \overline{w}^{(R)}\right)\right].$$

We define the empirical risk as

$$\hat{\mathcal{L}}\left(\mathbf{s}, \overline{w}^{(R)}\right) := \frac{1}{KR} \sum_{k=1}^{K} \sum_{r=1}^{R} \hat{\mathcal{L}}\left(s_k^{(r)}, \overline{w}^{(R)}\right) \tag{16}$$
$$= \frac{1}{KR} \sum_{k=1}^{K} \sum_{r=1}^{R} \frac{1}{n_{k,r}} \sum_{j=1}^{n_{k,r}} \ell\left(z_{k,r,j}, \overline{w}^{(R)}\right).$$

The *generalization error* of the model $\overline{w}^{(R)}$ for dataset $\mathbf{s} = s_{[K]}^{[R]}$ is evaluated as

$$\text{gen}\left(\mathbf{s}, \overline{w}^{(R)}\right) = \mathcal{L}\left(\overline{w}^{(R)}\right) - \hat{\mathcal{L}}\left(\mathbf{s}, \overline{w}^{(R)}\right) = \frac{1}{KR} \sum_{r=1}^{R} \sum_{k=1}^{K} \text{gen}\left(s_k^{(r)}, \overline{w}^{(R)}\right), \tag{17}$$

where $\text{gen}\left(s_k^{(r)}, \overline{w}^{(R)}\right) = E_{Z_k \sim \mu_k}\left[\ell\left(Z_k, \overline{w}^{(R)}\right)\right] - \hat{\mathcal{L}}\left(s_k^{(r)}, \overline{w}^{(R)}\right)$.

For this setup, we derive an equivalent lossless version of Theorem 3. The equivalent results for the lossy version, as well as other theorems, can be established similarly, but with cumbersome notation and complex dependencies between each round's data for each client and their models at all rounds.

For every $k \in [K]$ and $1 \leqslant r \leqslant r' \leqslant R$, denote

$$W_k^{[r:r']} := W_k^{(r)}, W_k^{(r+1)}, \ldots, W_k^{(r')},$$

with the convention that $W_k^{[r:r]} := W_k^{(r)}$.

**Theorem 5.** *Assume that the loss $\ell(Z_k, w)$ is $\sigma$-subgaussian for every $w \in \mathcal{W}$ and any $k \in [K]$. Then,*

$$\mathbb{E}_{\mathbf{S}, \mathbf{W} \sim P_{\mathbf{S}, \mathbf{w}}}\left[\text{gen}\left(\mathbf{S}, \overline{W}^{(R)}\right)\right] \leqslant \sqrt{\frac{2\sigma^2}{KR} \sum_{k \in [K], r \in [R]} \frac{1}{n_{k,r}} I\left(S_k^{(r)}; \overline{W}^{(r-1)}, W_k^{[r:R]}\right)}. \tag{18}$$

This result is proved in Section F.5.

Using chain rule, we can write each term in the summation as

$$
\begin{aligned}
I\left(S_k^{(r)}; \overline{W}^{(r-1)}, W_k^{[r:R]}\right) = & I\left(S_k^{(r)}; \overline{W}^{(r-1)}\right) \\
& + I\left(S_k^{(r)}; W_k^{(r)} \big| \overline{W}^{(r-1)}\right) \\
& + I\left(S_k^{(r)}; W_k^{(r+1)} \big| \overline{W}^{(r-1)}, W_k^{(r)}\right) \\
& + \dots \\
& + I\left(S_k^{(r)}; W_k^{(R)} \big| \overline{W}^{(r-1)}, W_k^{[r:R-1]}\right).
\end{aligned}
$$

The term $I\left(S_k^{(r)}; \overline{W}^{(r-1)}\right)$ captures the dependence of $S_k^{(r)}$ on the aggregated model $\overline{W}^{(r-1)}$ and accounts for the appearance of elements of $S_k^{(r)}$ in the previous rounds. If $\{S_k^{(r)}\}_{r \in [R]}$ are disjoint, *i.e.,* one epoch of the without replacement setup as considered throughout the paper, then this term is equal to zero. Furthermore, the terms $I\left(S_k^{(r)}; W_k^{(r+1)} \big| \overline{W}^{(r-1)}, W_k^{(r)}\right), \dots, I\left(S_k^{(r)}; W_k^{(R)} \big| \overline{W}^{(r-1)}, W_k^{[r:R-1]}\right)$ capture the appearance of elements of $S_k^{(r)}$ in the following rounds. This term is also equal to zero for one epoch of the without replacement setup. Thus, whenever $\{S_k^{(r)}\}_{r \in [R]}$ are disjoint, Theorem 5 reduces to

$$
\mathbb{E}_{\mathbf{S}, \mathbf{W} \sim P_{\mathbf{S}, \mathbf{w}}}\left[\operatorname{gen}\left(\mathbf{S}, \overline{W}^{(R)}\right)\right] \leqslant \sqrt{\frac{2\sigma^2}{KR} \sum_{k \in [K], r \in [R]} \frac{1}{n_{k,r}} I\left(S_k^{(r)}; W_k^{(r)} \big| \overline{W}^{(r-1)}\right)}, \qquad (19)
$$

which is the lossless version of Theorem 3. This result therefore *extends* the lossless version of Theorem 3. Now, considering for example the round $(r+1)$ of client $k$, then only few samples of $S_k^{(r)}$ are used in this round (together with other samples from $S_k$) to generate $W_k^{(r+1)}$. Moreover, the samples used from $S_k^{(r)}$ are also used before in round $r$ to generate $W_k^{(r)}$, which is then aggregated by other client models and used as the initialization in round $r + 1$. Thus, one *expects* $I\left(S_k^{(r)}; W_k^{(r+1)} \big| \overline{W}^{(r-1)}, W_k^{(r)}\right)$ to be smaller than $I\left(S_k^{(r)}; W_k^{(r)} \big| \overline{W}^{(r-1)}\right)$. In addition, the term $I\left(S_k^{(r)}; \overline{W}^{(r-1)}\right)$ captures the dependence of $S_k^{(r)}$ on the *aggregated* model $\overline{W}^{(r-1)}$. Again, if the overlap between datasets used in different rounds is small and if the number of clients $K$ is sufficiently large, we can expect this term to be small. Thus, in the regime where the amount of data used from previous rounds is small and the number of clients is large, then one expects the RHS of (19) (which is the lossless version of Theorem 3) to be not very far from the RHS of (18). However, this *justification* requires rigorous analysis which is left for future work.

Finally, suppose that the setup considered in Section A is repeated for $e' \in \mathbb{N}^*$ epochs, *i.e.,* in total $e'R$ rounds. This means that for every $k \in [K]$ and $r \in [R]$, the sets $\{S_k^{(r)}\}_{r \in [R]}$ are disjoint and

$$
S_k^{(r)} = S_k^{(R+r)} = \dots = S_k^{((e'-1)R+r)}.
$$

Note that in this setup, the total number of rounds is $e'R$, and the final global model is $\overline{W}^{e'R}$. This setup for each client is called "SingleShuffle" without-replacement SGD with $e'$-epochs (Ahn et al., 2020). It can be easily verified that in this case, the bound of (19) can be further upper bounded as

$$
\mathbb{E}_{\mathbf{S}, \mathbf{W} \sim P_{\mathbf{S}, \mathbf{w}}}\left[\operatorname{gen}\left(\mathbf{S}, \overline{W}^{(eR)}\right)\right] \leqslant \sqrt{\frac{2\sigma^2}{nK} \sum_{k \in [K], r \in [R]} \sum_{j \in [e']} I\left(S_k^{(r)}; W_k^{((j-1)R+r)} \big| \overline{W}^{((j-1)R+r-1)}\right)}.
$$

$$(20)$$

Once again, intuitively, for each $k$ and $r$, only the first few terms of the summation $\sum_{j \in [e']} I\left(S_k^{(r)}; W_k^{((j-1)R+r)} \big| \overline{W}^{((j-1)R+r-1)}\right)$ are expected to form the dominant term.

## C  ASSUMPTIONS FOR GENERALIZATION BOUND OF FSVM

In this section, we state and discuss rigorously the Assumptions that are used for establishing the generalization bound of FSVM in Theorem 4 in Section 4.

**Assumption 1.** *Consider an* $(K, R, n, e, b)$*-FL-SGD model. For every* $r \in [2 : R]$*, every pair* $(\overline{w}^{(r-1)}, \overline{w}'^{(r-1)})$ *and every* $k \in [K]$*, the following holds for any pair* $(w_k^{(r)}, w_k'^{(r)})$ *that are both generated using* $s_k^{(r)}$ *with exactly similar mini-batches* $\mathcal{B}_{k,r,t} \subseteq s_k^{(r)}$*,* $t \in [e\,n_R/b]$*: there exists* $q_{e,b} \in \mathbb{R}^{+3}$ *such that*

$$\left\| w_k^{(r)} - w_k'^{(r)} \right\| \leqslant q_{e,b} \left\| \overline{w}^{(r-1)} - \overline{w}'^{(r-1)} \right\|. \tag{21}$$

If $q_{e,b} < 1$, we say that SGD is *contractive*. The contractivity property of SGD was previously studied and theoretically proved under some assumptions, e.g., when the surrogate loss function $\tilde{\ell}$ is smooth and strongly convex as in (Dieuleveut et al., 2018; Park et al., 2022; Kozachkov et al., 2022). Note that in general the value of $q_{e,b} < 1$ depends on the running round $r$ and on the learning rate $\eta_{r,t}$. However, in our case, as stated in Section 2, for simplicity we constrain the learning rates $\eta_{r,t}$ to be identical across devices and rounds. This is assumed only for the sake of simplicity, and Theorem 4 can be extended straightforwardly to the case where $q_{e,b}$ could depend on $r$.

Theorem 4 holds for any value of $q_{e,b}$. However, the result becomes particularly interesting when $q_{e,b} \leqslant 1$. It can be shown that for a convex, Lipschitz, smooth (surrogate) loss function, $q_{e,b}$ is at most 1. For the case of strict inequality, i.e. $q_{e,b} < 1$, as mentioned above strongly convex loss functions (with some extra assumptions) indeed satisfy this condition. In FSVM, the considered surrogate loss, which is the hinge loss, is convex, which becomes strongly convex when $L_2$ regularizer is added. Even for the no-regularization case, for which the loss is not strongly convex and we only know theoretically $q_{e,b} \leqslant 1$, the numerical findings suggest that they are strictly less than one. Indeed, we computed the simulated values of $q_{e,b}$ with respect to $R$ as displayed in Fig. 9. There, the values are computed for $K = 50$ clients and averaged over 10 runs. The standard deviation is extremely small. One can observe that all values are below 1.

Next, we state an assumption on the first-order approximation error of the local steps of SGD for one client at one round.

**Assumption 2.** *Consider the* $(K, R, n, e, b)$*-FL-SGD model with* $\mathcal{W} = \mathfrak{B}_d(1)$*. Consider any* $k \in [K]$*,* $r \in [2 : R]$*, and* $\overline{w}^{(r-1)}$*. Fix some dataset* $s_k^{(r)}$ *as well as the mini-batches* $\mathcal{B}_{k,r,t} \subseteq s_k^{(r)}$*,* $t \in [e\,n_R/b]$*. Then, there exists a constant* $\alpha \in \mathbb{R}^+$*, such that for any* $w_\varepsilon \in \mathcal{W}$*,*

$$\left\| w_k'^{(r)} - \left( w_k^{(r)} + \frac{1}{K} \mathsf{D}_{(\overline{w}^{(r-1)}, \{\mathcal{B}_{k,r,t}\}_t)} \, w_\varepsilon \right) \right\| \leqslant \alpha \|w_\varepsilon\|^2 / K^2, \tag{22}$$

*where* $w_k^{(r)}$ *and* $w_k'^{(r)}$ *are the local models of client* $k$ *at round* $r$*, when initialized by* $\overline{w}^{(r-1)}$ *and* $\overline{w}^{(r-1)} + \frac{w_\varepsilon}{K}$*, respectively, and when the same mini-batches* $\{\mathcal{B}_{k,r,t}\}_t$ *are used. Moreover, the matrix* $\mathsf{D}_{(\overline{w}^{(r-1)}, \{\mathcal{B}_{k,r,t}\}_t)} \in \mathbb{R}^{d \times d}$ *represents the "overall gradient of SGD" of one client over one round, that depends on the initialization* $\overline{w}^{(r-1)}$ *and mini-batches* $\{\mathcal{B}_{k,r,t}\}_t$*. We further assume that the spectral norm of this matrix is bounded by* $q_{e,b}$*.*

The inequality (22), which can be obtained using an easy expansion argument applied to the output of the perturbed initialization, is reasonable for moderate or large values of $K$, e.g., by assuming the boundedness of higher-order derivatives.

Moreover, it can be shown that

$$\mathsf{D}_{(\overline{w}^{(r-1)}, \{\mathcal{B}_{k,r,t}\}_t)} = \prod_{t \in [\tau]} \left( \mathrm{I}_d - \mathbf{H}^t_{(\overline{w}^{(r-1)}, \{\mathcal{B}_{k,r,t'}\}_{t' \in [t]})} \right), \tag{23}$$

where $\tau = en_R/b$, $\mathrm{I}_d$ is the $d \times d$ identity matrix, and

$$\mathbf{H}^t_{(\overline{w}^{(r-1)}, \{\mathcal{B}_{k,r,t'}\}_{t' \in [t]})} := \mathrm{Hess}_{g_{\mathcal{B}_{k,r,t}}} \left( W_{k,t}^{(r)} \right).$$

Here $W_{k,t}^{(r)}$ is the model achieved at iteration $t$, when initialized by $\overline{w}^{(r-1)}$ and updated using the mini-batches $\{\mathcal{B}_{k,r,t'}\}_{t' \in [t]}$, $\mathrm{Hess}_g(\cdot)$ denotes the Hessian matrix, and

$$g_{\mathcal{B}}(w) := \frac{\eta}{b} \sum_{z \in \mathcal{B}} \tilde{\ell}(z, w).$$

Lastly, we state an assumption on $K$ being sufficiently large.

---

[3]Note that in general $q_{e,b}$ may depend on $n_R$ as well. However, the dependence is dropped for brevity.

**Assumption 3.** *Suppose Assumptions 1 & 2 hold for some constants $q_{e,b}$ and $\alpha$. Then, for $R \geqslant 2$,[4] the number of clients $K$ satisfies at least one of the following conditions.*

- ***Condition 1:***

$$K^2 \geqslant \min_{\beta \in (0,1]} \max \left( \max_{r \in [R-2]} \frac{\alpha^2 \left((\beta+1)q_{e,b}^2\right)^r}{\beta q_{e,b}^2}, \max_{r \in [R-1]} \frac{6\alpha B\left(q_{e,b}^r - \left((\beta+1)q_{e,b}^2\right)^r\right)}{\left(q_{e,b} - (\beta+1)q_{e,b}^2\right)\theta} \right), \quad (24)$$

- ***Condition 2:*** *There exists some $\nu \geqslant 0$ such that $\frac{1}{K^2} \leqslant \frac{1-q_{e,b}^2}{\alpha} - \nu$ and*

$$K^2 \geqslant \max_{r \in [R-1]} \frac{6\alpha B\left(q_{e,b}^r - (1-\alpha\nu)^r\right)}{\left(q_{e,b} - (1-\alpha\nu)\right)\theta}. \quad (25)$$

This assumption is merely used for the simplification of the technical steps of the proof and more precisely the recursive steps in the part Bounding (73) in the proof of Theorem 4 in Appendix F.4. We *expect* Theorem 4 to hold for moderate values of $K$ as well. It is insightful to note that in many cases the terms involving the maximization over $r$ in (24) and (25) are either maximized for $r = 1$ or can be bounded by $R$, and hence the conditions can be simplified. As an example, whenever $q_{e,b} < 1$, then a sufficient condition to satisfy (24) is to have

$$K^2 \geqslant \min_{\beta \in (0,\min(1,1/q_{e,b}^2-1)]} \max \left( \alpha^2(\beta+1)/\beta, \max_{r \in [R-1]} \frac{6\alpha Br \max\left(q_{e,b}, (\beta+1)q_{e,b}^2\right)^{r-1}}{\theta} \right).$$

Note that the second term goes to zero as $r \to \infty$. The above condition can be further made loose to achieve the below sufficient condition

$$K^2 \geqslant \min_{\beta \in (0,\min(1,1/q_{e,b}^2-1)]} \max \left( \alpha^2(\beta+1)/\beta, 6\alpha BR/\theta \right).$$

# D ADDITIONAL THEORETICAL RESULTS

## D.1 A RATE-DISTORTION THEORETIC TAIL BOUND ON FL

In this section, we state a rate-distortion theoretic tail bound. For this bound, let the randomness of the learning algorithm used by client $k \in [k]$ during round $r \in [R]$ be represented by some variable $U_k^{(r)}$, assumed to be independent of every other random variable. Denote $\mathbf{U} := U_{[K]}^{[R]}$. This assumption implies that $W_k^{(r)}$ is a *deterministic* function of $S_k^{(r)}, U_k^{(r)}$, and the initialization $\overline{W}^{(r-1)}$. For any fixed $\delta > 0$, denote $\mathcal{G}_{\mathbf{S},\mathbf{U},\mathbf{W}}^\delta := \{\nu_{\mathbf{S},\mathbf{U},\mathbf{W}} : D_{KL}\left(\nu_{\mathbf{S},\mathbf{U},\mathbf{W}} \| P_{\mathbf{S},\mathbf{U},\mathbf{W}}\right) \leqslant \log(1/\delta)\}$. Note that under $\nu$, for every $k \in [K]$ and $r \in [R]$, $W_k^{(r)}$ is a *deterministic* function of $S_k^{(r)}, U_k^{(r)}$ and $\overline{W}^{(r-1)}$. Moreover, $\overline{W}^{(r)}$ is a deterministic function of $W_{[K]}^{(r)}$. Hence, $\nu_{\mathbf{W}|\mathbf{S},\mathbf{U}} = P_{\mathbf{W}|\mathbf{S},\mathbf{U}}$. Also, let for given $\nu_{\mathbf{S},\mathbf{U},\mathbf{W}}$

$$\mathfrak{RD}(\nu_{\mathbf{S},\mathbf{U},\mathbf{W}}, k, r, \epsilon) = \inf_{p_{\hat{W}_k^{(r)}|S_k^{(r)}, U_k^{(r)}, \overline{W}^{(r-1)}}} I(S_k^{(r)}; \hat{W}_k^{(r)}|U_k^{(r)}, \overline{W}^{(r-1)}), \quad (26)$$

where the mutual information is calculated with respect to $\nu_{S_k^{(r)}, U_k^{(r)}, \overline{W}^{(r-1)}} p_{\hat{W}_k^{(r)}|S_k^{(r)}, \overline{W}^{(r-1)}}$ and the infimum is over all conditional Markov kernels $p_{\hat{W}_k^{(r)}|S_k^{(r)}, \overline{W}^{(r-1)}}$ that satisfy

$$\mathbb{E}_{\nu_{\mathbf{S},\mathbf{U},\mathbf{W}}} \left[ \operatorname{gen}(S_k^{(r)}, \overline{W}^{(R)}) \right] - \mathbb{E}_{\nu_{\mathbf{S},\mathbf{U}}(\nu p)_{k,r}} \left[ \operatorname{gen}(S_k^{(r)}, \hat{\overline{W}}^{(R)}) \right] \leqslant \epsilon, \quad (27)$$

with

$$(\nu p)_{k,r} = \nu_{\overline{W}^{(r-1)}, W_{[K]\backslash k}^{(r)}|S_{[K]}^{[r]}\backslash S_k^{(r)}, U_{[K]}^{[r]}\backslash U_k^{(r)}} \, p_{\hat{W}_k^{(r)}|S_k^{(r)}, U_k^{(r)}, \overline{W}^{(r-1)}} \, \nu_{\hat{\overline{W}}^{(R)}|\hat{W}_k^{(r)}, W_{[K]\backslash k}^{(r)}, S_{[K]}^{[r+1:R]}, U_{[K]}^{[r+1:R]}}.$$

It can be easily seen that this definition is consistent with (11).

---

[4]No condition is needed for $R = 1$.

**Theorem 6.** *Consider a $(P_{\mathbf{W}|\mathbf{S}}, K, R, n)$-FL model with distributed dataset $\mathbf{S} \sim P_{\mathbf{S}}$. Suppose that the loss $\ell(Z_k, w)$ is $\sigma$-subgaussian for every $w \in \mathcal{W}$ and any $k \in [K]$. Fix any distortion level $\epsilon \in \mathbb{R}$. Then, for any $\delta > 0$, with probability at least $(1 - \delta)$, we have*

$$\text{gen}(\mathbf{S}, \overline{W}^{(R)}) \leqslant \sqrt{\frac{\sup_{\nu_{\mathbf{S},\mathbf{U},\mathbf{W}} \in \mathcal{G}_{\mathbf{S},\mathbf{U},\mathbf{W}}^{\delta}} \sum_{k\in[K], r\in[R]} \mathfrak{RD}(\nu_{\mathbf{S},\mathbf{U},\mathbf{W}}, k, r, \epsilon_{k,r}) + \log(1/\delta)}{nK/(2\sigma^2)}} + \epsilon,$$

*for any $\{\epsilon_{k,r}\}_{k\in[K],r\in[R]} \subset \mathbb{R}$ such that*

$$\frac{1}{KR} \sum_{k\in[K]} \sum_{r\in[R]} \epsilon_{k,r} \leqslant \epsilon. \tag{28}$$

In a sense, this result, proved in Appendix F.6, says that in order to have a good generalization performance (not only in-expectation as in Theorem 3 but also in terms of tails), the algorithm should be *compressible* not only under the true distribution $P_{\mathbf{S},\mathbf{W}}$ but also for all *all* those distributions $\nu_{\mathbf{S},\mathbf{U}}$ that are in the vicinity of $P_{\mathbf{S},\mathbf{U}}$.

### D.2 LOSSY COMPRESSION BOUNDS REVISITED

In all *lossy* bounds, *i.e.,* Theorems 2, 3, and 6, we had three common assumptions on the *quantizations* of the local models:

1. $\hat{\mathcal{W}} \subseteq \mathcal{W}$,
2. the loss and generalization error definitions considered for the models $\hat{\mathcal{W}}$ are the same as the one considered for $\mathcal{W}$,
3. the *quantization* of the model $W_k^{(r)}$ is performed according to $p_{\hat{W}_k^{(r)}|\overline{W}^{(r-1)}, S_k^{(r)}}$.

These choices are natural and intuitive ones for the FL learning algorithms. However, it turns out that all these conditions can be relaxed, *i.e.,* in the following extended results, we consider

1. $\hat{\mathcal{W}}$ to be any arbitrary set in $\mathbb{R}^m$, for some arbitrary $m \in \mathbb{N}^*$,
2. the loss function $\hat{\ell}(z, \hat{w}) \colon \mathcal{Z} \times \hat{\mathcal{W}} \to \mathbb{R}^+$ to be defined arbitrarily, possibly different than $\ell$. The generalization error for $\hat{\mathcal{W}}$ is defined then with respect to this loss function,
3. for every $k \in [K]$ and $r \in [R]$, quantizing *the aggregated global model* $\hat{\overline{W}}^{(R)}$ according to

$$p_{\hat{\overline{W}}_k^{(R)}|\overline{W}^{(r-1)}, S_k^{(r)}, V_k^{(r)}, W_{[K]\backslash k}^{(r)}, S_{[K]}^{[r+1:R]}, V_{[K]}^{[r+1:R]}}, \tag{29}$$

where $V_k^{(r)} \in \mathcal{V}$ presents some mutually independent available randomness used by client $k$ during round $r$. Note that unlike the Section 3.2, they do not necessarily represent *all* the randomness used by client $k$ during round $r$, *e.g.,* they can be set to some constants or they can be set to be $U_k^{(r)}$ which is all the used randomness. Denote $\mathbf{V} = V_{[K]}^{[R]}$. Note that

$$P_{\mathbf{S},\mathbf{V},\mathbf{W}} = P_{\mathbf{S}} P_{\mathbf{V}} P_{\mathbf{W}|\mathbf{S},\mathbf{V}}. \tag{30}$$

The new choice of the quantization distribution implies that for every $k \in [K]$ and $r \in [R]$, in a sense we fix $\overline{W}^{(r-1)}, W_{[K]\backslash k}^{(r)}, V_{[K]}^{[r+1:R]}, S_{[K]}^{[r+1:R]}$, and consider *quantization* of the different global models obtained for different $S_k^{(r)}$.

Now, we state the extended results. For the following results, consider arbitrary set $\hat{\mathcal{W}} \in \mathbb{R}^m$, loss function $\hat{\ell}(z, \hat{w})$, and the generalization error that is defined with respect to this loss function. Next, we use the shorthand notation:

$$\mathfrak{V}_{k,r} = \left( V_k^{(r)}, \overline{W}^{(r-1)}, W_{[K]\backslash k}^{(r)}, S_{[K]}^{[r+1:R]}, V_{[K]}^{[r+1:R]} \right). \tag{31}$$

First, we state the extended version of Theorem 3, that is used in the proof of Theorem 4. The proofs of these extended results follow the same lines as the original results. As an example, we show the proof of Proposition 1 in Appendix F.7. The rest of the proofs follow the proof of their corresponding theorems.

For $k \in [K]$, $r \in [R]$ and $\epsilon \in \mathbb{R}$, let $\mathcal{P}_{k,r}$ include all conditional Markov kernels

$$p_{\hat{\overline{W}}_k^{(R)} | S_k^{(r)}, \mathfrak{V}_{k,r}}, \tag{32}$$

that satisfy

$$\mathbb{E}_{\mathbf{S}, \mathbf{V}, \mathbf{W}, \hat{\overline{W}}^{(R)}}\left[ \text{gen}(S_k^{(r)}, \overline{W}^{(R)}) - \text{gen}(S_k^{(r)}, \hat{\overline{W}}^{(R)}) \right] \leqslant \epsilon, \tag{33}$$

where the joint distribution of $\mathbf{S}, \mathbf{V}, \mathbf{W}, \hat{\overline{W}}^{(R)}$ factorizes as $P_{\mathbf{S}, \mathbf{V}, \mathbf{W}} p_{\hat{\overline{W}}_k^{(R)} | S_k^{(r)}, \mathfrak{V}_{k,r}}$.

Now, define the rate-distortion function

$$\mathfrak{RD}^{\star}(P_{\mathbf{S}, \mathbf{W}}, k, r, \epsilon) = \inf_{\mathcal{P}_{k,r}} I(S_k^{(r)}; \hat{\overline{W}}_k^{(R)} | \mathfrak{V}_{k,r}), \tag{34}$$

where the mutual information is evaluated with respect to

$$P_{S_k^{(r)}} P_{\mathfrak{V}_{k,r}} p_{\hat{\overline{W}}_k^{(R)} | S_k^{(r)}, \mathfrak{V}_{k,r}}. \tag{35}$$

**Proposition 1.** *For any* $(P_{\mathbf{W}|\mathbf{S}}, K, R, n)$*-FL model with distributed dataset and randomness* $(\mathbf{S}, \mathbf{V}) \sim P_{\mathbf{S}} P_{\mathbf{V}}$*, if the loss* $\ell(Z_k, w)$ *is* $\sigma$*-subgaussian for every* $w \in \mathcal{W}$ *and any* $k \in [K]$*, then for every* $\epsilon \in \mathbb{R}$ *it holds that*

$$\mathbb{E}_{\mathbf{S}, \mathbf{W} \sim P_{\mathbf{S}, \mathbf{W}}}\left[ \text{gen}(\mathbf{S}, \overline{W}^{(R)}) \right] \leqslant \sqrt{\frac{2\sigma^2 \sum_{k \in [K], r \in [R]} \mathfrak{RD}^{\star}(P_{\mathbf{S}, \mathbf{W}}, k, r, \epsilon_{k,r})}{nK}} + \epsilon,$$

*for any set of parameters* $\{\epsilon_{k,r}\}_{k \in [K], r \in [R]} \subset \mathbb{R}$ *which satisfy*

$$\frac{1}{KR} \sum_{k \in [K]} \sum_{r \in [R]} \epsilon_{k,r} \leqslant \epsilon.$$

Next, we state the extension of Theorem 2.

**Proposition 2.** *Suppose that* $\ell(z, w) \in [0, C] \subset \mathbb{R}^+$*. Consider any set of priors* $\{\mathsf{P}_{k,r}\}_{k \in [K], r \in [R]}$ *where* $\mathsf{P}_{k,r}$ *is a conditional prior on* $\hat{\overline{W}}^{(R)}$ *given* $\mathfrak{V}_{k,r}$*. Fix any distortion level* $\epsilon \in \mathbb{R}^+$*. Consider any* $(P_{\mathbf{W}|\mathbf{S}}, K, R, n)$*-FL model and any choices of*

$$\left\{ p_{\hat{\overline{W}}_k^{(R)} | S_k^{(r)}, \mathfrak{V}_{k,r}} \right\}_{k \in [K], r \in [R]}, \tag{36}$$

*that for every* $\mathbf{s} \in \mathcal{Z}^{nK}$ *satisfy the* $\epsilon$*-distortion criterion*

$$\left| \mathbb{E}_{P_{\mathbf{W}|\mathbf{s}}}\left[ \text{gen}(\mathbf{s}, \overline{W}^{(R)}) \right] - \frac{1}{KR} \sum_{k \in [K], r \in [R]} \mathbb{E}_{(Pp)_{k,r}^{\star}}\left[ \text{gen}(s_k^{(r)}, \hat{\overline{W}}^{(R)}) \right] \right| \leqslant \epsilon/2, \tag{37}$$

*where*

$$(Pp)_{k,r}^{\star} = P_{\overline{W}^{(r-1)}, W_{[K] \backslash k}^{(r)} | s_{[K]}^{[r-1]}, s_{[K] \backslash k}^{(r)}} P_{V_k^{(r)}, V_{[K]}^{[r+1:R]}} p_{\hat{\overline{W}}^{(R)} | S_k^{(r)}, \mathfrak{V}_{k,r}}.$$

*Then, with probability at least* $(1 - \delta)$ *over* $\mathbf{S} \sim P_{\mathbf{S}}$*, we have that* $\mathbb{E}_{\mathbf{W} \sim P_{\mathbf{W}|\mathbf{S}}}\left[ \text{gen}(\mathbf{S}, \overline{W}^{(R)}) \right]$ *is upper bounded by*

$$\sqrt{\frac{\frac{1}{KR} \sum_{k \in [K], r \in [R]} \mathbb{E}\left[ D_{KL}\left( p_{\hat{\overline{W}}_k^{(R)} | S_k^{(r)}, \mathfrak{V}_{k,r}} \middle\| \mathsf{P}_{k,r} \right) \right] + \log(\frac{\sqrt{2n}}{\sqrt{R}\delta})}{(2n/R - 1)/C^2}} + \epsilon,$$

*where the expectations are with respect to* $P_{\overline{W}^{(r-1)}, W_{[K] \backslash k}^{(r)} | s_{[K]}^{[r-1]}, s_{[K] \backslash k}^{(r)}} P_{V_k^{(r)}, V_{[K]}^{[r+1:R]}}$*.*

Finally, we present the extension of the rate-distortion theoretic tail bound. For the rest of this section, assume that $V_k^{(r)} = U_k^{(r)}$, for $k \in [K]$ and $r \in [R]$ and for a better clarity, denote $\mathfrak{V}_{k,r}$ by $\mathfrak{U}_{k,r}$ for this choice.

Consider any $\nu_{\mathbf{S},\mathbf{U},\mathbf{W}} \in \mathcal{G}_{\mathbf{S},\mathbf{U},\mathbf{W}}^{\delta}$. For this distribution and $k \in [K]$ and $r \in [R]$, let the $\mathcal{P}_{\nu,k,r}$ include all conditional Markov kernels

$$p_{\hat{\overline{W}}_k^{(R)} | S_k^{(r)}, \mathfrak{U}_{k,r}}, \tag{38}$$

that satisfy

$$\mathbb{E}_{\mathbf{S},\mathbf{U},\mathbf{W},\hat{\overline{W}}^{(R)}}\left[\text{gen}(S_k^{(r)}, \overline{W}^{(R)}) - \text{gen}(S_k^{(r)}, \hat{\overline{W}}^{(R)})\right] \leqslant \epsilon, \tag{39}$$

where the expectation is with respect to $\nu_{\mathbf{S},\mathbf{U},\mathbf{W}} p_{\hat{\overline{W}}_k^{(R)} | S_k^{(r)}, \mathfrak{U}_{k,r}}$. Now, let

$$\mathfrak{RD}^{\star}(\nu_{\mathbf{S},\mathbf{U},\mathbf{W}}, k, r, \epsilon) = \inf_{\mathcal{P}_{\nu,k,r}} I(S_k^{(r)}; \hat{\overline{W}}_k^{(R)} | \mathfrak{U}_{k,r}), \tag{40}$$

where the mutual information is calculated with respect to $\nu_{S_k^{(r)}, \mathfrak{U}_{k,r}} p_{\hat{\overline{W}}_k^{(R)} | S_k^{(r)}, \mathfrak{U}_{k,r}}$.

**Proposition 3.** *Consider a $(P_{\mathbf{W}|\mathbf{S}}, K, R, n)$-FL model with distributed dataset and randomness $(\mathbf{S}, \mathbf{U}) \sim P_{\mathbf{S}} P_{\mathbf{U}}$. Suppose that the loss $\ell(Z_k, w)$ is $\sigma$-subgaussian for every $w \in \mathcal{W}$ and any $k \in [K]$. Fix any distortion level $\epsilon \in \mathbb{R}$. Then, for any $\delta > 0$, with probability at least $(1 - \delta)$, we have*

$$\text{gen}(\mathbf{S}, \overline{W}^{(R)}) \leqslant \sqrt{\frac{\sup_{\nu_{\mathbf{S},\mathbf{U},\mathbf{W}} \in \mathcal{G}_{\mathbf{S},\mathbf{U},\mathbf{W}}^{\delta}} \sum_{k \in [K], r \in [R]} \mathfrak{RD}^{\star}(\nu_{\mathbf{S},\mathbf{U},\mathbf{W}}, k, r, \epsilon_{k,r}) + \log(1/\delta)}{nK/(2\sigma^2)}} + \epsilon,$$

*for any $\{\epsilon_{k,r}\}_{k \in [K], r \in [R]} \subset \mathbb{R}$ such that*

$$\frac{1}{KR} \sum_{k \in [K]} \sum_{r \in [R]} \epsilon_{k,r} \leqslant \epsilon. \tag{41}$$

# E  ADDITIONAL EXPERIMENTAL RESULTS & DETAILS

In this section, we first provide experimental and implementation details that were omitted in the main text. Then, some additional numerical simulations are shown and described.

## E.1  EXPERIMENTAL & IMPLEMENTATION DETAILS

**Tasks description**  In our experiments, we simulate a Federated Learning (FL) framework, according to the setup of this paper (see Section 1), on a single machine, where each "client" is an instance of the SVM model, equipped with a dataset. All tasks are performed on the same machine and we do not consider any communication constraint as it is not of interest in this paper. Once all clients' models are trained, their weights are averaged and used by another instance of the SVM model (meant to be the one at the parameter server). This is utilized for computing the empirical risk $\hat{\mathcal{L}}_\theta(\mathbf{S}, \overline{W})$ and population risk $\mathcal{L}(\overline{W})$. In particular, $\mathcal{L}(\overline{W})$ is estimated using a test set of fixed size $N_{test}$. Using these quantities, the generalization error $\text{gen}(\mathbf{S}, \overline{W})$ is computed.

The (local) dataset of client # $k \in [K]$, denoted as $S_k$, is composed of $n$ samples that are drawn uniformly without replacement from $S$. In other words, the dataset $S$ is initially split into $K$ subsets. Each local dataset $S_k$ is then further split into $S_k^{(r)}$, $r \in [R]$ *i.e.*, the training samples used at each round $r$ by client $k$.

**Datasets**  The learning task is binary image classification for the FSVM experiments. More precisely, we trained the SVM model to classify images of digits 1 and 6 from the standard MNIST (LeCun et al., 2010), which results in approximately 12000 training images and 2000 test images. Experiments were performed for different pairs of digits, giving similar results, and thus are not reported. For the additional experiments in Section 5, we considered the whole CIFAR-10 dataset (10-class classification) (Krizhevsky & Hinton, 2009) composed of 50000 training images and 10000 test images.

**Model**  Each client is equipped with SVM with Radial Basis Function (RBF) kernel (FSVM experiments) or ResNet-56 with batch normalization (additional experiments with NNs). Stochastic Gradient Descent (SGD) is used for optimization.

| Hyperparameter | Symbol | Value |
|---|---|---|
| Local epochs | $e$ | 40 |
| Learning rate | $\eta$ | 0.01 |
| Batch size | $b$ | 1 |
| Kernel parameter | $\gamma$ | 0.05 |
| Dimension of the approximated kernel feature space | $d$ | 4000 |

(a) FSVM

| Hyperparameter | Symbol | Value |
|---|---|---|
| Number of clients | $K$ | 16 |
| Epochs | $e$ | 150 |
| Learning rate | $\eta$ | 1.0 |
| Batch size | $b$ | 128 |

(b) Additional experiments (ResNet-56 with CIFAR-10)

Table 1: Hyperparameters for experiments in Section 5

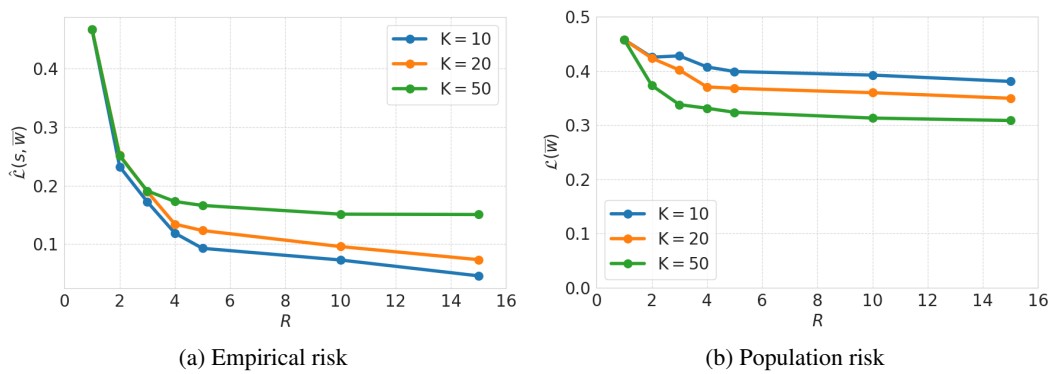

(a) Empirical risk

(b) Population risk

Figure 4: Empirical and population risk of FSVM w.r.t. $R$, for $n = 100$

**Training and hyperparameters**  All local models were trained with the same hyperparameters, which are of common usage, and are given in Table 1. The data is scaled and normalized so as to get zero-mean and unit variance.

The training scheme of the SVM experiments follows the setup described in 4. All clients have the same fixed learning rate $\eta_t = \eta$. Each local model is trained for $e$ local epoch at each round $r \in [R]$. A local epoch refers to a pass over the training samples at each round $r \in [R]$ *i.e.,* $S_k^{(r)}$. Before each local epoch, $S_k^{(r)}$ is shuffled. The training algorithm ends when each client # $k \in [K]$ has done $e$ local epochs on each of the subsets $S_k^{(r)}$, $r \in [R]$.

For the ResNet-56 simulations, we did not stick to the previous setup and used a more standard training scheme. Precisely, each client does only one pass over its data $S_k^{(r)}$ at each round $r = 1, \ldots, R$. When the $R$ rounds are completed, a *global* epoch is achieved, and each client starts again to train with $S_k^{(1)}$ until another epoch is completed.

**Hardware and implementation**  We performed our experiments on a server equipped with 56 CPUs Intel Xeon E5-2690v4 2.60GHz and 4 GPUs Nvidia Tesla P-100 PCIe 16GB.

Our implementations use the Python language. The open-source machine learning library scikit-learn (Pedregosa et al., 2011) is used to implement SVM. In particular, we use *SGDClassifier* to implement SGD optimization and *RBFSampler* for the Gaussian kernel feature map approximation. The deep learning library PyTorch is used for the CIFAR-10 experiments using ResNet-56.

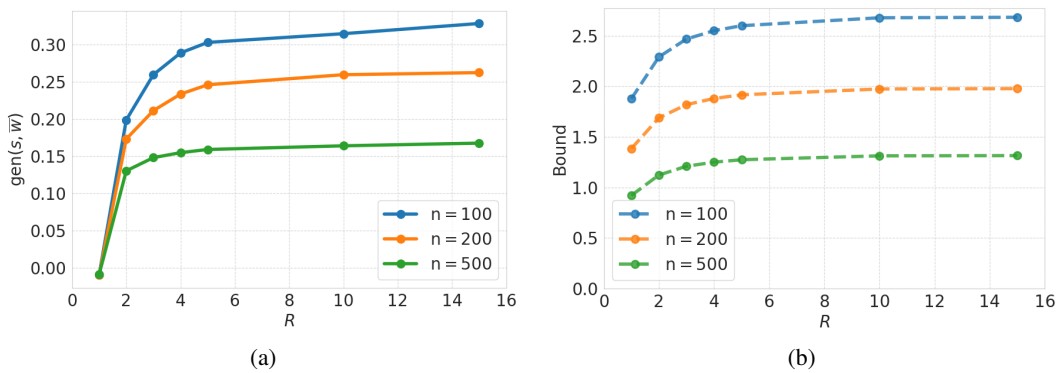

Figure 5: Generalization error of FSVM and bound of Theorem 4 w.r.t. $R$, for $K = 10$

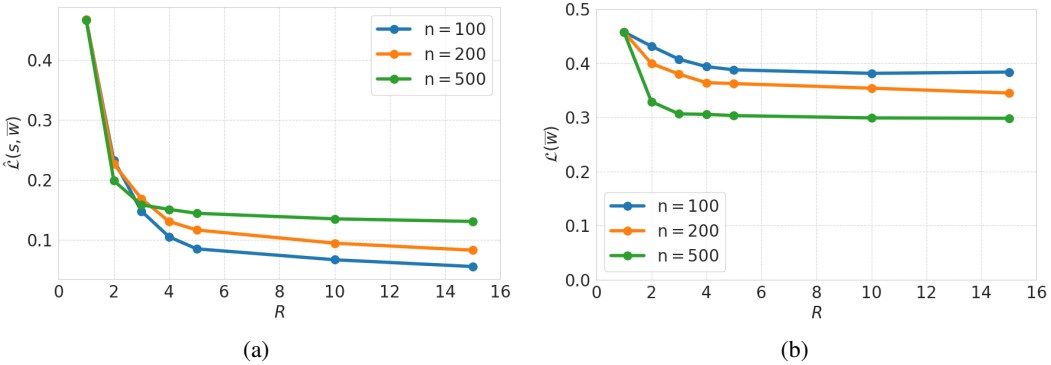

Figure 6: (a) Empirical risk and (b) population risk of FSVM w.r.t. $R$, for $K = 10$

### E.2    EMPIRICAL AND POPULATION RISK FOR FSVM

Fig. 4 shows the empirical risk $\hat{\mathcal{L}}(\mathbf{S}, \overline{W}^{(R)})$ and the (estimated) population risk $\mathcal{L}(\overline{W}^{(R)})$, as functions of $R$, for the FSVM experiment in Section 5. The population risk is estimated using the entire test dataset of MNIST, for the two classes, with $N_{test} = 2093$. Interestingly, while the empirical risk keeps decreasing for $R \geqslant 5$, the population risk no longer decreases beyond $R = 5$. This numerically validates the claim that the population risk may "converge" faster than "empirical risk". Hence, fewer rounds may be needed, if one can effectively take the "estimated" generalization error into account.

### E.3    GENERALIZATION ERROR OF FSVM FOR DIFFERENT VALUES OF $n$

Fig. 5 shows the (expected) generalization error (Fig. 5a) of FSVM and the bound of Theorem 4 (Fig. 5b) with respect to $R$ for $K = 10$ fixed and $n \in \{100, 200, 500\}$. Fig. 6 shows the empirical risk and the estimated population risk.

These plots suggest that our observed behaviors in Section 5 hold for fixed $K$ and for various values of $n$. That is:

- The generalization error increases with $R$ for any fixed $n$,
- For any fixed $R$, both the generalization error and bound improve as $n$ increases,
- Above findings are compatible with the behavior predicted by the bound of Theorem 4,
- While the empirical risk keeps decreasing for $R \geqslant 5$, the population seems to have reached its minimum for $R = 5$.

Note that similar results were obtained for other values of $n$.

### E.4    EXPERIMENTS FOR A HETEROGENEOUS DATA SETTING

We simulate a heterogeneous data setting (*i.e.*, non-i.i.d.) by adding Gaussian white noise with standard deviation $\sigma = 0.2$ to the training and testing images of a fraction $f = 0.2$ of the clients. Therefore, as in the

setup described in Section 1, the data distributions of clients $\mu_k$, $k \in [K]$ are different. The hyperparameters remain, however, identical as we aim at comparing the numerical results with the homogeneous data case.

Similar to Section 5, we compute the generalization error of FSVM, as well as the bound of Theorem 4, with respect to $R$ (see Fig. 7) and show the corresponding train and test risks (see Fig. 8).

The same observations can be made for the experiments in the homogeneous setting (Section 5). Remark that, on one hand, in comparison to Fig. 4, the empirical risk values on Fig. 8 are larger. This is expected, as the final global model is a simple arithmetic average of local models and hence may struggle to achieve the optimum of each local objective function in the presence of data heterogeneity. Therefore, more communication rounds $R$ are needed to achieve the same level of optimization as in the homogeneous case. On the other hand, the generalization error values (see Fig. 7a) are smaller than in the i.i.d. setup (see Fig. 2a). The global model is indeed less likely to overfit due to the "noise" that is injected into some of the local datasets.

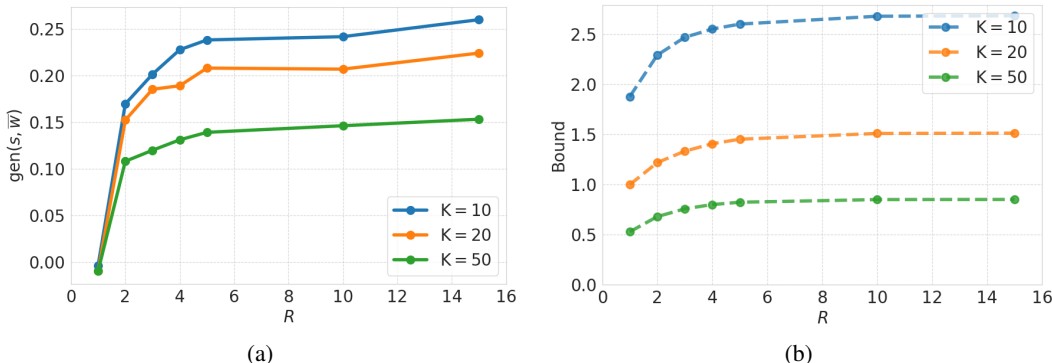

(a)                (b)

Figure 7: Generalization error of FSVM (non-iid setting) and bound of Theorem 4 w.r.t. $R$, for $n = 100$

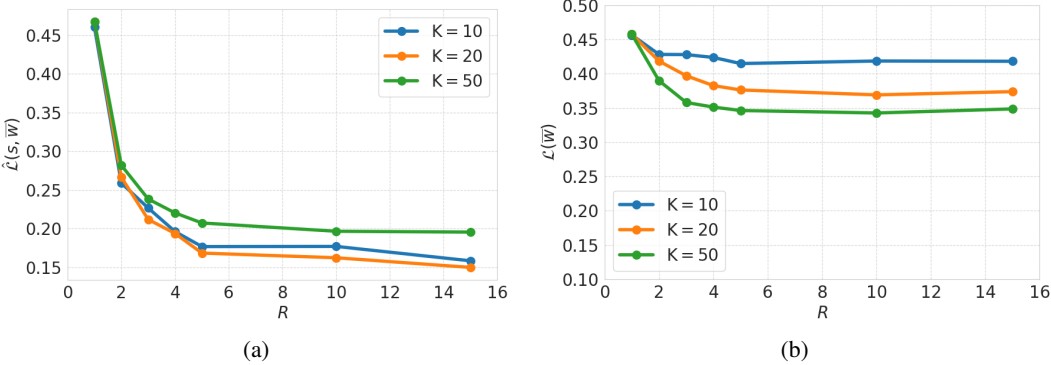

(a)                (b)

Figure 8: Empirical risk and population risk for FSVM (non-iid setting) w.r.t. $R$, for $n = 100$

### E.5 EXPERIMENTAL VERIFICATION OF ASSUMPTION 1 FOR THEOREM 4

In this section, we show the computed values of the contraction coefficient $q_{e,b}$ in Assumption 1 of Theorem 4 with respect to $R$ and with $K = 50$ clients. The values are averaged over 10 runs. The standard deviation is extremely small. The reader is referred to Appendix C for the discussion about the assumption.

### E.6 ADDITIONAL EXPERIMENTS FOR RESNET-56

The experiments on ResNet-56 presented on Fig. 3 were conducted for several sets of hyperparameters and showed similar behaviors. For example, we provide on Fig. 10 the generalization error and the empirical and population risks for two different values of the learning rate $\eta$. The other hyperparameters are similar to the experiment of Fig. 3 and are hence provided in Table 1b.

One can observe the same increasing trend of the generalization error with respect to $R$ that was observed in the previous experiments. Also, the presence of a global minimizer $R^*$ for the population risk is noticeable.

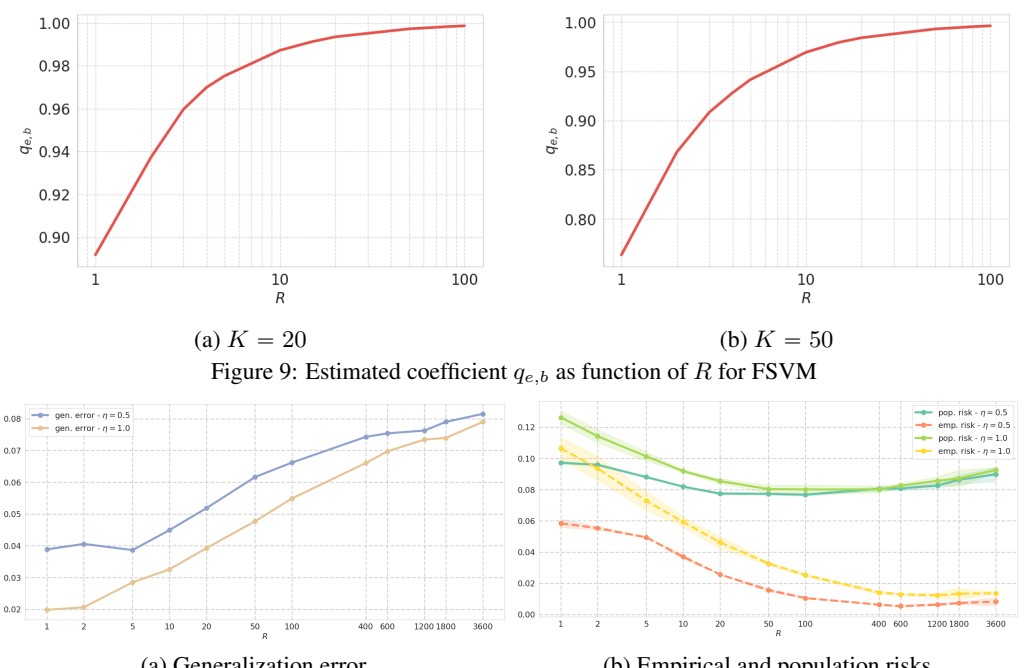

(a) $K = 20$          (b) $K = 50$

Figure 9: Estimated coefficient $q_{e,b}$ as function of $R$ for FSVM

(a) Generalization error      (b) Empirical and population risks

Figure 10: Performance of FL-SGD with ResNet-56 local models as a function of $R$ for different learning rates

## F  PROOFS

In this section, we provide the proofs of all theoretical results established in the paper and the appendices, by the order of their appearances.

### F.1  PROOF OF THEOREM 1

Without loss of generality assume $\sigma = 1/2$, otherwise, consider the properly scaled loss function.

#### F.1.1  PART I.

*Proof.* The proof of this theorem is a particular case of Theorem 2, proved in Appendix F.2 for bounded losses and is similar for the $\sigma$-subgaussian case. We, however, provide the proof for the sake of completeness.

Similar to the proof of (Sefidgaran & Zaidi, 2023, Theorem 6), it is easy to see that for any choice of the prior $\mathsf{P}_{k,r}$, it suffices to only consider an arbitrary $P_{\mathbf{W}|\mathbf{S}}$. Now, let $\lambda := 2n/R - 1$. Let the RHS of the bound as $\sqrt{\Delta(\mathbf{S})}$, *i.e.,* let

$$\Delta(\mathbf{S}) := \frac{\frac{1}{KR} \sum_{k \in [K], r \in [R]} \mathbb{E}_{\overline{W}^{(r-1)} \sim P_{\overline{W}^{(r-1)}|S_{[K]}^{[r-1]}}} \left[ D_{KL}\left( p_{W_k^{(r)}|S_k^{(r)}, \overline{W}^{(r-1)}} \| \mathsf{P}_{k,r} \right) \right] + \log\left(\frac{\sqrt{2n}}{\sqrt{R}\delta}\right)}{\lambda}.$$

Recall that

$$\mathbb{E}_{\mathbf{W} \sim P_{\mathbf{W}|\mathbf{S}}} \left[ \text{gen}(\mathbf{S}, \overline{W}^{(R)}) \right] = \frac{1}{KR} \sum_{k \in [K], r \in [R]} \mathbb{E}_{\mathbf{W} \sim P_{\mathbf{W}|\mathbf{S}}} \left[ \text{gen}(S_k^{(r)}, \overline{W}^{(R)}) \right].$$

For any fixed $\delta > 0$, denote

$$\mathcal{G}_{\mathbf{S}}^{\delta} := \{\nu_{\mathbf{S}} : D_{KL}(\nu_{\mathbf{S}} \| P_{\mathbf{S}}) \leqslant \log(1/\delta)\}, \tag{42}$$

as the set of all distributions $\nu_{\mathbf{S}}$ over $\mathcal{Z}^{nK}$, whose KL-divergence with $P_{\mathbf{S}} = \mu^{\otimes nK}$ does not exceed $\log(1/\delta)$. Then,

$$\log \mathbb{P}_{\mathbf{S}} \left( \mathbb{E}_{\mathbf{W} \sim P_{\mathbf{W}|\mathbf{S}}} \left[ \text{gen}(\mathbf{S}, \overline{W}^{(R)}) \right] > \sqrt{\Delta(\mathbf{S})} \right)$$

$$= \log \mathbb{P}_{\mathbf{S}} \left( \mathbb{E}_{\mathbf{W} \sim P_{\mathbf{W}|\mathbf{S}}} \left[ \text{gen}(\mathbf{S}, \overline{W}^{(R)}) \right]^2 > \Delta(\mathbf{S}) \right)$$

$$\overset{(a)}{\leqslant} \max \left( \log(\delta), \sup_{\nu_{\mathbf{S}} \in \mathcal{G}_{\mathbf{S}}^{\delta}} \left\{ -D_{KL}(\nu_{\mathbf{S}} \| P_{\mathbf{S}}) - \lambda \mathbb{E}_{\nu_{\mathbf{S}}} \left[ \Delta(\mathbf{S}) - \mathbb{E}_{\mathbf{W} \sim P_{\mathbf{W}|\mathbf{S}}} \left[ \text{gen}(\mathbf{S}, \overline{W}^{(R)}) \right]^2 \right] \right\} \right),$$

$$\overset{(b)}{\leqslant} \max \left( \log(\delta), \sup_{\nu_{\mathbf{S}} \in \mathcal{G}_{\mathbf{S}}^{\delta}} \left\{ -D_{KL}(\nu_{\mathbf{S}} \| P_{\mathbf{S}}) - \lambda \mathbb{E}_{\nu_{\mathbf{S}}} \left[ \Delta(\mathbf{S}) - \frac{1}{KR} \sum_{k,r} \mathbb{E}_{\mathbf{W} \sim P_{\mathbf{W}|\mathbf{S}}} \left[ \text{gen}(S_k^{(r)}, \overline{W}^{(R)})^2 \right] \right] \right\} \right),$$

$$(43)$$

where $(a)$ is shown in the next lemma, proved in Appendix F.8, and $(b)$ is due to the Jensen inequality.

**Lemma 1.** *The inequality* (43) *holds.*

Now, showing that the second term of (43) is bounded by $\log(\delta)$ completes the proof. For any $\mathbf{s}$ and any $k \in [K]$ and $r \in [R]$, we have

$$\lambda \mathbb{E}_{\nu_{\mathbf{S}} P_{\mathbf{W}|\mathbf{S}}} \left[ \text{gen}(S_k^{(r)}, \overline{W}^{(R)})^2 \right] \overset{(a)}{\leqslant} D_{KL} \left( \nu_{\mathbf{S}} P_{\mathbf{W}|\mathbf{S}} \| P_{\mathbf{S}} (P\mathsf{P}_{k,r})_{k,r} \right) + \log \mathbb{E}_{P_{\mathbf{S}} (P\mathsf{P}_{k,r})_{k,r}} \left[ e^{\lambda \text{gen}(s_k^{(r)}, \hat{\overline{W}}^{(R)})^2} \right]$$

$$\overset{(b)}{\leqslant} D_{KL} \left( \nu_{\mathbf{S}} P_{\mathbf{W}|\mathbf{S}} \| P_{\mathbf{S}} (P\mathsf{P}_{k,r})_{k,r} \right) + \log(\sqrt{2n/R}) \qquad (44)$$

$$= \mathbb{E}_{\nu_{\mathbf{S}} P_{\overline{W}^{(r-1)}|S_{[K]}^{[r-1]}}} \left[ D_{KL} \left( P_{W_k^{(r)}|S_k^{(r)}, W^{(r-1)}} \| \mathsf{P}_{W_k^{(r)}|\overline{W}^{(r-1)}} \right) \right]$$

$$+ D_{KL} \left( \nu_{\mathbf{S}} \| P_{\mathbf{S}} \right) + \log(\sqrt{2n/R}), \qquad (45)$$

where $(a)$ holds by Donsker-Varadhan's inequality, and where

$$(P\mathsf{P}_{k,r})_{k,r} := P_{\overline{W}^{(r-1)}, W_{[K]\backslash k}^{(r)}|S_{[K]}^{[r-1]}, S_{[K]\backslash k}^{(r)}} \mathsf{P}_{W_k^{(r)}|\overline{W}^{(r-1)}} P_{\overline{W}^{(R)}|W_{[K]}^{(r)}, S_{[K]}^{[r+1:R]}},$$

and $(b)$ is derived by using (Wainwright, 2019) and the fact that $\text{gen}(s_k^{(r)}, \hat{\overline{W}}^{(R)})$ is $1/\sqrt{4n/R}$-subgaussian. More precisely, using (Wainwright, 2019, Theorem 2.6.IV.), we have that for any $\lambda_1 \in [0, 1)$,

$$\mathbb{E} \left[ e^{\frac{\lambda_1 \text{gen}\left(s_k^{(r)}, \hat{\overline{W}}^{(R)}\right)^2}{(R/2n)}} \right] \leqslant \frac{1}{\sqrt{1 - \lambda_1}}.$$

Now, letting $\lambda_1 = \frac{2n/R - 1}{2n/R} = \frac{\lambda}{2n/R}$, we drive that

$$\mathbb{E} \left[ e^{\lambda \text{gen}\left(s_k^{(r)}, \hat{\overline{W}}^{(R)}\right)^2} \right] = \mathbb{E} \left[ e^{\frac{\lambda_1 \text{gen}\left(s_k^{(r)}, \hat{\overline{W}}^{(R)}\right)^2}{(2R/n)}} \right] \leqslant \frac{1}{\sqrt{1 - \lambda_1}} = \sqrt{2n/R}.$$

Combining (45) with (43) completes the proof. $\qquad \square$

### F.1.2 PART II.

*Proof.* Consider any FL-model $P_{\mathbf{W}|\mathbf{S}}$. Consider any set of *priors* $\{\mathsf{P}_{k,r}\}_{k \in [K], r \in [R]}$ such that $\mathsf{P}_{k,r}$ could depend on $\overline{W}^{(r-1)}$, *i.e.*, $\mathsf{P}_{k,r}$ is a conditional prior on $W_K^{(r)}$ given $\overline{W}^{(r-1)}$. Let $\lambda := 2n/R - 1$. Let the RHS of the bound as $\sqrt{\Delta(\mathbf{S}, \mathbf{W})}$, *i.e.*, let

$$\Delta(\mathbf{S}, \mathbf{W}) := \frac{\frac{1}{KR} \sum_{k \in [K], r \in [R]} \log \left( \frac{\mathrm{d} P_{W_k^{(r)}|S_k^{(r)}, \overline{W}^{(r-1)}}}{\mathrm{d} \mathsf{P}_{k,r} r} \right) + \log(\frac{\sqrt{2n}}{\sqrt{R}\delta})}{\lambda}.$$

Further, recall that

$$\text{gen}(\mathbf{S}, \overline{W}^{(R)}) = \frac{1}{KR} \sum_{k \in [K], r \in [R]} \text{gen}(S_k^{(r)}, \overline{W}^{(R)}).$$

Denote

$$(PP)_{k,r} \coloneqq P_{\overline{W}^{(r-1)}, W_{[K]\setminus k}^{(r)} | S_{[K]}^{[r-1]}, S_{[K]\setminus k}^{(r)}} \mathsf{P}_{k,r} P_{\overline{W}^{(R)} | \overline{W}^{(r)}, S_{[K]}^{[r+1:R]}}, \quad k \in [K], r \in [R].$$

For any fixed $\delta > 0$, denote

$$\mathcal{G}_{\mathbf{S}, \mathbf{W}}^{\delta} \coloneqq \{ \nu_{\mathbf{S}, \mathbf{W}} \colon D_{KL}(\nu_{\mathbf{S}, \mathbf{W}} \| P_{\mathbf{S}, \mathbf{W}}) \leqslant \log(1/\delta) \}, \tag{46}$$

as the set of all distributions $\nu_{\mathbf{S}, \mathbf{W}}$ over $\mathcal{Z}^{nK} \times \mathcal{W}^{(n+1)K}$, whose KL-divergence with $P_{\mathbf{S}, \mathbf{W}}$ does not exceed $\log(1/\delta)$. Then,

$$\mathbb{P}\left( \text{gen}(\mathbf{S}, \overline{W}^{(R)}) \geqslant \sqrt{\Delta(\mathbf{S}, \mathbf{W})} \right) \tag{47}$$

$$\overset{(a)}{\leqslant} \mathbb{P}\left( \frac{1}{KR} \sum_{k \in [K], r \in [R]} \text{gen}(S_k^{(r)}, \overline{W}^{(R)})^2 \geqslant \Delta(\mathbf{S}, \mathbf{W}) \right)$$

$$\overset{(b)}{\leqslant} \max\left( \log(\delta), \sup_{\nu_{\mathbf{S}, \mathbf{W}} \in \mathcal{G}_{\mathbf{S}, \mathbf{W}}^{\delta}} \left\{ -D_{KL}(\nu_{\mathbf{S}, \mathbf{W}} \| P_{\mathbf{S}, \mathbf{W}}) - \lambda \mathbb{E}_{\nu_{\mathbf{S}, \mathbf{W}}} \left[ \Delta(\mathbf{S}, \mathbf{W}) - \frac{1}{KR} \sum_{r,k} \text{gen}(S_k^{(r)}, \overline{W}^{(R)})^2 \right] \right\} \right), \tag{48}$$

where $(a)$ holds by Jensen inequality for the convex function $g(x) = x^2$ and $(b)$ is due Lemma 2, proved in Appendix F.9.

**Lemma 2.** *The inequality* (48) *holds.*

Next, note that using Donsker-Varadhan's inequality we have

$$\mathbb{E}_{\nu_{\mathbf{S}, \mathbf{W}}} \left[ \lambda \text{gen}(S_k^{(r)}, \overline{W}^{(R)})^2 \right]$$

$$\leqslant \mathbb{E}_{\nu_{\mathbf{S}}} \left[ D_{KL}\left( \nu_{\mathbf{W}|\mathbf{S}} \| (PP)_{k,r} \right) + \log \mathbb{E}_{(PP)_{k,r}} \left[ e^{\lambda \text{gen}(S_k^{(r)}, \overline{W}^{(R)})^2} \right] \right]$$

$$\leqslant D_{KL}\left( \nu_{\mathbf{S}, \mathbf{W}} \| \nu_{\mathbf{S}} (PP)_{k,r} \right) + D_{KL}(\nu_{\mathbf{S}} \| P_{\mathbf{S}}) + \log \mathbb{E}_{P_{\mathbf{S}}(PP)_{k,r}} \left[ e^{\lambda \text{gen}(S_k^{(r)}, \overline{W}^{(R)})^2} \right]$$

$$\leqslant D_{KL}\left( \nu_{\mathbf{S}, \mathbf{W}} \| \nu_{\mathbf{S}} (PP)_{k,r} \right) + D_{KL}(\nu_{\mathbf{S}} \| P_{\mathbf{S}}) + \log\left( \sqrt{2n/R} \right)$$

$$= D_{KL}\left( \nu_{\mathbf{S}, \mathbf{W}} \| \nu_{\mathbf{S}} P_{\mathbf{W}|\mathbf{S}} \right) + \mathbb{E}_{\nu_{\mathbf{S}, \mathbf{W}}} \left[ \log\left( \frac{\mathrm{d}P_{W_k^{(r)} | S_k^{(r)}, \overline{W}^{(r-1)}}}{\mathrm{d}\mathsf{P}_{k,r} r} \right) \right] + D_{KL}(\nu_{\mathbf{S}} \| P_{\mathbf{S}}) + \log\left( \sqrt{2n/R} \right)$$

$$= D_{KL}\left( \nu_{\mathbf{S}, \mathbf{W}} \| P_{\mathbf{S}, \mathbf{W}} \right) + \mathbb{E}_{\nu_{\mathbf{S}, \mathbf{W}}} \left[ \log\left( \frac{\mathrm{d}P_{W_k^{(r)} | S_k^{(r)}, \overline{W}^{(r-1)}}}{\mathrm{d}\mathsf{P}_{k,r} r} \right) \right] + \log\left( \sqrt{2n/R} \right).$$

Putting everything together conclude that

$$-D_{KL}(\nu_{\mathbf{S}, \mathbf{W}} \| P_{\mathbf{S}, \mathbf{W}}) - \lambda \mathbb{E}_{\nu_{\mathbf{S}, \mathbf{W}}} \left[ \Delta(\mathbf{S}, \mathbf{W}) - \frac{1}{KR} \sum_{r,k} \text{gen}(S_k^{(r)}, \overline{W}^{(R)})^2 \right] \leqslant \log(\delta).$$

This completes the proof.

$$\square$$

## F.2 PROOF OF THEOREM 2

*Proof.* Similar to the proof of (Sefidgaran & Zaidi, 2023, Theorem 6), it is easy to see that for any choice of the prior $\mathsf{P}_{k,r}$, it suffices to only consider an arbitrary $\mathsf{P}_{\mathbf{W}|\mathbf{S}}$. Furthermore, without loss of generality, we can assume $C = 1$, otherwise, one can re-scale the loss function.

Let $\lambda \coloneqq 2n/R - 1$ and the RHS of the bound as $\sqrt{\Delta(\mathbf{S})}$, *i.e.*, let

$$\Delta(\mathbf{S}) \coloneqq \frac{\frac{1}{KR} \sum_{k \in [K], r \in [R]} \mathbb{E}_{\overline{W}^{(r-1)} \sim P_{\overline{W}^{(r-1)} | S_{[K]}^{[r-1]}}} \left[ D_{KL}\left( p_{\hat{W}_k^{(r)} | S_k^{(r)}, \overline{W}^{(r-1)}} \| \mathsf{P}_{k,r} \right) \right] + \log(\frac{\sqrt{2n}}{\sqrt{R}\delta})}{\lambda} + \epsilon,$$

where $p_{\hat{\mathbf{W}}|\mathbf{S}}$ satisfy the $\epsilon$-distortion criterion:

$$\left| \mathbb{E}_{\mathbf{W} \sim P_{\mathbf{W}|\mathbf{S}}} \left[ \mathrm{gen}(\mathbf{S}, \overline{W}^{(R)}) \right] - \frac{1}{KR} \sum_{k \in [K], r \in [R]} \mathbb{E}_{\mathbf{W} \sim (Pp)_{k,r}} \left[ \mathrm{gen}(S_k^{(r)}, \hat{\overline{W}}^{(R)}) \right] \right| \leqslant \epsilon/2, \quad (49)$$

where

$$(Pp)_{k,r} := P_{\overline{W}^{(r-1)}, W_{[K] \setminus k}^{(r)} | S_{[K]}^{[r-1]}, S_{[K] \setminus k}^{(r)}} \, p_{\hat{W}_k^{(r)} | S_k^{(r)}, \overline{W}^{(r-1)}} P_{\hat{\overline{W}}^{(R)} | W_{[K] \setminus k}^{(r)}, \hat{W}_k^{(r)}, S_{[K]}^{[r+1:R]}}.$$

Further, recall that

$$\mathbb{E}_{\mathbf{W} \sim P_{\mathbf{W}|\mathbf{S}}} \left[ \mathrm{gen}(\mathbf{S}, \overline{W}^{(R)}) \right] = \frac{1}{KR} \sum_{k \in [K], r \in [R]} \mathbb{E}_{\mathbf{W} \sim P_{\mathbf{W}|\mathbf{S}}} \left[ \mathrm{gen}(S_k^{(r)}, \overline{W}^{(R)}) \right].$$

First, note that

$$\mathbb{P}_{\mathbf{S}} \left( \mathbb{E}_{\mathbf{W} \sim P_{\mathbf{W}|\mathbf{S}}} \left[ \mathrm{gen}(\mathbf{S}, \overline{W}^{(R)}) \right] > \sqrt{\Delta(\mathbf{S})} \right) = \mathbb{P}_{\mathbf{S}} \left( \mathbb{E}_{\mathbf{W} \sim P_{\mathbf{W}|\mathbf{S}}} \left[ \mathrm{gen}(\mathbf{S}, \overline{W}^{(R)}) \right]^2 > \Delta(\mathbf{S}) \right). \quad (50)$$

Next, we state the following lemma, proved in Appendix F.10.

**Lemma 3.** *For any $\delta > 0$,*

$$\log \mathbb{P}_{\mathbf{S}} \left( \mathbb{E}_{\mathbf{W} \sim P_{\mathbf{W}|\mathbf{S}}} \left[ \mathrm{gen}(\mathbf{S}, \overline{W}^{(R)}) \right]^2 > \Delta(\mathbf{S}) \right)$$

$$\leqslant \max \left( \log(\delta), \sup_{\nu_{\mathbf{S}} \in \mathcal{G}_{\mathbf{S}}^{\delta}} \left\{ -D_{KL}(\nu_{\mathbf{S}} \| P_{\mathbf{S}}) - \lambda \mathbb{E}_{\nu_{\mathbf{S}}} \left[ \Delta(\mathbf{S}) - \epsilon - \tilde{\Delta}(\mathbf{S})^2 \right] \right\} \right), \quad (51)$$

*where $\mathcal{G}_{\mathbf{S}}^{\delta}$ is defined in* (42) *and*

$$\tilde{\Delta}(\mathbf{S}) := \frac{1}{KR} \sum_{r,k} \mathbb{E}_{\mathbf{W} \sim (Pp)_{k,r}} \left[ \mathrm{gen}(s_k^{(r)}, \hat{\overline{W}}^{(R)}) \right].$$

Now, for any $\mathbf{s}$, we have

$$\lambda \tilde{\Delta}(\mathbf{S})^2 \overset{(a)}{\leqslant} \frac{1}{KR} \sum_{r,k} \mathbb{E}_{\mathbf{W} \sim (Pp)_{k,r}} \left[ \lambda \, \mathrm{gen}(s_k^{(r)}, \hat{\overline{W}}^{(R)})^2 \right]$$

$$\overset{(b)}{\leqslant} \frac{1}{KR} \sum_{r,k} \left( D_{KL} \left( (Pp)_{k,r} \| (P\mathsf{P}_{k,r})_{k,r} \right) + \log \mathbb{E}_{\mathbf{W} \sim (P\mathsf{P}_{k,r})_{k,r}} \left[ e^{\lambda \, \mathrm{gen}(s_k^{(r)}, \hat{\overline{W}}^{(R)})^2} \right] \right)$$

$$= \frac{1}{KR} \sum_{r,k} \left( \mathbb{E}_{\overline{W}^{(r-1)} \sim P_{\overline{W}^{(r-1)} | S_{[K]}^{[r-1]}}} \left[ D_{KL} \left( p_{\hat{W}_k^{(r)} | s_k^{(r)}, W^{(r-1)}} \| \mathsf{P}_{\hat{W}_k^{(r)} | \overline{W}^{(r-1)}} \right) \right] \right.$$

$$\left. + \log \mathbb{E}_{\mathbf{W} \sim (P\mathsf{P}_{k,r})_{k,r}} \left[ e^{\lambda \, \mathrm{gen}(s_k^{(r)}, \hat{\overline{W}}^{(R)})^2} \right] \right),$$

where $(a)$ is derived by using Jensen inequality for the function $g(x) = x^2$, $(b)$ using Donsker-Varadhan's inequality, and where

$$(P\mathsf{P}_{k,r})_{k,r} := P_{\overline{W}^{(r-1)}, W_{[K] \setminus k}^{(r)} | S_{[K]}^{[r-1]}, S_{[K] \setminus k}^{(r)}} \, \mathsf{P}_{\hat{W}_k^{(r)} | \overline{W}^{(r-1)}} P_{\hat{\overline{W}}^{(R)} | W_{[K] \setminus k}^{(r)}, \hat{W}_k^{(r)}, S_{[K]}^{[r+1:R]}}.$$

Hence, for any $\nu_{\mathbf{S}}$, we have

$$-D_{KL}(\nu_{\mathbf{S}} \| P_{\mathbf{S}}) - \lambda \mathbb{E}_{\nu_{\mathbf{S}}} \left[ \Delta(\mathbf{s}) - \epsilon - \tilde{\Delta}(\mathbf{S})^2 \right] - \log(\delta)$$

$$\leqslant -\log \left( \frac{\sqrt{2n}}{\sqrt{R}} \right) - D_{KL}(\nu_{\mathbf{S}} \| P_{\mathbf{S}}) + \frac{1}{KR} \sum_{r,k} \mathbb{E}_{\mathbf{S} \sim \nu_{\mathbf{S}}} \log \mathbb{E}_{\mathbf{W} \sim (P\mathsf{P}_{k,r})_{k,r}} \left[ e^{\lambda \, \mathrm{gen}(s_k^{(r)}, \hat{\overline{W}}^{(R)})^2} \right]$$

$$\overset{(a)}{\leqslant} -\log \left( \frac{\sqrt{2n}}{\sqrt{R}} \right) + \frac{1}{KR} \sum_{r,k} \log \mathbb{E}_{\mathbf{S}, \mathbf{W} \sim P_{\mathbf{S}}(P\mathsf{P}_{k,r})_{k,r}} \left[ e^{\lambda \, \mathrm{gen}(s_k^{(r)}, \hat{\overline{W}}^{(R)})^2} \right]$$

$$= -\log \left( \frac{\sqrt{2n}}{\sqrt{R}} \right) + \frac{1}{KR} \sum_{r,k} \log \mathbb{E}_{\mathbf{S} \setminus S_k^{(r)}, \mathbf{W} \sim P_{\mathbf{S} \setminus S_k^{(r)}}(P\mathsf{P}_{k,r})_{k,r}} \mathbb{E}_{S_k^{(r)} \sim \mu_k^{\otimes(n/R)}} \left[ e^{\lambda \, \mathrm{gen}(s_k^{(r)}, \hat{\overline{W}}^{(R)})^2} \right]$$

$$\leqslant 0,$$

where $(a)$ holds due to Donsker-Varadhan's inequality. This completes the proof. $\qquad \square$

## F.3 PROOF OF THEOREM 3

*Proof.* Consider any set of distributions $\left\{p_{\hat{W}_k^{(r)}|S_k^{(r)},\overline{W}^{(r-1)}}\right\}_{k\in[K],r\in[R]}$ that satisfy the distortion criterion (12) for any $k \in [K]$ and $r \in [R]$. Then,

$$\mathbb{E}_{\mathbf{S},\mathbf{W}\sim P_{\mathbf{S},\mathbf{w}}}\left[\text{gen}(\mathbf{S},\overline{W}^{(R)})\right]$$

$$\leqslant \frac{1}{KR}\sum_{k\in[K],r\in[R]}\left(\mathbb{E}_{S_k^{(r)},\overline{W}^{(r-1)},W_{[K]\backslash k}^{(r)},\hat{\overline{W}}^{(R)}\sim P_{S_k^{(r)}}(Pp)_{k,r}}\left[\text{gen}(S_k^{(r)},\hat{\overline{W}}^{(R)})\right]+\epsilon_{k,r}\right)$$

$$\leqslant \frac{1}{KR}\sum_{k\in[K],r\in[R]}\mathbb{E}_{S_k^{(r)},\overline{W}^{(r-1)},W_{[K]\backslash k}^{(r)},\hat{\overline{W}}^{(R)}\sim P_{S_k^{(r)}}(Pp)_{k,r}}\left[\text{gen}(S_k^{(r)},\hat{\overline{W}}^{(R)})\right]+\epsilon, \quad (52)$$

where

$$(Pp)_{k,r} := P_{\overline{W}^{(r-1)},W_{[K]\backslash k}^{(r)}}p_{\hat{W}_k^{(r)}|S_k^{(r)},\overline{W}^{(r-1)}}P_{\hat{\overline{W}}^{(R)}|W_{[K]\backslash k}^{(r)},\hat{W}_k^{(r)}}.$$

Let $q_{\hat{W}_k^{(r)}|\overline{W}^{(r-1)}}$ be the marginal conditional distribution of $\hat{W}_k^{(r)}$ given $\overline{W}^{(r-1)}$ under $P_{S_k^{(r)}}p_{\hat{W}_k^{(r)}|S_k^{(r)},\overline{W}^{(r-1)}}$, and denote similarly

$$(Pq)_{k,r} := P_{\overline{W}^{(r-1)},W_{[K]\backslash k}^{(r)}}q_{\hat{W}_k^{(r)}|\overline{W}^{(r-1)}}P_{\hat{\overline{W}}^{(R)}|W_{[K]\backslash k}^{(r)},\hat{W}_k^{(r)}}.$$

Then,

$$\lambda\mathbb{E}_{S_k^{(r)},\overline{W}^{(r-1)},W_{[K]\backslash k}^{(r)},\hat{\overline{W}}^{(R)}\sim P_{\mathbf{S}}(Pp)_{k,r}}\left[\text{gen}(S_k^{(r)},\hat{\overline{W}}^{(R)})\right]$$

$$\overset{(a)}{\leqslant} D_{KL}\left(P_{S_k^{(r)}}(Pp)_{k,r}\|P_{S_k^{(r)}}(Pq)_{k,r}\right)+\log\mathbb{E}_{P_{S_k^{(r)}}(Pq)_{k,r}}\left[e^{\lambda\,\text{gen}(S_k^{(r)},\hat{\overline{W}}^{(R)})}\right]$$

$$= I(S_k^{(r)};\hat{W}_k^{(r)}|\overline{W}^{(r-1)})+\log\mathbb{E}_{(Pq)_{k,r}}\mathbb{E}_{P_{S_k^{(r)}}}\left[e^{\lambda\,\text{gen}(S_k^{(r)},\hat{\overline{W}}^{(R)})}\right]$$

$$\overset{(b)}{\leqslant} I(S_k^{(r)};\hat{W}_k^{(r)}|\overline{W}^{(r-1)})+\frac{\lambda^2\sigma^2}{2n/R}, \quad (53)$$

where $(a)$ is deduced using Donsker-Varadhan's inequality and $(b)$ using the fact that for any $w \in \mathcal{W}$, $\text{gen}(S_k^{(r)},w)$ is $\sigma/\sqrt{n/R}$-subgaussian.

Combining (52) and (53), and taking the infimum over all admissible choices of the conditional Markov kernels $\left\{p_{\hat{W}_k^{(r)}|S_k^{(r)},\overline{W}^{(r-1)}}\right\}_{k\in[K],r\in[R]}$, we get

$$\mathbb{E}_{\mathbf{S},\mathbf{W}\sim P_{\mathbf{S},\mathbf{w}}}\left[\text{gen}(\mathbf{S},\overline{W}^{(R)})\right]$$

$$\leqslant\frac{1}{KR}\sum_{k\in[K],r\in[R]}\inf_{p_{\hat{W}_k^{(r)}|S_k^{(r)},\overline{W}^{(r-1)}}}\left\{\mathbb{E}_{S_k^{(r)},\overline{W}^{(r-1)},W_{[K]\backslash k}^{(r)},\hat{\overline{W}}^{(R)}\sim P_{S_k^{(r)}}(Pp)_{k,r}}\left[\text{gen}(S_k^{(r)},\hat{\overline{W}}^{(R)})\right]\right\}+\epsilon$$

$$\leqslant\frac{1}{KR}\sum_{k\in[K],r\in[R]}\mathfrak{RD}(P_{\mathbf{S},\mathbf{W}},k,r,\epsilon_{k,r})/\lambda+\frac{\lambda\sigma^2}{2n/R}+\epsilon$$

$$\overset{(a)}{\leqslant}\sqrt{\frac{2\sigma^2\sum_{k\in[K],r\in[R]}\mathfrak{RD}(P_{\mathbf{S},\mathbf{W}},k,r,\epsilon_{k,r})}{nK}}+\epsilon, \quad (54)$$

where the last step is established by letting

$$\lambda := \sqrt{\frac{2n/R}{\sigma^2 KR}\sum_{k\in[K],r\in[R]}\mathfrak{RD}(P_{\mathbf{S},\mathbf{W}},k,r,\epsilon_{k,r})}.$$

This completes the proof. $\qquad\square$

## F.4 PROOF OF THEOREM 4

*Proof.* We use an extension of Theorem 3, that is Proposition 1, to prove this theorem. Let $\epsilon_{k,r} = \epsilon$ for all $k \in [K]$ and $r \in [R]$. Fix a $k \in [K]$ and $r \in [R]$. We upper bound the rate-distortion term $\mathfrak{RD}^\star(P_{\mathbf{S},\mathbf{W}}, k, r, \epsilon_{k,r})$, defined in (34). To this end, let $V_k^{(r)}$ denote all the randomness in round $r$ for client $k$, *i.e.,* given $s_k^{(r)}$, all mini-batches $\{\mathcal{B}_{k,r,t}\}_t$ will be fixed. We define a proper

$$p_{\hat{\overline{W}}_k^{(R)}|S_k^{(r)},\mathfrak{V}_{k,r}}, \tag{55}$$

that satisfy (33), where

$$\mathfrak{V}_{k,r} = \left(V_k^{(r)}, \overline{W}^{(r-1)}, W_{[K]\backslash k}^{(r)}, S_{[K]}^{[r+1:R]}, V_{[K]}^{[r+1:R]}\right). \tag{56}$$

For simplicity, let

$$\mathfrak{U}_{k,r} = \left(W_{[K]\backslash k}^{(r)}, S_{[K]}^{[r+1:R]}, V_{[K]}^{[r+1:R]}\right). \tag{57}$$

To define $p_{\hat{\overline{W}}_k^{(R)}|S_k^{(r)},\mathfrak{V}_{k,r}}$, first we define $p_{\hat{\overline{W}}_k^{(R)}|W_k^{(r)},\mathfrak{U}_{k,r}}$. Then, we let

$$p_{\hat{\overline{W}}_k^{(R)}|W_k^{(r)},\mathfrak{V}_{k,r},S_k^{(r)}} = p_{\hat{\overline{W}}_k^{(R)}|W_k^{(r)},\mathfrak{U}_{k,r}}. \tag{58}$$

Now, we have

$$p_{\hat{\overline{W}}_k^{(R)}|S_k^{(r)},\mathfrak{V}_{k,r}} = \mathbb{E}_{W_k^{(r)}\sim P_{W_k^{(r)}|S_k^{(r)},V_k^{(r)},\overline{W}^{(r-1)}}}\left[p_{\hat{\overline{W}}_k^{(R)}|W_k^{(r)},\mathfrak{U}_{k,r}}\right]. \tag{59}$$

Note that $P_{W_k^{(r)}|S_k^{(r)},\mathfrak{V}_{k,r}} = P_{W_k^{(r)}|S_k^{(r)},V_k^{(r)},\overline{W}^{(r-1)}}$. We proceed to define $p_{\hat{\overline{W}}_k^{(R)}|W_k^{(r)},\mathfrak{U}_{k,r}}$. Consider an integer $m \in \mathbb{N}^*$. Fix an $\mathfrak{U}_{k,r}$, *i.e.,* fix $w_{[K]\backslash k}^{(r)}$, $(s_{[K]}^{[r+1:R]}, v_{[K]}^{[r+1:R]})$, and consequently all mini-batches

$$\left\{\mathcal{B}_{j,r',t}\right\}_{j\in[K],r'\in[r+1:R],t\in[\tau]}.$$

Assume the aggregated model at round $(r)$ to be

$$\overline{w}^{(r),\star} = \frac{1}{K}\sum_{j\neq k}w_j^{(r)}. \tag{60}$$

Then, with this aggregated model at step $(r)$, and the considered fixed $\mathfrak{U}_{k,r}$, if $r < R$, consider all the induced $\mathsf{D}^\star_{(\overline{w}^{(r'-1)},\{\mathcal{B}_{j,r',t}\}_t)}$, for $j \in [K]$, $r' \in [r+1:R]$. Denote

$$\mathsf{D}^\star_{r'} := \frac{1}{K}\sum_{k\in[K]}\mathsf{D}^\star_{(\overline{w}^{(r'-1)},\{\mathcal{B}_{k,r',t}\}_t)}, \quad r' \in [r+1:R],$$

$$\mathsf{D}^\star := \prod_{r'\in[r+1:R]}\mathsf{D}^\star_{r'}. \tag{61}$$

Denote also the resulting global model as $\overline{w}^{(R),\star}$. If $r = R$, let $\mathsf{D}^\star = \mathsf{I}_d$, where $\mathsf{I}_d$ is the identity matrix.

Consider the random matrix A, whose elements are distributed in an i.i.d. manner according to $\mathcal{N}(0, 1/m)$. The matrix is used for Johnson-Lindenstrauss transformation (Johnson & Lindenstrauss, 1984). Considering such matrix for dimension reduction in SVM for centralized and one-round distributed learning was previously considered in (Grønlund et al., 2020; Sefidgaran et al., 2022a).

Fix some $c_1, c_2, \nu > 0$. Let

$$\epsilon := 8e^{-\frac{m}{7}\left(\frac{K\theta}{6Bq_{e,b}^{R-r}}\right)^2} + 2e^{-0.21m(c_1^2-1)} + 2e^{-0.21m(c_2^2q_{e,b}^{2(r-R)}-1)} + \frac{4m\nu^m}{\sqrt{\pi}}e^{-\frac{(m+1)}{2}\left(\frac{K\theta}{6c_1\nu B}\right)^2}. \tag{62}$$

Now, for a given $w_k^{(r)}$, if $\|\mathsf{AD}^\star w_k^{(r)}\| \leq c_2$, then let $M$ be chosen uniformly over $\mathcal{B}_m(\mathsf{AD}^\star w_k^{(r)}, \nu)$; otherwise let $M$ be chosen uniformly over $\mathcal{B}_m(0, \nu)$. Define

$$\hat{\overline{W}}^{(R)} := \mathsf{A}\overline{W}^{(R),\star} + \frac{1}{K}M. \tag{63}$$

Note that we defined $\hat{\overline{W}}^{(R)} \in \mathbb{R}^m$, while $\overline{W}^{(R)} \in \mathbb{R}^d$. By this definition, we have

$$
\begin{aligned}
I(S_k^{(r)}; \hat{\overline{W}}_k^{(R)}|\mathfrak{V}_{k,r}) &\overset{(a)}{\leqslant} I(W_k^{(r)}; \hat{\overline{W}}_k^{(R)}|\mathfrak{V}_{k,r}) \\
&\overset{(b)}{=} I(W_k^{(r)}; \hat{\overline{W}}_k^{(R)}|\mathfrak{U}_{k,r}) \\
&\overset{(c)}{=} I(W_k^{(r)}; M|\mathfrak{U}_{k,r}) \\
&= h(M|\mathfrak{U}_{k,r}) - h(M|W_k^{(r)}, \mathfrak{U}_{k,r}) \\
&= h(M|\mathfrak{U}_{k,r}) - \log(\mathrm{Vol}_m(\nu)) \\
&\overset{(d)}{\leqslant} \log(\mathrm{Vol}_m(c_2 + \nu)) - \log(\mathrm{Vol}_m(\nu)) \\
&= m\log((c_2 + \nu)/\nu),
\end{aligned}
\tag{64}
$$

where $\mathrm{Vol}_m(r)$ denote the volume of $m$-dimensional ball with radius $r$,

- $(a)$ follows by data-processing inequality,

- $(b)$ and $(c)$ due to the construction of $\hat{\overline{W}}_k^{(R)}$ and since given $\mathfrak{U}_{k,r}$, $\mathsf{D}^\star$ is a fixed matrix,

- and $(d)$ since $M \in \mathbb{R}^m$ is bounded always in the ball of radius $c_2 + \nu$.

Now, we investigate the distortion criterion (33). For this, first, we define the loss function $\hat{\ell}$. For fixed $\mathfrak{u}$ and $\mathsf{A}$, let the 0-1 loss function $\hat{\ell}$ be,

$$
\hat{\ell}_{\mathfrak{u},\mathsf{A},\theta}\left(z, \hat{\overline{w}}^{(R)}\right) := \mathbb{1}_{\{y \langle\!\langle x, \hat{\overline{w}}^{(R)}\rangle\!\rangle_{\mathfrak{u},\mathsf{A}} > \theta/2\}},
\tag{65}
$$

where we define

$$
\langle\!\langle x, \hat{\overline{w}}^{(R)}\rangle\!\rangle_{\mathfrak{u},\mathsf{A}} := \langle x, \overline{w}^{(R),\star}\rangle - \langle \mathsf{A}x, \mathsf{A}\overline{w}^{(R),\star}\rangle + \langle \mathsf{A}x, \hat{\overline{w}}^{(R)}\rangle.
\tag{66}
$$

Note that $\overline{w}^{(R),\star}$ and $\mathsf{D}^\star$ are deterministically determined by $\mathfrak{u}$.

It is easy to verify that

$$
\begin{aligned}
&\mathbb{E}_{\mathbf{S},\mathbf{V},\mathbf{W},\hat{\overline{W}}^{(R)}}\left[\mathrm{gen}_\theta(S_k^{(r)}, \overline{W}^{(R)}) - \mathrm{gen}(S_k^{(r)}, \hat{\overline{W}}^{(R)})\right] \\
&= \mathbb{E}_{P_{S_k^{(r)},W_k^{(r)},\mathfrak{u}_{k,r},\overline{W}^{(R)}}} \mathbb{E}_{P_{\hat{\overline{W}}_k^{(R)}|W_k^{(r)},\mathfrak{u}_{k,r}}}\left[\mathrm{gen}(S_k^{(r)}, \overline{W}^{(R)}) - \mathrm{gen}(S_k^{(r)}, \hat{\overline{W}}^{(R)})\right] \\
&= \mathbb{E}_{P_{W_k^{(r)},\mathfrak{u}_{k,r},\overline{W}^{(R)}}} \mathbb{E}_{P_{\hat{\overline{W}}_k^{(R)}|W_k^{(r)},\mathfrak{u}_{k,r}}} \mathbb{E}_{Z_k \sim \mu_k}\left[\ell_0(Z_k, \overline{W}^{(R)}) - \hat{\ell}_{\mathfrak{U},\mathsf{A},\theta}(Z_k, \hat{\overline{W}}_k^{(R)})\right] \\
&\quad - \mathbb{E}_{P_{S_k^{(r)},W_k^{(r)},\mathfrak{u}_{k,r},\overline{W}^{(R)}}} \mathbb{E}_{P_{\hat{\overline{W}}_k^{(R)}|W_k^{(r)},\mathfrak{u}_{k,r}}}\left[\frac{1}{n_R}\sum_{i=1}^{n_R}\ell_\theta(Z_{k,r,i}, \overline{W}_k^{(R)}) - \hat{\ell}_{\mathfrak{U},\mathsf{A},\theta}(Z_{k,r,i}, \hat{\overline{W}}_k^{(R)})\right] \\
&\leqslant \mathbb{E}_{P_{W_k^{(r)},\mathfrak{u}_{k,r},\overline{W}^{(R)}}} \mathbb{E}_{P_{\hat{\overline{W}}_k^{(R)}|W_k^{(r)},\mathfrak{u}_{k,r}}} \mathbb{E}_{Z_k \sim \mu_k}\left[\mathbb{1}_{\left\{\left|\langle X_k, \overline{W}^{(R)}\rangle - \langle\!\langle X_k, \hat{\overline{W}}^{(R)}\rangle\!\rangle_{\mathfrak{u},\mathsf{A}}\right| > \theta/2\right\}}\right] \\
&\quad + \mathbb{E}_{P_{S_k^{(r)},W_k^{(r)},\mathfrak{u}_{k,r},\overline{W}^{(R)}}} \mathbb{E}_{P_{\hat{\overline{W}}_k^{(R)}|W_k^{(r)},\mathfrak{u}_{k,r}}}\left[\frac{1}{n_R}\sum_{i=1}^{n_R}\mathbb{1}_{\left\{\left|\langle X_{k,r,i}, \overline{W}^{(R)}\rangle - \langle\!\langle X_{k,r,i}, \hat{\overline{W}}^{(R)}\rangle\!\rangle_{\mathfrak{u},\mathsf{A}}\right| > \theta/2\right\}}\right] \\
&:= \mathcal{E}_\mathsf{A}.
\end{aligned}
\tag{67}
$$

In the rest of the proof we show that

$$
\begin{aligned}
\mathbb{E}_\mathsf{A}[\mathcal{E}_\mathsf{A}] &\leqslant 8e^{-\frac{m}{7}\left(\frac{K\theta}{6Bq_{e,b}^{R-r}}\right)^2} + 2e^{-0.21m(c_1^2-1)} + 2e^{-0.21m(c_2^2q_{e,b}^{2(r-R)}-1)} + \frac{4m\nu^m}{\sqrt{\pi}}e^{-\frac{(m+1)}{2}\left(\frac{K\theta}{6c_1\nu B}\right)^2} \\
&= \epsilon,
\end{aligned}
\tag{68}
$$

where $\epsilon$ is defined in (62). Then, this shows that there exists a least one A for which $\mathcal{E}_\mathsf{A} \leqslant \epsilon$. Combining this with (64), yields

$$\mathfrak{RD}^\star(P_{\mathbf{S},\mathbf{W}}, k, r, \epsilon) \leqslant m \log((c_2 + \nu)/\nu).$$

Letting

$$m := \lceil 252 \left( \frac{Bt}{K\theta} \right)^2 \log(nK\sqrt{K}) \rceil,$$

$$c_1 := \sqrt{\frac{K^2\theta^2}{50B^2t^2} - 1},$$

$$c_2 := q_{e,b}^{R-r} \sqrt{\frac{K^2\theta^2}{50B^2t^2} - 1},$$

$$\nu := t/(2c_1), \tag{69}$$

where $t \geqslant q_{e,n}^{R-r}$, and using Theorem 3 completes the proof.

Hence, it remains to show (68). A sufficient condition to show that is to prove for every $x \in \mathbb{R}^d$, and every $w_k^{(r)}$ and $\mathfrak{u}_{k,r}$, we have

$$\mathbb{E}_{P_{\widehat{W}_k^{(R)} | w_k^{(r)}, \mathfrak{u}_{k,r}}} \mathbb{E}_\mathsf{A} \left[ \mathbb{1}_{\left\{ \left| \langle x, \overline{w}^{(R)} \rangle - \langle\!\langle x, \widehat{\overline{W}}^{(R)} \rangle\!\rangle_{\mathfrak{u},\mathsf{A}} \right| > \theta/2 \right\}} \right] \leqslant \epsilon/2. \tag{70}$$

Note that $\overline{w}^{(R)}$ is a deterministic function of $w_k^{(r)}$ and $\mathfrak{u}_{k,r}$. Next, we decompose the difference of the inner products into three terms:

$$\left| \langle x, \overline{w}^{(R)} \rangle - \langle\!\langle x, \widehat{\overline{W}}^{(R)} \rangle\!\rangle_{\mathfrak{u},\mathsf{A}} \right|$$

$$= \left| \langle x, \Delta \rangle + \frac{1}{K} \langle x, \mathsf{D}^\star w_k^{(r)} \rangle - \frac{1}{K} \langle \mathsf{A}x, \mathsf{A}\mathsf{D}^\star w_k^{(r)} \rangle + \frac{1}{K} \langle \mathsf{A}x, \mathsf{A}\mathsf{D}^\star w_k^{(r)} - M \rangle \right|$$

$$\leqslant |\langle x, \Delta \rangle| + \frac{1}{K} \left| \langle x, \mathsf{D}^\star w_k^{(r)} \rangle - \langle \mathsf{A}x, \mathsf{A}\mathsf{D}^\star w_k^{(r)} \rangle \right| + \frac{1}{K} \left| \langle \mathsf{A}x, \mathsf{A}\mathsf{D}^\star w_k^{(r)} - M \rangle \right|, \tag{71}$$

where

$$\Delta := \overline{w}^{(R)} - \left( \overline{w}^{(R),\star} + \frac{1}{K} \mathsf{D}^\star w_k^{(r)} \right). \tag{72}$$

Note that $\Delta$ is deterministic given $w_k^{(r)}$ and $\mathfrak{u}_{k,r}$.

Hence,

$$\mathbb{E}_{P_{\widehat{W}_k^{(R)} | w_k^{(r)}, \mathfrak{u}_{k,r}}} \mathbb{E}_\mathsf{A} \left[ \mathbb{1}_{\left\{ \left| \langle x, \overline{w}^{(R)} \rangle - \langle\!\langle x, \widehat{\overline{W}}^{(R)} \rangle\!\rangle_{\mathfrak{u},\mathsf{A}} \right| > \theta/2 \right\}} \right]$$

$$\leqslant \mathbb{1}_{\{|\langle x, \Delta \rangle| > \theta/6\}} \tag{73}$$

$$+ \mathbb{E}_\mathsf{A} \left[ \mathbb{1}_{\left\{ \frac{1}{K} \left| \langle x, \mathsf{D}^\star w_k^{(r)} \rangle - \langle \mathsf{A}x, \mathsf{A}\mathsf{D}^\star w_k^{(r)} \rangle \right| > \theta/6 \right\}} \right] \tag{74}$$

$$+ \mathbb{E}_{P_{M | w_k^{(r)}}} \mathbb{E}_\mathsf{A} \left[ \mathbb{1}_{\left\{ \frac{1}{K} \left| \langle \mathsf{A}x, \mathsf{A}\mathsf{D}^\star w_k^{(r)} - M \rangle \right| > \theta/6 \right\}} \right]. \tag{75}$$

Now, we proceed to bound the probability that each of the three terms in the RHS of the above inequality.

**Bounding (73).** We show that with probability one, $|\langle x, \Delta \rangle| < \theta/6$. Note that for $r = R$, we have $\mathsf{D}^\star = \mathsf{I}_d$ and this term is zero. Now, for $R > 1$, a sufficient condition to prove $|\langle x, \Delta \rangle| < \theta/6$ is to show that for any $r < R$, we have $\|\Delta\| \leqslant \theta/(6B)$, when (24) holds. We show this if any of two conditions in Assumption 3 hold.

For $r' \in [r + 1 : R]$, define

$$\xi_{r'} := \overline{w}^{(r')} - \left( \overline{w}^{(r'),\star} + \mathsf{D}_{r'}^\star \left( \overline{w}^{(r'-1)} - \overline{w}^{(r'-1),\star} \right) \right). \tag{76}$$

1. **If Condition 1 holds:** Assume (24) holds. Fix some $\beta \in (0, 1]$. Then, first using induction we simultaneously show that for $r' = r + 1, \ldots, R$,

$$\|\xi_{r'}\| \leqslant \frac{\alpha((1+\beta)q_{e,b}^2)^{r'-r-1}}{K^2}, \tag{77}$$

$$\|\overline{w}^{(r'-1)} - \overline{w}^{(r'-1),\star}\|^2 \leqslant \frac{((1+\beta)q_{e,b}^2)^{r'-r-1}}{K^2}. \tag{78}$$

For $r' = r + 1$, due to Assumption 2, we have

$$\|\xi_{r+1}\| \leqslant \alpha\|\overline{w}^{(r)} - \overline{w}^{(r),\star}\|^2/K^2 = \alpha\|w_k^{(r)}\|^2/K^2 \leqslant \alpha/K^2. \tag{79}$$

This proves the induction base for both (77) and (78).

Now, assume that (77) and (78) hold for $r' = r + 1, \ldots, j, j < R$. We show that they hold, for $j + 1$ as well. First, we have

$$\begin{aligned}
\|\overline{w}^{(j)} - \overline{w}^{(j),\star}\|^2 &= \left\|\xi_j + \mathsf{D}_j^\star\left(\overline{w}^{(j-1)} - \overline{w}^{(j-1),\star}\right)\right\|^2 \\
&\stackrel{(a)}{\leqslant} \|\xi_j\|^2 + q_{e,b}^2\left\|\overline{w}^{(j-1)} - \overline{w}^{(j-1),\star}\right\|^2 \\
&\stackrel{(b)}{\leqslant} \frac{\alpha^2((1+\beta)q_{e,b}^2)^{2(j-r-1)}}{K^4} + q_{e,b}^2\frac{((1+\beta)q_{e,b}^2)^{j-r-1}}{K^2} \\
&\stackrel{(c)}{\leqslant} \frac{((1+\beta)q_{e,b}^2)^{j-r}}{K^2},
\end{aligned} \tag{80}$$

where $(a)$ holds due to the triangle inequality and since spectral norm of $\mathsf{D}_j^\star$ is bounded by $q_{e,b}$, $(b)$ using the assumption of the induction, and $(c)$ holds when

$$K^2 > \frac{\alpha^2((1+\beta)q_{e,b}^2)^{j-r-1}}{\beta q_{e,b}^2},$$

which holds by (24). Hence, (78) holds for $r' = j + 1$ as well. Now, we show (77) also holds for $r' = j + 1$.

$$\begin{aligned}
\|\xi_{j+1}\| &:= \left\|\overline{w}^{(j+1)} - \left(\overline{w}^{(j+1),\star} + \mathsf{D}_{j+1}^\star\left(\overline{w}^{(j)} - \overline{w}^{(j),\star}\right)\right)\right\| \\
&\stackrel{(a)}{\leqslant} \alpha\left\|\overline{w}^{(j)} - \overline{w}^{(j),\star}\right\|^2 \\
&\stackrel{(b)}{\leqslant} \frac{\alpha((1+\beta)q_{e,b}^2)^{j-r}}{K^2},
\end{aligned} \tag{81}$$

where $(a)$ holds due to Assumption 2 and $(b)$ by (80). This completes the proof of the induction.

Now, for any $r' \in [r + 1 : R - 1]$, denote

$$\overline{\mathsf{D}}_{r'}^\star := \prod_{j=r'+1}^{R} \mathsf{D}_j^\star, \tag{82}$$

and $\overline{\mathsf{D}}_R^\star = \mathsf{I}_d$. Then,

$$\begin{aligned}
\left\|\overline{w}^{(R)} - \left(\overline{w}^{(R),\star} + \frac{1}{K}\mathsf{D}^\star w_k^{(r)}\right)\right\| &= \left\|\sum_{r'=r+1}^{R} \overline{\mathsf{D}}_{r'}^\star \xi_{r'}\right\| \\
&\leqslant \sum_{r'=r+1}^{R} q_{e,b}^{R-r'}\|\xi_{r'}\| \\
&\leqslant \sum_{r'=r+1}^{R} q_{e,b}^{R-r'} \frac{\alpha((1+\beta)q_{e,b}^2)^{r'-r-1}}{K^2} \\
&= \frac{\alpha q_{e,b}^{R-r-1}}{K^2} \sum_{r'=r+1}^{R} ((1+\beta)q_{e,b})^{r'-r-1} \\
&= \frac{\alpha q_{e,b}^{R-r-1}\left(1 - ((1+\beta)q_{e,b})^{R-r}\right)}{K^2\left(1 - (1+\beta)q_{e,b}\right)}.
\end{aligned} \tag{83}$$

This term is always less than $\theta/(6B)$, if

$$\frac{6\alpha B q_{e,b}^{R-r-1}\left(1 - ((1+\beta)q_{e,b})^{R-r}\right)}{\theta\left(1 - (1+\beta)q_{e,b}\right)} \leqslant K^2,$$

which holds by (24). This completes the proof of this step.

2. **If Condition 2 holds:** Note that similar to the previous case, it can be shown that for $r' = r+1, \ldots, R$

$$\|\xi_{r'}\| \leqslant \alpha\|\overline{w}^{(r'-1)} - \overline{w}^{(r'-1),\star}\|^2, \tag{84}$$

$$\|\overline{w}^{(r')} - \overline{w}^{(r'),\star}\|^2 \leqslant \|\xi_{r'}\|^2 + q_{e,b}^2\|\overline{w}^{(r'-1)} - \overline{w}^{(r'-1),\star}\|^2. \tag{85}$$

Define the sequence $B_j \in \mathbb{R}^+$, $j = 0, \ldots, R-r-1$, recursively as

$$B_j = \alpha B_{j-1}^2 + q_{e,b}^2 B_{j-1},$$

where $B_0 = \frac{1}{K^2}$. Inequalities (84) and (85) conclude that

$$\|\xi_{r'}\| \leqslant \alpha B_{r'-1}. \tag{86}$$

We claim under Condition 2 in Assumption 3, $B_j \leqslant \frac{1-q_{e,b}^2}{\alpha} - \nu$. Once this claim is shown, then we have

$$\begin{aligned} B_j &= \alpha B_{j-1}^2 + q_{e,b}^2 B_{j-1} \\ &\leqslant \alpha\left(\frac{1-q_{e,b}^2}{\alpha} - \nu\right)B_{j-1} + q_{e,b}^2 B_{j-1} \\ &= (1 - \nu\alpha)B_{j-1}. \end{aligned}$$

Hence $B_j \leqslant \frac{(1-\alpha\nu)^j}{K^2}$, and $\|\xi_{r'}\| \leqslant \frac{\alpha(1-\alpha\nu)^{j-1}}{K^2}$. Finally, similar to (83),

$$\begin{aligned} \left\|\overline{w}^{(R)} - \left(\overline{w}^{(R),\star} + \frac{1}{K}\mathsf{D}^\star w_k^{(r)}\right)\right\| &= \left\|\sum_{r'=r+1}^{R} \overline{\mathsf{D}}_{r'}^\star \xi_{r'}\right\| \\ &\leqslant \sum_{r'=r+1}^{R} q_{e,b}^{R-r'}\|\xi_{r'}\| \\ &\leqslant \sum_{r'=r+1}^{R} q_{e,b}^{R-r'}\frac{\alpha(1-\alpha\nu)^{r'-r-1}}{K^2} \\ &= \frac{\alpha q_{e,b}^{R-r-1}}{K^2}\sum_{r'=r+1}^{R}\left((1-\alpha\nu)/q_{e,b}\right)^{r'-r-1} \\ &= \frac{\alpha\left(q_{e,b}^{R-r} - (1-\alpha\nu)^{R-r}\right)}{K^2\left(q_{e,b} - (1-\alpha\nu)\right)}. \end{aligned} \tag{87}$$

This term is less than $\theta/(6B)$ due to (25), which concludes this step. Hence, it remains to show the claim that $B_j \leqslant \frac{1-q_{e,b}^2}{\alpha} - \nu$. To show this, first note that

$$B_0 = \frac{1}{K^2} \overset{(a)}{\leqslant} \frac{1-q_{e,b}^2}{\alpha} - \nu \leqslant \frac{1-q_{e,b}^2}{\alpha},$$

where $(a)$ holds by Assumption 3. Now, recursively, if $B_{j-1} \leqslant \frac{1-q_{e,b}^2}{\alpha}$, then

$$B_j = \alpha B_{j-1}^2 + q_{e,b}^2 B_{j-1} \leqslant B_{j-1}.$$

Hence the sequence $\{B_j\}$ is non-increasing and

$$B_j \leqslant B_0 = \frac{1}{K^2} \leqslant \frac{1-q_{e,b}^2}{\alpha} - \nu.$$

This completes the proof of this part.

**Bounding** (74). Note that $\|\mathsf{D}^\star w_k^{(r)}\| \leqslant q_{e,b}^{R-r}$, due to the fact that spectral norm of all $\mathsf{D}_{r'}^*$ are bounded by $q_{e,b}$ due to Assumption 2. Then, for every $w_k^{(r)}$,

$$\mathbb{E}_{\mathsf{A}}\left[\mathbb{1}_{\left\{\frac{1}{K}\left|\langle x, \mathsf{D}^\star w_k^{(r)}\rangle - \langle \mathsf{A}x, \mathsf{AD}^\star w_k^{(r)}\rangle\right| > \theta/6\right\}}\right] = \mathbb{P}_{\mathsf{A}}\left(\frac{1}{K}\left|\langle x, \mathsf{D}^\star w_k^{(r)}\rangle - \langle \mathsf{A}x, \mathsf{AD}^\star w_k^{(r)}\rangle\right| > \theta/6\right)$$

$$\leqslant 4e^{-\frac{m}{7}\left(\frac{K\theta}{6Bq_{e,b}^{R-r}}\right)^2}, \tag{88}$$

where the last step is due to (Grønlund et al., 2020, Lemma 8, part 2.).

**Bounding** (75). Fix some $c_1, c_2 > 0$. Now, to bound this, we have

$$\mathbb{E}_{p_{M|w_k^{(r)}}}\mathbb{E}_{\mathsf{A}}\left[\mathbb{1}_{\left\{\frac{1}{K}\left|\langle \mathsf{A}x, \mathsf{AD}^\star w_k^{(r)} - M\rangle\right| > \theta/6\right\}}\right]$$

$$\leqslant \mathbb{P}_{\mathsf{A}}(\|\mathsf{A}x\| > c_1 B) + \mathbb{P}_{\mathsf{A}}\left(\|\mathsf{AD}^\star w_k^{(r)}\| > c_2\right)$$

$$+ \mathbb{E}_{p_{M|w_k^{(r)}}}\mathbb{E}_{\mathsf{A}}\left[\mathbb{1}_{\left\{\frac{1}{K}\left|\langle \mathsf{A}x, \mathsf{AD}^\star w_k^{(r)} - M\rangle\right| > \theta/6\right\}}\middle| \|\mathsf{A}x\| \leqslant c_1 B, \|\mathsf{AD}^\star w_k^{(r)}\| \leqslant c_2\right]$$

$$= \mathbb{P}_{\mathsf{A}}(\|\mathsf{A}x\| > c_1 B) + \mathbb{P}_{\mathsf{A}}\left(\|\mathsf{AD}^\star w_k^{(r)}\| > c_2\right)$$

$$+ \mathbb{E}_{M\sim\text{Unif}\left(\mathcal{B}_m(\mathsf{AD}^\star w_k^{(r)}, \nu)\right)}\mathbb{E}_{\mathsf{A}}\left[\mathbb{1}_{\left\{\frac{1}{K}\left|\langle \mathsf{A}x, \mathsf{AD}^\star w_k^{(r)} - M\rangle\right| > \theta/6\right\}}\middle| \|\mathsf{A}x\| \leqslant c_1 B, \|\mathsf{AD}^\star w_k^{(r)}\| \leqslant c_2\right]$$

$$= \mathbb{P}_{\mathsf{A}}(\|\mathsf{A}x\| > c_1 B) + \mathbb{P}_{\mathsf{A}}\left(\|\mathsf{AD}^\star w_k^{(r)}\| > c_2\right) + \mathbb{E}_{U\sim\text{Unif}(\mathcal{B}_m(\nu))}\mathbb{E}_{\mathsf{A}}\left[\mathbb{1}_{\left\{\frac{1}{K}|\langle \mathsf{A}x, U\rangle| > \theta/6\right\}}\middle| \|\mathsf{A}x\| \leqslant c_1 B\right]$$

$$\leqslant e^{-0.21m(c_1^2-1)} + e^{-0.21m(c_2^2 q_{e,b}^{2(r-R)}-1)} + \frac{2m\nu^m}{\sqrt{\pi}}e^{-\frac{(m+1)}{2}\left(\frac{K\theta}{6c_1\nu B}\right)^2}, \tag{89}$$

where the last step follows from (Grønlund et al., 2020, Lemma 8, part 1.) and (Sefidgaran et al., 2022a, Proof of Lemma 3), and since $\|\mathsf{D}^\star w_k^{(r)}\| \leqslant q_{e,b}^{R-r}$.

This completes the proof. $\qquad\square$

## F.5 PROOF OF THEOREM 5

*Proof.* We have

$$\mathbb{E}_{\mathbf{S}, \mathbf{W}\sim P_{\mathbf{S}, \mathbf{W}}}\left[\text{gen}(\mathbf{S}, \overline{W}^{(R)})\right]$$

$$= \frac{1}{KR}\sum_{k\in[K], r\in[R]}\mathbb{E}_{S_k^{(r)}, \overline{W}^{(r-1)}, W_k^{[r:R]}, \overline{W}^{(R)}\sim P_{S_k^{(r)}}P_{k,r}}\left[\text{gen}(S_k^{(r)}, \overline{W}^{(R)})\right], \tag{90}$$

where

$$P_{k,r} := P_{\overline{W}^{(r-1)}, W_k^{[r:R]}|S_k^{(r)}}P_{\overline{W}^{(R)}|\overline{W}^{(r-1)}, W_k^{[r:R]}}.$$

Let $q_{\overline{W}^{(r-1)}, W_k^{[r:R]}}$ be the marginal distribution of $(\overline{W}^{(r-1)}, W_k^{[r:R]})$ under $P_{S_k^{(r)}}P_{\overline{W}^{(r-1)}, W_k^{[r:R]}|S_k^{(r)}}$, and denote similarly

$$(Pq)_{k,r} := q_{\overline{W}^{(r-1)}, W_k^{[r:R]}}P_{\overline{W}^{(R)}|\overline{W}^{(r-1)}, W_k^{[r:R]}}.$$

Then for any $\lambda_{k,r} > 0$,

$$\lambda_{k,r}\mathbb{E}_{S_k^{(r)}, \overline{W}^{(r-1)}, W_k^{[r:R]}, \overline{W}^{(R)}\sim P_{S_k^{(r)}}P_{k,r}}\left[\text{gen}(S_k^{(r)}, \overline{W}^{(R)})\right]$$

$$\stackrel{(a)}{\leqslant} D_{KL}\left(P_{S_k^{(r)}}P_{k,r}\|P_{S_k^{(r)}}(Pq)_{k,r}\right) + \log\mathbb{E}_{P_{S_k^{(r)}}(Pq)_{k,r}}\left[e^{\lambda_{k,r}\text{gen}(S_k^{(r)}, \overline{W}^{(R)})}\right]$$

$$= I(S_k^{(r)}; \overline{W}^{(r-1)}, W_k^{[r:R]}) + \log\mathbb{E}_{(Pq)_{k,r}}\mathbb{E}_{P_{S_k^{(r)}}}\left[e^{\lambda_{k,r}\text{gen}(S_k^{(r)}, \overline{W}^{(R)})}\right]$$

$$\stackrel{(b)}{\leqslant} I(S_k^{(r)}; \overline{W}^{(r-1)}, W_k^{[r:R]}) + \frac{\lambda_{k,r}^2\sigma^2}{2n_{k,r}}, \tag{91}$$

where $(a)$ is deduced using Donsker-Varadhan's inequality and $(b)$ using the fact that for any $w \in \mathcal{W}$, $\mathrm{gen}(S_k^{(r)}, w)$ is $\sigma/\sqrt{n_{k,r}}$-subgaussian.

Combining (90) and (91) we get

$$
\mathbb{E}_{\mathbf{S},\mathbf{W} \sim P_{\mathbf{S},\mathbf{W}}}\Big[\mathrm{gen}(\mathbf{S}, \overline{W}^{(R)})\Big] \leqslant \frac{1}{KR}\sum_{k \in [K], r \in [R]}\left(I(S_k^{(r)}; \overline{W}^{(r-1)}, W_k^{[r:R]})/\lambda_{k,r} + \frac{\lambda_{k,r}\sigma^2}{2n_{k,r}}\right)
$$

$$
\overset{(a)}{\leqslant} \sqrt{\frac{2\sigma^2}{KR}\sum_{k \in [K], r \in [R]}\frac{1}{n_{k,r}}I(S_k^{(r)}; \overline{W}^{(r-1)}, W_k^{[r:R]})}, \tag{92}
$$

where the last step is established by letting $\lambda_{k,r} := n_{k,r}\lambda$ and

$$
\lambda := \sqrt{\frac{2}{\sigma^2 KR}\sum_{k \in [K], r \in [R]}\frac{1}{n_{k,r}}I(S_k^{(r)}; \overline{W}^{(r-1)}, W_k^{[r:R]})}.
$$

This completes the proof. $\qquad\square$

## F.6 PROOF OF THEOREM 6

*Proof.* We start by a lemma proved in Appendix F.11.

**Lemma 4.** *For any $\delta > 0$,*

$$
\log \mathbb{P}\Big(\mathrm{gen}(\mathbf{S}, \overline{W}^{(R)}) > \Delta\Big) \leqslant \max\Big(\log(\delta), \tag{93}
$$

$$
\sup_{\nu_{\mathbf{S},\mathbf{U},\mathbf{W}} \in \mathcal{G}^{\delta}_{\mathbf{S},\mathbf{U},\mathbf{W}}} \inf_{\left\{p_{\hat{W}_k^{(r)}|S_k^{(r)}, U_k^{(r)}, \overline{W}^{(r-1)}} \in \mathcal{Q}_{k,r}(\nu_{\mathbf{S},\mathbf{U},\mathbf{W}})\right\}_{k \in [K], r \in [R]}} \inf_{\lambda \geqslant 0}\Big\{-D_{KL}(\nu_{\mathbf{S},\mathbf{U},\mathbf{W}} \| P_{\mathbf{S},\mathbf{U},\mathbf{W}}) - \frac{\lambda}{KR}\sum_{r,k}C_{k,r}(\nu,p)\Big\}\Big)
$$

*where*

$$
C_{k,r}(\nu,p) := \Delta - \epsilon - \mathbb{E}_{\nu_{\mathbf{S},\mathbf{U}}(\nu p)_{k,r}}\left[\mathrm{gen}(S_k^{(r)}, \hat{\overline{W}}^{(R)})\right],
$$

*with the notation*

$$
(\nu p)_{k,r} := \nu_{\overline{W}^{(r-1)}, W_{[K]\backslash k}^{(r)}|S_{[K]}^{[r]}\backslash S_k^{(r)}, U_{[K]}^{[r]}\backslash U_k^{(r)}} \, p_{\hat{W}_k^{(r)}|S_k^{(r)}, U_k^{(r)}, \overline{W}^{(r-1)}} \, \nu_{\hat{\overline{W}}^{(R)}|\hat{W}_k^{(r)}, W_{[K]\backslash k}^{(r)}, S_{[K]}^{[r+1:R]}, U_{[K]}^{[r+1:R]}},
$$

*and where $\mathcal{Q}_{k,r}(\nu_{\mathbf{S},\mathbf{U},\mathbf{W}})$ contains all distributions $\left\{p_{\hat{W}_{k,r}|S_k^{(r)}, U_k^{(r)}, \overline{W}^{(r-1)}}\right\}_{k \in [K], r \in [R]}$ that satisfy*

$$
\mathbb{E}_{\nu_{\mathbf{S},\mathbf{U},\mathbf{W}}}\left[\mathrm{gen}(\mathbf{S}, \overline{W}^{(R)})\right] - \frac{1}{KR}\sum_{k \in [K], r \in [R]}\mathbb{E}_{\nu_{\mathbf{S},\mathbf{U}}(\nu p)_{k,r}}\left[\mathrm{gen}(S_k^{(r)}, \hat{\overline{W}}^{(R)})\right] \leqslant \epsilon. \tag{94}
$$

Next, consider the set $\tilde{\mathcal{Q}}_{k,r}(\nu_{\mathbf{S},\mathbf{U},\mathbf{W}})$ that contains all distributions $\left\{p_{\hat{W}_{k,r}|S_k^{(r)}, U_k^{(r)}, \overline{W}^{(r-1)}}\right\}_{k \in [K], r \in [R]}$ such that for any $k \in [k]$ and $r \in [R]$,

$$
\mathbb{E}_{\nu_{\mathbf{S},\mathbf{U},\mathbf{W}}}\left[\mathrm{gen}(S_k^{(r)}, \overline{W}^{(R)})\right] - \mathbb{E}_{\nu_{\mathbf{S},\mathbf{U}}(\nu p)_{k,r}}\left[\mathrm{gen}(S_k^{(r)}, \hat{\overline{W}}^{(R)})\right] \leqslant \epsilon_{k,r}, \tag{95}
$$

where $\frac{1}{KR}\sum_{k \in [K], r \in [R]}\epsilon_{k,r} = \epsilon$. Trivially, $\tilde{\mathcal{Q}}_{k,r}(\nu_{\mathbf{S},\mathbf{U},\mathbf{W}}) \subseteq \mathcal{Q}_{k,r}(\nu_{\mathbf{S},\mathbf{U},\mathbf{W}})$, and hence, for a given $\nu_{\mathbf{S},\mathbf{U},\mathbf{W}} \in \mathcal{G}^{\delta}_{\mathbf{S},\mathbf{U},\mathbf{W}}$, it suffices to bound

$$
\inf_{\left\{p_{\hat{W}_k^{(r)}|S_k^{(r)}, \overline{W}^{(r-1)}} \in \tilde{\mathcal{Q}}_{k,r}(\nu_{\mathbf{S},\mathbf{U},\mathbf{W}})\right\}_{k \in [K], r \in [R]}} \inf_{\lambda \geqslant 0}\Big\{-D_{KL}(\nu_{\mathbf{S},\mathbf{U},\mathbf{W}} \| P_{\mathbf{S},\mathbf{U},\mathbf{W}}) - \frac{\lambda}{KR}\sum_{r,k}C_{k,r}(\nu,p)\Big\}.
$$

$$
\tag{96}
$$

Fix a $\nu_{\mathbf{S},\mathbf{U},\mathbf{W}} \in \mathcal{G}^{\delta}_{\mathbf{S},\mathbf{U},\mathbf{W}}$. Let $q_{\hat{W}^{(r)}_k|U^{(r)}_k,\overline{W}^{(r-1)}}$ be the marginal conditional distribution of $\hat{W}^{(r)}_k$ given $\overline{W}^{(r-1)}$ and $U^{(r)}_k$ under $\nu_{S^{(r)}_k,U^{(r)}_k,\overline{W}^{(r-1)}}p_{\hat{W}^{(r)}_k|S^{(r)}_k,U^{(r)}_k,\overline{W}^{(r-1)}}$, and denote similarly

$$(\nu q)_{k,r} := \nu_{\overline{W}^{(r-1)},W^{(r)}_{[K]\setminus k}|S^{[r]}_{[K]}\setminus S^{(r)}_k,U^{[r]}_{[K]}\setminus U^{(r)}_k}\, q_{\hat{W}^{(r)}_k|U^{(r)}_k,\overline{W}^{(r-1)}}\, \nu_{\hat{\overline{W}}^{(R)}|\hat{W}^{(r)}_k,W^{(r)}_{[K]\setminus k},S^{[r+1:R]}_{[K]},U^{[r+1:R]}_{[K]}},$$

Note that

$$\lambda\mathbb{E}_{\nu_{\mathbf{S},\mathbf{U}}(\nu p)_{k,r}}\left[\mathrm{gen}(S^{(r)}_k,\hat{\overline{W}}^{(R)})\right]$$

$$\overset{(a)}{\leqslant} \mathbb{E}_{\nu_{\mathbf{S},\mathbf{U}}}\left[D_{KL}\big((\nu p)_{k,r}\|(\nu q)_{k,r}\big) + \log\mathbb{E}_{(\nu q)_{k,r}}\left[e^{\lambda\,\mathrm{gen}(S^{(r)}_k,\hat{\overline{W}}^{(R)})}\right]\right]$$

$$\overset{(b)}{\leqslant} \mathbb{E}_{\nu_{\mathbf{S},\mathbf{U}}}\left[D_{KL}\big((\nu p)_{k,r}\|(\nu q)_{k,r}\big)\right] + D_{KL}\big(\nu_{\mathbf{S},\mathbf{U}}\|P_{\mathbf{S},\mathbf{U}}\big) + \log\mathbb{E}_{P_{\mathbf{S},\mathbf{U}}(\nu q)_{k,r}}\left[e^{\lambda\,\mathrm{gen}(S^{(r)}_k,\hat{\overline{W}}^{(R)})}\right]$$

$$\overset{(c)}{\leqslant} D_{KL}(\nu_{\mathbf{S},\mathbf{U},\mathbf{W}}\|P_{\mathbf{S},\mathbf{U},\mathbf{W}}) + \mathbb{E}_{\nu_{S^{(r)}_k,U^{(r)}_k,\overline{W}^{(r-1)}}}\left[D_{KL}\big(p_{\hat{W}^{(r)}_k|S^{(r)}_k,U^{(r)}_k,\overline{W}^{(r-1)}}\|q_{\hat{W}^{(r)}_k|U^{(r)}_k,\overline{W}^{(r-1)}}\big)\right]$$

$$\qquad + \log\mathbb{E}_{P_{\mathbf{S},\mathbf{U}}(\nu q)_{k,r}}\left[e^{\lambda\,\mathrm{gen}(S^{(r)}_k,\hat{\overline{W}}^{(R)})}\right]$$

$$= D_{KL}(\nu_{\mathbf{S},\mathbf{U},\mathbf{W}}\|P_{\mathbf{S},\mathbf{U},\mathbf{W}}) + I(S^{(r)}_k;\hat{W}^{(r)}_k|U^{(r)}_k,\overline{W}^{(r-1)}) + \log\mathbb{E}_{P_{\mathbf{S},\mathbf{U}}(\nu q)_{k,r}}\mathbb{E}_{P_{S^{(r)}_k}}\left[e^{\lambda\,\mathrm{gen}(S^{(r)}_k,\hat{\overline{W}}^{(R)})}\right]$$

$$\overset{(d)}{\leqslant} D_{KL}(\nu_{\mathbf{S},\mathbf{U},\mathbf{W}}\|P_{\mathbf{S},\mathbf{U},\mathbf{W}}) + I(S^{(r)}_k;\hat{W}^{(r)}_k|U^{(r)}_k,\overline{W}^{(r-1)}) + \frac{\lambda^2\sigma^2}{2n/R}, \tag{97}$$

where $(a)$ and $(b)$ are derived using Donsker-Varadhan's inequality, $(c)$ by definitions of $(\nu p)_{k,r}$ and $(\nu q)_{k,r}$, and $(d)$ using the fact that for any $w \in \mathcal{W}$, $\mathrm{gen}(S^{(r)}_k,w)$ is $\sigma/\sqrt{n/R}$-subgaussian.

Hence,

$$\inf_{\left\{p_{\hat{W}^{(r)}_k|S^{(r)}_k,U^{(r)}_k,\overline{W}^{(r-1)}}\in\mathcal{Q}_{k,r}(\nu_{\mathbf{S},\mathbf{U},\mathbf{W}})\right\}_{k\in[K],r\in[R]}}\ \inf_{\lambda\geqslant 0}\left\{-D_{KL}(\nu_{\mathbf{S},\mathbf{U},\mathbf{W}}\|P_{\mathbf{S},\mathbf{U},\mathbf{W}}) - \frac{\lambda}{KR}\sum_{r,k}C_{k,r}(\nu,p)\right\}$$

$$\leqslant \inf_{\left\{p_{\hat{W}^{(r)}_k|S^{(r)}_k,U^{(r)}_k,\overline{W}^{(r-1)}}\in\tilde{\mathcal{Q}}_{k,r}(\nu_{\mathbf{S},\mathbf{U},\mathbf{W}})\right\}_{k\in[K],r\in[R]}}\ \inf_{\lambda\geqslant 0}\left\{-D_{KL}(\nu_{\mathbf{S},\mathbf{U},\mathbf{W}}\|P_{\mathbf{S},\mathbf{U},\mathbf{W}}) - \frac{\lambda}{KR}\sum_{r,k}C_{k,r}(\nu,p)\right\}$$

$$\leqslant \inf_{\left\{p_{\hat{W}^{(r)}_k|S^{(r)}_k,U^{(r)}_k,\overline{W}^{(r-1)}}\in\tilde{\mathcal{Q}}_{k,r}(\nu_{\mathbf{S},\mathbf{U},\mathbf{W}})\right\}_{k\in[K],r\in[R]}}\ \inf_{\lambda\geqslant 0}\left\{-\lambda(\Delta-\epsilon) + \frac{1}{KR}\sum_{r,k}I(S^{(r)}_k;\hat{W}^{(r)}_k|U^{(r)}_k,\overline{W}^{(r-1)}) + \frac{\lambda^2\sigma^2}{2n/R}\right\}$$

$$= \inf_{\lambda\geqslant 0}\left\{-\lambda(\Delta-\epsilon) + \frac{1}{KR}\sum_{r,k}\mathfrak{RD}(\nu_{\mathbf{S},\mathbf{U},\mathbf{W}},k,r,\epsilon_{k,r}) + \frac{\lambda^2\sigma^2}{2n/R}\right\}$$

$$\leqslant \log(\delta), \tag{98}$$

where the last step is established by letting

$$\lambda := \sqrt{\frac{(2n/R)}{\sigma^2}\left(\frac{1}{KR}\sum_{k\in[K],r\in[R]}\mathfrak{RD}(\nu_{\mathbf{S},\mathbf{U},\mathbf{W}},k,r,\epsilon_{k,r}) + \log(1/\delta)\right)}.$$

This completes the proof. $\qquad\square$

## F.7 PROOF OF PROPOSITION 1

*Proof.* Recall that

$$\mathfrak{V}_{k,r} = \left(V^{(r)}_k,\overline{W}^{(r-1)},W^{(r)}_{[K]\setminus k},S^{[r+1:R]}_{[K]},V^{[r+1:R]}_{[K]}\right).$$

Consider any set of distributions $\left\{ p_{\hat{\overline{W}}_k^{(R)}|S_k^{(r)},\mathfrak{V}_{k,r}} \right\}_{k\in[K],r\in[R]}$ that satisfy the distortion criterion (33) for any $k\in[K]$ and $r\in[R]$. Then,

$$
\begin{aligned}
\mathbb{E}_{P_{\mathbf{S},\mathbf{w}}}\left[\text{gen}(\mathbf{S},\overline{W}^{(R)})\right] &= \mathbb{E}_{P_{\mathbf{S},\mathbf{v},\mathbf{w}}}\left[\text{gen}(\mathbf{S},\overline{W}^{(R)})\right] \\
&\leqslant \frac{1}{KR}\sum_{k\in[K],r\in[R]}\left(\mathbb{E}_{P_{S_k^{(r)},\mathfrak{V}_{k,r}}p_{\hat{\overline{W}}_k^{(R)}|S_k^{(r)},\mathfrak{V}_{k,r}}}\left[\text{gen}(S_k^{(r)},\hat{\overline{W}}^{(R)})\right]+\epsilon_{k,r}\right) \\
&\leqslant \frac{1}{KR}\sum_{k\in[K],r\in[R]}\mathbb{E}_{P_{S_k^{(r)},\mathfrak{V}_{k,r}}p_{\hat{\overline{W}}_k^{(R)}|S_k^{(r)},\mathfrak{V}_{k,r}}}\left[\text{gen}(S_k^{(r)},\hat{\overline{W}}^{(R)})\right]+\epsilon. \quad (99)
\end{aligned}
$$

Let $q_{\hat{W}_k^{(R)}|\mathfrak{V}_{k,r}}$ be the marginal conditional distribution of $\hat{W}_k^{(R)}$ given $\mathfrak{V}_{k,r}$ under $P_{S_k^{(r)}}p_{\hat{W}_k^{(R)}|S_k^{(r)},\mathfrak{V}_{k,r}}$. Then,

$$
\begin{aligned}
&\lambda\mathbb{E}_{P_{S_k^{(r)},\mathfrak{V}_{k,r}}p_{\hat{\overline{W}}_k^{(R)}|S_k^{(r)},\mathfrak{V}_{k,r}}}\left[\text{gen}(S_k^{(r)},\hat{\overline{W}}^{(R)})\right] \\
&\stackrel{(a)}{\leqslant} D_{KL}\left(P_{S_k^{(r)},\mathfrak{V}_{k,r}}p_{\hat{\overline{W}}_k^{(R)}|S_k^{(r)},\mathfrak{V}_{k,r}}\Big\|P_{S_k^{(r)},\mathfrak{V}_{k,r}}q_{\hat{\overline{W}}_k^{(R)}|\mathfrak{V}_{k,r}}\right)+\log\mathbb{E}_{P_{S_k^{(r)},\mathfrak{V}_{k,r}}q_{\hat{\overline{W}}_k^{(R)}|\mathfrak{V}_{k,r}}}\left[e^{\lambda\text{gen}(S_k^{(r)},\hat{\overline{W}}^{(R)})}\right] \\
&= I(S_k^{(r)};\hat{W}_k^{(R)}|\mathfrak{V}_{k,r})+\log\mathbb{E}_{P_{\mathfrak{V}_{k,r}}q_{\hat{\overline{W}}_k^{(R)}|\mathfrak{V}_{k,r}}}\mathbb{E}_{P_{S_k^{(r)}}}\left[e^{\lambda\text{gen}(S_k^{(r)},\hat{\overline{W}}^{(R)})}\right] \\
&\stackrel{(b)}{\leqslant} I(S_k^{(r)};\hat{W}_k^{(R)}|\mathfrak{V}_{k,r})+\frac{\lambda^2\sigma^2}{2n/R}, \quad (100)
\end{aligned}
$$

where $(a)$ is deduced using Donsker-Varadhan's inequality and $(b)$ using the fact that for any $w\in\mathcal{W}$, $\text{gen}(S_k^{(r)},w)$ is $\sigma/\sqrt{n/R}$-subgaussian,

Combining (99) and (100), and taking the infimum over all admissible $\left\{p_{\hat{\overline{W}}_k^{(R)}|S_k^{(r)},\mathfrak{V}_{k,r}}\right\}_{k\in[K],r\in[R]}$, we get

$$
\begin{aligned}
&\mathbb{E}_{\mathbf{S},\mathbf{W}\sim P_{\mathbf{S},\mathbf{w}}}\left[\text{gen}(\mathbf{S},\overline{W}^{(R)})\right] \\
&\leqslant \frac{1}{KR}\sum_{k\in[K],r\in[R]}\inf_{p_{\hat{W}_k^{(R)}|S_k^{(r)},\mathfrak{V}_{k,r}}}\left\{\mathbb{E}_{P_{S_k^{(r)},\mathfrak{V}_{k,r}}p_{\hat{\overline{W}}_k^{(R)}|S_k^{(r)},\mathfrak{V}_{k,r}}}\left[\text{gen}(S_k^{(r)},\hat{\overline{W}}^{(R)})\right]\right\}+\epsilon \\
&\leqslant \frac{1}{KR}\sum_{k\in[K],r\in[R]}\mathfrak{RD}^\star(P_{\mathbf{S},\mathbf{W}},k,r,\epsilon_{k,r})/\lambda+\frac{\lambda\sigma^2}{2n/R}+\epsilon \\
&\stackrel{(a)}{\leqslant}\sqrt{\frac{2\sigma^2\sum_{k\in[K],r\in[R]}\mathfrak{RD}^\star(P_{\mathbf{S},\mathbf{W}},k,r,\epsilon_{k,r})}{nK}}+\epsilon, \quad (101)
\end{aligned}
$$

where the last step is established by letting

$$
\lambda := \sqrt{\frac{2n/R}{\sigma^2 KR}\sum_{k\in[K],r\in[R]}\mathfrak{RD}^\star(P_{\mathbf{S},\mathbf{W}},k,r,\epsilon_{k,r})}.
$$

This completes the proof. $\qquad\square$

## F.8 PROOF OF LEMMA 1

*Proof.* This step is similar to (Sefidgaran et al., 2022b, Lemma 24) and (Sefidgaran & Zaidi, 2023, Lemma 12). Denote $\mathcal{B} := \left\{\mathbf{s}\in\text{supp}(P_{\mathbf{S}}): \mathbb{E}_{\mathbf{W}\sim P_{\mathbf{W}|\mathbf{s}}}\left[\text{gen}(\mathbf{s},\overline{W}^{(R)})\right]^2 > \Delta(\mathbf{s})\right\}$. If $P_{\mathbf{S}}(\mathcal{B}) = 0$, then the lemma is proved. Assume then $P_{\mathbf{S}}(\mathcal{B}) > 0$. Consider the distribution $\nu_{\mathbf{S}}$ such that for any $\mathbf{s}\in\mathcal{B}$, $\nu_{\mathbf{S}}(\mathbf{s}) := P_{\mathbf{S}}(\mathbf{s})/P_{\mathbf{S}}(\mathcal{B})$, and otherwise $\nu_{\mathbf{S}}(\mathbf{s}) := 0$.

If $\nu_{\mathbf{S}}\notin\mathcal{G}_{\mathbf{S}}^\delta$, then it means that $D_{KL}(\nu_{\mathbf{S}}\|P_{\mathbf{S}})\geqslant\log(1/\delta)$. Hence,

$$
\log(1/\delta)\leqslant D_{KL}(\nu_{\mathbf{S}}\|P_{\mathbf{S}}) = -\log(P_{\mathbf{S}}(\mathcal{B})) = -\log\mathbb{P}\left(\mathbb{E}_{\mathbf{W}\sim P_{\mathbf{W}|\mathbf{s}}}\left[\text{gen}(\mathbf{s},\overline{W}^{(R)})\right]^2 > \Delta(\mathbf{s})\right),
$$

Thus, $\log \mathbb{P}\left(\mathbb{E}_{\mathbf{W} \sim P_{\mathbf{W}|\mathbf{s}}}\left[\text{gen}(\mathbf{s}, \overline{W}^{(R)})\right]^2 > \Delta(\mathbf{s})\right) \leqslant \log(\delta)$. This completes the proof of the lemma.

Otherwise, suppose that $\nu_{\mathbf{S}} \in \mathcal{G}_{\mathbf{S}}^{\delta}$. Then, for this distribution

$$\mathbb{E}_{\nu_{\mathbf{S}}}\left[\Delta(\mathbf{S}) - \mathbb{E}_{\mathbf{W} \sim P_{\mathbf{W}|\mathbf{s}}}\left[\text{gen}(\mathbf{S}, \overline{W}^{(R)})\right]^2\right] \overset{(a)}{\leqslant} 0,$$

where $(a)$ holds by the way $\nu_{\mathbf{S}}$ is constructed. Therefore, for this $\nu_{\mathbf{S}}$, since $\lambda \geqslant 0$,

$$-D_{KL}(\nu_{\mathbf{S}} \| P_{\mathbf{S}}) - \lambda \mathbb{E}_{\nu_{\mathbf{S}}}\left[\Delta(\mathbf{S}) - \mathbb{E}_{\mathbf{W} \sim P_{\mathbf{W}|\mathbf{s}}}\left[\text{gen}(\mathbf{S}, \overline{W}^{(R)})\right]^2\right]$$
$$\geqslant -D_{KL}(\nu_{\mathbf{S}} \| P_{\mathbf{S}})$$
$$= \log(P_{\mathbf{S}}(\mathcal{B}))$$
$$= \log \mathbb{P}\left(\mathbb{E}_{\mathbf{W} \sim P_{\mathbf{W}|\mathbf{s}}}\left[\text{gen}(\mathbf{S}, \overline{W}^{(R)})\right]^2 > \Delta(\mathbf{S})\right).$$

This proves the lemma. □

### F.9 PROOF OF LEMMA 2

*Proof.* This step is similar to (Sefidgaran et al., 2022b, Lemma 24) and (Sefidgaran & Zaidi, 2023, Lemma 12).

Denote $f(\mathbf{s}, \overline{w}^{(R)}) := \frac{1}{KR} \sum_{k \in [K], r \in [R]} \text{gen}(s_k^{(r)}, \overline{w}^{(R)})^2$ and define the set

$$\mathcal{B} := \left\{(\mathbf{s}, \mathbf{w}) \in \text{supp}(P_{\mathbf{S}, \mathbf{W}}) : f(\mathbf{s}, \overline{w}^{(R)}) > \Delta(\mathbf{s}, \mathbf{w})\right\}.$$

If $P_{\mathbf{S}, \mathbf{W}}(\mathcal{B}) = 0$, then the lemma is proved. Assume then $P_{\mathbf{S}, \mathbf{W}}(\mathcal{B}) > 0$. Consider the distribution $\nu_{\mathbf{S}, \mathbf{W}}$ such that for any $(\mathbf{s}, \mathbf{w}) \in \mathcal{B}$, $\nu_{\mathbf{S}, \mathbf{W}}(\mathbf{s}, \mathbf{w}) := P_{\mathbf{S}, \mathbf{W}}(\mathbf{s}, \mathbf{w})/P_{\mathbf{S}, \mathbf{W}}(\mathcal{B})$, and otherwise $\nu_{\mathbf{S}, \mathbf{W}}(\mathbf{s}, \mathbf{w}) := 0$.

If $\nu_{\mathbf{S}, \mathbf{W}} \notin \mathcal{G}_{\mathbf{S}, \mathbf{W}}^{\delta}$, then it means that $D_{KL}(\nu_{\mathbf{S}, \mathbf{W}} \| P_{\mathbf{S}, \mathbf{W}}) \geqslant \log(1/\delta)$. Hence,

$$\log(1/\delta) \leqslant D_{KL}(\nu_{\mathbf{S}, \mathbf{W}} \| P_{\mathbf{S}, \mathbf{W}}) = -\log(P_{\mathbf{S}, \mathbf{W}}(\mathcal{B})) = -\log \mathbb{P}\left(f(\mathbf{s}, \overline{w}^{(R)}) > \Delta(\mathbf{s}, \mathbf{w})\right),$$

Thus, $\log \mathbb{P}\left(f(\mathbf{s}, \overline{w}^{(R)}) > \Delta(\mathbf{s}, \mathbf{w})\right) \leqslant \log(\delta)$. This completes the proof of the lemma.

Otherwise, suppose that $\nu_{\mathbf{S}, \mathbf{W}} \in \mathcal{G}_{\mathbf{S}, \mathbf{W}}^{\delta}$. Then, for this distribution, we have

$$\mathbb{E}_{\nu_{\mathbf{S}, \mathbf{W}}}\left[\Delta(\mathbf{S}, \mathbf{W}) - f(\mathbf{S}, \overline{W}^{(R)})\right] \overset{(a)}{\leqslant} 0,$$

where $(a)$ holds due to the way $\nu_{\mathbf{S}, \mathbf{W}}$ is constructed. Therefore, it is optimal to let $\lambda = 0$ in $(b)$ for this distribution. Thus, the infimum over $\lambda > 0$ of

$$-D_{KL}(\nu_{\mathbf{S}, \mathbf{W}} \| P_{\mathbf{S}, \mathbf{W}}) - \lambda \mathbb{E}_{\nu_{\mathbf{S}, \mathbf{W}}}\left[\Delta(\mathbf{S}, \mathbf{W}) - f(\mathbf{S}, \overline{W}^{(R)})\right],$$

is equal to

$$-D_{KL}(\nu_{\mathbf{S}, \mathbf{W}} \| P_{\mathbf{S}, \mathbf{W}}) = \log(P_{\mathbf{S}, \mathbf{W}}(\mathcal{B})) = \log \mathbb{P}\left(f(\mathbf{S}, \overline{W}^{(R)}) > \Delta(\mathbf{S}, \mathbf{W})\right).$$

This completes the proof of the lemma. □

### F.10 PROOF OF LEMMA 3

*Proof.* This step is similar to (Sefidgaran et al., 2022b, Lemma 24) and (Sefidgaran & Zaidi, 2023, Lemma 12). Denote $\mathcal{B} := \left\{\mathbf{s} \in \text{supp}(P_{\mathbf{S}}) : \mathbb{E}_{\mathbf{W} \sim P_{\mathbf{W}|\mathbf{s}}}\left[\text{gen}(\mathbf{s}, \overline{W}^{(R)})\right]^2 > \Delta(\mathbf{s})\right\}$. If $P_{\mathbf{S}}(\mathcal{B}) = 0$, then the lemma is proved. Assume then $P_{\mathbf{S}}(\mathcal{B}) > 0$. Consider the distribution $\nu_{\mathbf{S}}$ such that for any $\mathbf{s} \in \mathcal{B}$, $\nu_{\mathbf{S}}(\mathbf{s}) := P_{\mathbf{S}}(\mathbf{s})/P_{\mathbf{S}}(\mathcal{B})$, and otherwise $\nu_{\mathbf{S}}(\mathbf{s}) := 0$.

If $\nu_{\mathbf{S}} \notin \mathcal{G}_{\mathbf{S}}^{\delta}$, then it means that $D_{KL}(\nu_{\mathbf{S}} \| P_{\mathbf{S}}) \geqslant \log(1/\delta)$. Hence,

$$\log(1/\delta) \leqslant D_{KL}(\nu_{\mathbf{S}} \| P_{\mathbf{S}}) = -\log(P_{\mathbf{S}}(\mathcal{B})) = -\log \mathbb{P}\left(\mathbb{E}_{\mathbf{W} \sim P_{\mathbf{W}|\mathbf{s}}}\left[\text{gen}(\mathbf{s}, \overline{W}^{(R)})\right]^2 > \Delta(\mathbf{s})\right)$$

Thus, $\log \mathbb{P}\left(\mathbb{E}_{\mathbf{W} \sim P_{\mathbf{W}|\mathbf{s}}}\left[\mathrm{gen}(\mathbf{s}, \overline{W}^{(R)})\right]^2 > \Delta(\mathbf{s})\right), \leqslant \log(\delta)$. This completes the proof of the lemma.

Otherwise, suppose that $\nu_{\mathbf{S}} \in \mathcal{G}_{\mathbf{S}}^\delta$. Then, for this distribution

$$\mathbb{E}_{\nu_{\mathbf{S}}}\left[\Delta(\mathbf{S}) - \epsilon - \tilde{\Delta}(\mathbf{S})^2\right] \overset{(a)}{\leqslant} \mathbb{E}_{\nu_{\mathbf{S}}}\left[\mathbb{E}_{\mathbf{W} \sim P_{\mathbf{W}|\mathbf{s}}}\left[\mathrm{gen}(\mathbf{s}, \overline{W}^{(R)})\right]^2 - \epsilon - \tilde{\Delta}(\mathbf{S})^2\right]$$

$$\overset{(b)}{\leqslant} \mathbb{E}_{\nu_{\mathbf{S}}}\left[2\left|\mathbb{E}_{\mathbf{W} \sim P_{\mathbf{W}|\mathbf{s}}}\left[\mathrm{gen}(\mathbf{s}, \overline{W}^{(R)})\right] - \tilde{\Delta}(\mathbf{S})\right| - \epsilon\right]$$

$$\overset{(c)}{\leqslant} 0,$$

where $(a)$ holds by the way $\nu_{\mathbf{S}}$ is constructed, $(b)$ holds since the loss function is bounded by one, and $(c)$ due to (49).

Therefore, for this $\nu_{\mathbf{S}}$, since $\lambda \geqslant 0$,

$$-D_{KL}(\nu_{\mathbf{S}} \| P_{\mathbf{S}}) - \lambda \mathbb{E}_{\nu_{\mathbf{S}}}\left[\Delta(\mathbf{S}) - \epsilon - \tilde{\Delta}(\mathbf{S})^2\right] \geqslant - D_{KL}(\nu_{\mathbf{S}} \| P_{\mathbf{S}})$$

$$= \log(P_{\mathbf{S}}(\mathcal{B}))$$

$$= \log \mathbb{P}\left(\mathbb{E}_{\mathbf{W} \sim P_{\mathbf{W}|\mathbf{s}}}\left[\mathrm{gen}(\mathbf{s}, \overline{W}^{(R)})\right]^2 > \Delta(\mathbf{s})\right).$$

This proves the lemma. $\qquad\square$

## F.11 Proof of Lemma 4

*Proof.* Denote $\mathcal{B} := \left\{(\mathbf{s}, \mathbf{u}, \mathbf{w}) \in \mathrm{supp}(P_{\mathbf{S},\mathbf{U},\mathbf{W}}) : \mathrm{gen}(\mathbf{s}, \overline{w}^{(R)}) > \Delta\right\}$. If $P_{\mathbf{S},\mathbf{U},\mathbf{W}}(\mathcal{B}) = 0$, then the lemma is proved. Assume then $P_{\mathbf{S},\mathbf{U},\mathbf{W}}(\mathcal{B}) > 0$. Consider the distribution $\nu_{\mathbf{S},\mathbf{U},\mathbf{W}}$ such that for any $(\mathbf{s}, \mathbf{u}, \mathbf{w}) \in \mathcal{B}$, $\nu_{\mathbf{S},\mathbf{U},\mathbf{W}}(\mathbf{s}, \mathbf{u}, \mathbf{w}) := P_{\mathbf{S},\mathbf{U},\mathbf{W}}(\mathbf{s}, \mathbf{u}, \mathbf{w})/P_{\mathbf{S},\mathbf{U},\mathbf{W}}(\mathcal{B})$, and otherwise $\nu_{\mathbf{S},\mathbf{U},\mathbf{W}}(\mathbf{s}, \mathbf{u}, \mathbf{w}) := 0$.

If $\nu_{\mathbf{S},\mathbf{U},\mathbf{W}} \notin \mathcal{G}_{\mathbf{S},\mathbf{U},\mathbf{W}}^\delta$, then it means that $D_{KL}(\nu_{\mathbf{S},\mathbf{U},\mathbf{W}} \| P_{\mathbf{S},\mathbf{U},\mathbf{W}}) \geqslant \log(1/\delta)$. Hence,

$$\log(1/\delta) \leqslant D_{KL}(\nu_{\mathbf{S},\mathbf{U},\mathbf{W}} \| P_{\mathbf{S},\mathbf{U},\mathbf{W}}) = -\log(P_{\mathbf{S},\mathbf{U},\mathbf{W}}(\mathcal{B})) = -\log \mathbb{P}\left(\mathrm{gen}(\mathbf{s}, \overline{w}^{(R)}) > \Delta\right),$$

Thus, $\log \mathbb{P}\left(\mathrm{gen}(\mathbf{s}, \overline{w}^{(R)}) > \Delta\right) \leqslant \log(\delta)$. This completes the proof of the lemma.

Otherwise, suppose that $\nu_{\mathbf{S},\mathbf{U},\mathbf{W}} \in \mathcal{G}_{\mathbf{S},\mathbf{U},\mathbf{W}}^\delta$. Then, for this distribution and any set of $\inf_{\left\{P_{\hat{W}_k^{(r)}|S_k^{(r)}, U_k^{(r)}, \overline{W}^{(r-1)}} \in \mathcal{Q}_{k,r}(\nu_{\mathbf{S},\mathbf{U},\mathbf{W}})\right\}_{k \in [K], r \in [R]}}$, we have

$$\frac{1}{KR} \sum_{k,r} C_{k,r}(\nu, p)$$

$$= \Delta - \epsilon - \frac{1}{KR} \sum_{k \in [K], r \in [R]} \left(\mathbb{E}_{\nu_{\mathbf{S},\mathbf{U}}(\nu p)_{k,r}}\left[\mathrm{gen}_{k,r}(S_k^{(r)}, \hat{\overline{W}}^{(R)})\right]\right)$$

$$\overset{(a)}{\leqslant} \mathbb{E}_{\nu_{\mathbf{S},\mathbf{U},\mathbf{W}}}\left[\mathrm{gen}(\mathbf{S}, \overline{W}^{(R)})\right] - \frac{1}{KR} \sum_{k \in [K], r \in [R]} \left(\mathbb{E}_{\nu_{\mathbf{S},\mathbf{U}}(\nu p)_{k,r}}\left[\mathrm{gen}_{k,r}(S_k^{(r)}, \hat{\overline{W}}^{(R)})\right]\right) - \epsilon$$

$$\overset{(b)}{\leqslant} 0,$$

where $(a)$ holds due to the way $\nu_{\mathbf{S},\mathbf{W}}$ is constructed and $(b)$ by distortion criterion (94).

Therefore, it is optimal to let $\lambda = 0$ in (93)for this distribution. Thus, the infimum over $\lambda > 0$ of

$$-D_{KL}(\nu_{\mathbf{S},\mathbf{U},\mathbf{W}} \| P_{\mathbf{S},\mathbf{U},\mathbf{W}}) - \frac{\lambda}{KR} \sum_{k,r} C_{k,r}(\nu, p),$$

is equal to

$$-D_{KL}(\nu_{\mathbf{S},\mathbf{U},\mathbf{W}} \| P_{\mathbf{S},\mathbf{U},\mathbf{W}}) = \log(P_{\mathbf{S},\mathbf{U},\mathbf{W}}(\mathcal{B})) = \log \mathbb{P}\left(\mathrm{gen}(\mathbf{S}, \overline{W}^{(R)}) > \Delta\right).$$

This completes the proof. $\qquad\square$

