# OpenReview forum: "Federated Learning, Lessons from Generalization Study: Communicate Less, Learn More"
_ICLR.cc/2024/Conference — Submitted to ICLR 2024_

### Official Review · Reviewer_yDVy · 2023-10-22

**Soundness:** 3 good
**Presentation:** 3 good
**Contribution:** 2 fair
**Rating:** 5
**Confidence:** 4

**Summary:**

The paper examines generalization error in Federated Learning (FL) and focuses on the impact of communication rounds (R) on this error. It introduces PAC-Bayes and rate-distortion bounds for FL and applies them to Federated Support Vector Machines (FSVM). The study finds that as R increases, the generalization error of FSVM worsens, suggesting that more frequent communication with the parameter server diminishes its generalization power. The paper also shows that, for any R, FL outperforms centralized learning by a factor proportional to O(log(K)/K). Experiments with neural networks support these findings.

**Strengths:**

The strengths of this paper are as follows:

1. **Novel Bounds**: The paper introduces novel PAC-Bayes and rate-distortion theoretic bounds that explicitly consider the influence of the number of communication rounds (R), the number of participating devices (K), and dataset size (n) on generalization error. These bounds are the first of their kind for the problem addressed in the study.
2. **Modeling Contributions**: The research provides insight into the structure of distributed interactive learning algorithms, showing how each client's contribution at each round affects the generalization error of the final model.
3. **Applicability to FSVM**: The paper applies these theoretical bounds to Federated Support Vector Machines (FSVM) and derives explicit bounds for generalization error. Notably, it reveals that more frequent communication with the parameter server reduces the generalization power of FSVM algorithms. This suggests that the parameter R can be optimized to minimize the population risk of FSVM.
4. **Generalization of Findings**: The research demonstrates that the generalization error of the FSVM setting decreases faster than that of centralized learning by a factor proportional to O(log(K)/K) for any value of R. This generalizes recent findings for "one-shot" Federated Learning to any arbitrary number of rounds.

**Weaknesses:**

1. **Problem Setting**: The paper assumes a partitioning of data (n samples) into R disjoint subsets for each client, with each subset used in one communication round. This assumption may be considered strong and unrealistic, as many Federated Learning (FL) approaches typically use the entire batch for training in each round. Furthermore, in real-world scenarios, a mix of old and new data is often present in an online approach.

2. **Counter-Intuitive Observation**: Moreover, if we consider a fixed n, each client ends up with n/R data for each communication round. To illustrate this with a special case, let's take R=1, which essentially reduces the scenario to centralized learning. Surprisingly, based on the results presented in the paper, which indicate that the generalization error increases with R, it might suggest that R=1 should be chosen. This finding appears counter-intuitive and raises questions about the practicality and relevance of the assumed setting for real-world federated learning applications.

**Questions:**

I suggest that the author undertake a more extensive exploration of the practical implications stemming from the theoretical findings. It is imperative to conduct an exhaustive examination of the interdependencies between key parameters across various scenarios. Such an in-depth analysis holds the potential to offer invaluable insights into the development of real-world federated learning algorithms. Personally, I hold the theoretical contribution of this paper in high regard, despite my reservations about the underlying settings. A more comprehensive analysis would significantly enhance the overall quality of this paper and can be translated into guidance for the refinement and enhancement of federated learning systems.

---

> ### Author Response · Authors · 2023-11-15
> **Response to Reviewer yDVy**
>
> We thank the reviewer for the time spent on our paper and their interest in our results.
>
> 1. Please refer to our response to all reviewers.
>
> 2. Note that the case $R=1$ is not the centralized learning algorithm, but the "one-shot" FL, where only one aggregation is performed. Indeed, in this case we obtain the best generalization performance, intuitively, since "the variance of the model" becomes small after one aggregation of the independent models. In contrast, it is hardly possible to minimize the empirical risk using only a single aggregation step. Therefore, $R=1$ is rarely the optimal value of $R$ when considering the population risk. In principle, there is a trade-off between the generalization error and the empirical error, as can be explicitly observed in Figure 3.
>
> - Regarding further experiments: We would like first to reiterate that the presented experiments on FSVM mainly aim at validating the behavior suggested by our Theorem 4, \ie that the generalization error of FSVM might increase with $R$. The additional experiments on neural networks are meant to suggest that this behavior may hold in other setups. It should be remarked that the experiment on ResNet-56, presented in the Section 5 of the paper, was reproduced for different sets of hyperparameters, giving similar results. As a complement, we added the reproduced experiments for two different values of the learning rate in Appendix E.6. Finally, we agree that extensive exploration of the influence of optimization hyperparameters on the generalization error as a function of $R$ is indeed an interesting question. We believe however, it is beyond the scope and motivation of this work and therefore left for future work.

---

### Official Review · Reviewer_9uEx · 2023-10-27

**Soundness:** 2 fair
**Presentation:** 2 fair
**Contribution:** 3 good
**Rating:** 5
**Confidence:** 4

**Summary:**

This work provides novel PAC-Bayesian and rate-distortion typed generalisation bounds tailored for federated learning. Contrary to classical bounds designed for batch learning, authors take into account the evolution of the learning phase through successive rounds and now consider the number of rounds as an hyperparameter to optimise to ensure a good tradeoff between empirical performances and generalisation ability. They particularise their results to the case of federated SVMs and provide associated experiments.

**Strengths:**

Having such theoretical bounds tailored for federated learning is novel and provide exciting new leads to understand the efficiency of FL.

**Weaknesses:**

I have concerns about correctness of Theorem 1, some presentation issues and the conclusion of the experiments, see the 'Questions' section.

**Questions:**

-  The definition of $P_{\mathbf{W},\mathbf{S}}$ is unclear. Is it a distribution over $\mathcal{W}^{(K+1)R}$ and then, is the use of the product $\Pi$ equivalent to $\otimes$ ? Otherwise, if it is a distribution over $\mathcal{W}$ can you make this explicit?
- Given the way $S_k$ is partitioned for any $k$, users are not allowed to use their whole dataset at each round but only a smaller fraction $n_R$. Is it a realistic in practice? For instance take the instance of federated learning between hospital to better detect a rare disease (i.e. each user has few data), is it reasonable in this case to force the users not to use their whole dataset each round? Furthermore, this would imply that if $n$ is small, then one would not be allowed to perform many training rounds.  What can the authors say about this?
- In section 2, you said that 'the aggregation function at the PS is set to be deterministic and arbitrary' while in Theorem 1,  $\bar{W}^{[r-1]}$ is drawn according to a probability distribution, what did I miss? Furthermore, how costful is this additional expectation in terms of computational time as it does not appear in many classical PAC-Bayes bounds?
- I found Theorem 1's proof poorly organised, for instance,  $\mathbf{\nu}_{S}$ is defined in Lemma 1's proof, which has been put on another appendix. Similarly, I did not find a definition of $\mathcal{G}_S^{\delta}$ although somewhat inferable from context.
- I don't understand the use of the subgausiannity assumption at the end of page 2022. Let rename $X= gen(s_k^{(r)}, \bar{W}^{(R)})$, then you affirm that because $X$ is $\sigma_{k,r}:= \sqrt{\frac{R}{4n}}$ subgaussian we have $\mathbb{E}[e^{\lambda X^2}]\leq \frac{2n}{R}$.  This is highly non-standard, how do you prove it?  To me, subgaussianity would only imply $\mathbb{E}[e^{\lambda X}]\leq \exp(\frac{\lambda^2 \sigma_{k,r}^2}{2})$.
- About the experiments, I am not convinced that communicating less implies a better learning phase, at least for FVSM. Indeed, this conclusion appears to be true for $K=10,20$ (not the case $K=50$ as the short decrease in the end of the curve exhibits more a stabilisation than a deterioration) and holds when considering the generalisation gap, instead of observing directly the population risk . Thus, if we focus on the notion of population risk, Figure 4 shows that increasing the number of rounds only leads to positive outcomes as the empirical risk continues to decrease while the population risk either decreases or stabilises. A similar conclusion can be derived from Figure 6. I acknowledge that you affirm in section D.2 that 'fewer rounds may be needed, if one can effectively take the “estimated” generalisation error into account', but a practitioner does not have access to this information and see only benefits to continue its training as the empirical risk decreases, and the test error does not vary: there is no stopping criterion in practice.
To me, the interest of a tradeoff in $R$, only appears in Figure 3, which is not covered by Theorem 4.


In conclusion, although my enthusiasm about having PAC-Bayesian guarantees tailored to FL, I believe this paper needs to be rewritten before its acceptance, given my current concerns about the correctness of Theorem 1 and the shift between the message conveyed by the paper (starting from its title) and the proposed experiments.

---

> ### Author Response · Authors · 2023-11-15
> **Response to Reviewer 9uEx**
>
> We thank the reviewer for the time spent on our paper and their interest in the results.
>
>  - **Regarding $P_{\mathbf{S},\mathbf{W}}$:** The reviewer's understanding is correct. It is a probability distribution over $\mathcal{W}^{(K+1)R}$. We agree with the reviewer that $\bigotimes$ is a more accurate notation, that is used in the new version.
>
> - **Regarding the setup:** Please refer to the response to all reviewers.
>
> - **Regarding the aggregation:** First, we mention that there exists a recurrent typo in some of the bounds: $\overline{W}^{[r-1]}$ should be read as $\overline{W}^{(r-1)}$, which is corrected in the new version. We are deeply sorry for this. Now, concerning the question of the reviewer (by considering $\overline{W}^{(r-1)}$): The aggregation function is deterministic. This means that once $W_{[K]}^{(r-1)}$ is known, then $\overline{W}^{(r-1)}$ is deterministic. However, in general, $W_{[K]}^{(r-1)}$ are random due to the randomness in the way the mini-batches $\beta_{k,r,t}$ are picked from $S_k^{(r)}$, for all clients/rounds/iterations and also due to initialization randomness. Hence, $\overline{W}^{(r-1)}$ is also a random variable. Now, regarding the cost of expectation, we agree that if expectation were need to be taken with respect to $\overline{W}^{[r-1]}$, it would be costly. However, we only need to consider the expectation with respect to $\overline{W}^{(r-1)}$. If we now consider, for example, the approach of Wang et al. ("PAC-Bayes Information Bottleneck", ICLR 2022) on estimating the mutual information or KL divergence terms, then this means that we need an extra bootstrapping phase for $\overline{W}^{(r-1)}$. An initial idea is to aggregate a subset of the clients' models $\{W_k^{(r-1)}\}_{k \in [K]}$, drawn uniformly at random, and repeat the process a number of times. This is in the same spirit as the bootstrapping (for other variables) in Wang et al. This further investigation, which is indeed interesting, is left for future work.
>
> - **Regarding $\mathcal{G}_{\textbf{S}}^{\delta}$:** We apologize for omitting the definition of  $\mathcal{G}_{\textbf{S}}^{\delta}$ (that contains $\nu_{\mathbf{S}}$) in the proof of Theorem 1, which happened after reorganization of that proof. We have added this in the new version.
>
> - **Regarding the subgausiannity:** This is a classical property of subgaussian random variables. Borrowing the reviewer's notation, if $X$ is $\sigma_{k,r}$-subgaussian, where $\sigma_{k,r}=\sqrt{\frac{R}{4n}}$, then by Theorem 2.6.IV. of Wainwright, 2019, (available at the link provided below), for any $\lambda_1 \in [0,1)$, $\mathbb{E}\left[e^{\frac{\lambda_1 X^2}{2\sigma_{k,r}^2}}\right]\leq \frac{1}{\sqrt{1-\lambda_1}}$. Now, letting $\lambda_1 =\frac{2n/R-1}{2n/R}=\frac{\lambda}{2n/R}$, we drive that
> $$\mathbb{E}[e^{\lambda X^2}]=\mathbb{E}\left[e^{\frac{\lambda_1 X^2}{2\sigma_{k,r}^2}}\right]\leq \frac{1}{\sqrt{1-\lambda_1}}=\sqrt{2n/R}\leq 2n/R.$$
> We added the reference in the proof for clarity. Moreover, previously for ease of expositions, we had reported the bound with the term $\log(2n/(R\delta))$ (and also since this term is rarely a dominant term). In the new version, we present them with the slightly tighter bound $\log(\sqrt{2n}/(\sqrt{R}\delta))$, for the clarity of the derivation.
>
> The link: https://www.cambridge.org/core/books/highdimensional-statistics/basic-tail-and-concentration-bounds/30AF7B572184787F4C99715838549721
>
> - **Regarding the experiments:** Please refer to our response to all reviewers.

---

> > ### Comment · Reviewer_9uEx · 2023-11-17
> > **Thank you for your response.**
> >
> > I thank the authors for their response, I increased my score to 5.
> >
> > I believe however that presenting your experimental results with respect to the generalisation gap and not the generalisation risk is misleading as the way curves are presented suggest simply to take $R=0$ to ensure a minimal generalisation gap. However, this is obviously undesirable as a random predictor, can both have a bad generalisation risk and empirical risk. The conveyed message is that we may need to communicate less often, however the ways results are presented suggest not to communicate at all, I believe a clear appearance of the tradeoff in $R$ in the curves is essential.

---

> ### Author Response · Authors · 2023-11-17
>
> Thank you much for the feedback and new comments, which are relevant.
>
> The reviewer is right indeed on that, if one’s focus is only the generalization error then the message would be “just communicate once” (note the minimum value is $R=1$, not $R=0$), which of course would not be reasonable as small values of $R$ may not be enough to drive the empirical risk into a sufficiently small value. We are aware of this; and did exercise caution on this in the exposition at places of the paper. Also, it is precisely for this reason that all the experimental results that we reported we also provide curves on the evolution of the population risk (which is the true right measure of performance). For example, please see Fig. 3(b) where it is clear that the minimum value of population risk is obtained with $R = 100$ (which is way smaller than the optimum choice of $R$ from an empirical risk perspective (which is $R=3600$), and way bigger than the optimum choice of $R$ if we were to base only on generalization error which is $R=1$). Please note that for FSVM too, we did provide curves of empirical and population risks versus $R$; but for lack of space they were deferred to the appendices (see Fig. 4). Observe that as is clear from those curves, $R=1$ is not enough to get the empirical risk small; and, in fact, in that setup the ERM is minimum for $R=15$. The figure also shows that the population risk decreases way less fast than the EMR; and it barely decreases after $R \approx 5$.  Other figures in the appendix also convey the same message such as Fig. 5-8. If the reviewer agrees, we can move those to the main body by merging Fig 2 and Fig 4.
>
> We hope this is now clearer. Nonetheless, based on the reviewer suggestion, we will revisit the exposition to make sure that no single sentence in the material may be misleading on this point.

---

> > ### Comment · Reviewer_9uEx · 2023-11-20
> >
> > Thank you for your reply.
> >
> > In my humble opinion, the clearest way to present your result would be to plot two curves: first, the population risk which satisfies empirically a tradeoff in $R$ as in Fig 3b) and also a new curve empirical risk + bound to see a similar tradeoff appearing in the upper bound. This would emphasise properly the role of $R$.

---

> > > ### Author Response · Authors · 2023-11-22
> > >
> > > We thank the reviewer for the provided feedback.  However, we never claimed to propose a method to predict such a value of $R$, and this is beyond the scope of the current work. We agree that this is an interesting question that deserves a separate work. We would appreciate it if the contributions of this paper were considered and evaluated in their context, as summarized in the response to all reviewers:
> > >
> > > **1)** the **first** PAC-Bayes and Rate-distortion theoretic generalization bounds for FL that explicitly capture its structure,
> > >
> > > **2)** showing the increasing behavior of the generalization bound with respect to $R$ for FSVM, which consequently leads to a slower decrease of the population risk with $R$ compared to the empirical risk,
> > >
> > > **3)** the experimental validation of this finding, and further the demonstration of a case (with local models using the ResNet-56 architecture) where the population risk may increase beyond a certain value of $R$.

---

> > > > ### Comment · Reviewer_9uEx · 2023-11-22
> > > >
> > > > I thank the authors for their reply.
> > > >
> > > > I agree that predicting such a value for $R$ is beyond the current scope. However, I believe the paper would be strengthened by showing such a tradeoff in $R$ appears in the theoretical bound (without trying to predict such a value). Indeed, 'showing the increasing behaviour of the generalisation bound' would be more impacting (to me) only if it reflects some similar behaviour for the population risk. Here, you show that your bound follows the same increasing behaviour wrt to $R$ as the generalisation gap, but this increase is not surprising as we start from a predictor uniformly performing poorly on both training and test sets. This generalisation gap does not translate properly the generalisation ability of either FSVM or neural nets as $R=1$ is the optimal value. The neural nets experiments are going in this direction but are not supported by numerical implementations of theoretical results.
> > > >
> > > > I am not saying that controlling the generalisation gap is not interesting, but I am not convinced by the experiments section in its current shape and believe it has to be reworked, therefore I am maintaining my current score.

---

### Official Review · Reviewer_jTnd · 2023-11-01

**Soundness:** 2 fair
**Presentation:** 3 good
**Contribution:** 2 fair
**Rating:** 5
**Confidence:** 3

**Summary:**

This paper examines the impact of the number of rounds on generalization errors in a federated learning setting. It presents generalization errors in the form of PAC-Bayes bounds and rate-distortion theoretic bounds and applies these results to federated learning support vector machines. The authors argue that the generalization errors increase with more rounds of communication (R).

**Strengths:**

1) It's a really interesting problem. There are many papers studying the convergence properties of FL algorithms, however, little work is done in terms of the generalization of these algorithms.

2) This paper proves three new bounds that explicitly have the number of rounds (R) in their bounds which allows the study of its effect on generalization.

3) They apply the bounds to FL-SVM to get an explicit result and have experiments validating their bound.

**Weaknesses:**

1) The assumption that R < N and each data point is only visited in one round is not realistic and doesn't capture what happens for FL in practice. In most settings in FL, the number of samples per client (N) is small, and the number of rounds is much higher. Additionally, I don't see if the current analysis is possible to extend to these cases.

2) Generally in FL papers with a focus on optimization, they report the curves about the loss and accuracy of the test set during the training, and it's common that the performance on the test set improves over more rounds, so the results can't extend to the more general cases, beyond FL-SVM.

3) The quantity of interest in ML is the true risk (population risk) and not the generalization. Even with the current assumptions, If the speed of decrease of training loss is more than the increase of generalization, it's not possible to argue the smaller number of rounds is better. In Fig 4. and Fig 8. of the appendix we can't see the increase in the population risk with the number of rounds for FL-SVM.

4) As mentioned in the paper, another approach would be to just apply basic PAC-Bayes bounds such as McAllester's without explicitly considering the dynamics of FL. Some numerical comparison of these bounds is needed. It's not clear to me that the bound in theorem 1 gives a better guarantee. McAllester's bound doesn't have the structure of the FL explicitly, however, it would be applied to the output of the algorithm and that might be enough. Also, it doesn't require the assumptions mentioned in the weakness 1.

5) Based on my understanding of the analysis, I assume that the same approach can be applied to the number of local steps per round if each data point would participate in just one local round. As a result, the number of local steps would also appear in the bound. It would be interesting to see its effect, as in the experiments of the paper, the total number of SGD steps is fixed, and with the increase of R, the number of local steps decreases.

**Questions:**

Please discuss the weaknesses.

---

> ### Author Response · Authors · 2023-11-15
> **Response to Reviewer jTnd**
>
> We thank the reviewer for the time spent on our paper and their interest in the problem studied in our paper.
>
> For the responses regarding the raised concerns 1-3, please refer to our response to all reviewers.
>
> 4. Using the classic McAllester's PAC Bayes bound has two major shortcomings. First, considering the lossy version of it (which is more general) for FSVM leads to bounds of order $\mathcal{O}\left(\sqrt{\frac{B^2 \log(nK)}{nK\theta^2}}\right)$. This bound does not reflect the effect of the number of clients (since it depends only on $nK$, i.e., the total number of samples available, in contrast to our bound that depends on $nK^2$), nor does it include the effect of the number of rounds. More importantly, considering the entire learning algorithm as a single centralized algorithm and applying McAllester's bound results in a bound that can only be computed when all datasets are available on a server, and also only when the final model $\overline{W}^{(R)}$ is obtained. In contrast, our bound requires only the "local" availability of each client's data in each round, and requires only that client's "local" model at the end of the corresponding round. This is indeed a realistic assumption, and also helpful for "estimating" the total cumulative contributions to the generalization error at each round.
>
> 5. We could not unfortunately understand what the reviewer is suggesting. If our interpretation is correct, we recall that each data point is already used in only one local round in our considered setup, since a local dataset is split into $R$ subsets. We would like to kindly ask the reviewer to give us more details about their suggestion.

---

> > ### Comment · Reviewer_jTnd · 2023-11-20
> >
> > I thank the authors for their response.
> >
> > I'm not convinced by the explanations regarding the studied setting. Unlike online learning, in Federated Learning (FL), we don't have streams of data, and clients use their samples multiple times. Usually, after many rounds of communication, both the training and test errors continue to decrease, especially when each sample can be used multiple times.
> >
> > As a consequence of this setup, an increase in $R$ means a decrease in samples per round to $\frac{n}{R}$. I think the effect of the smaller number of samples per round is more significant for these results than the effect of the number of rounds.
> >
> > Regarding 5: Consider the scenario where, in each global round, every client performs $e$ local steps, and each sample appears in only one local step (thus, each client has $Re$ disjoint subsets of $\frac{n}{Re}$ samples). If we apply the same analysis here, which factors in an explicit effect of $e$, can we expect a similar conclusion? (That the increase in $Re$ worsens the generalization.)

---

> > > ### Author Response · Authors · 2023-11-21
> > >
> > > We thank the reviewer for the new feedback. The comments by the reviewer on the practicality of the setup in which each sample can be used multiple times and what the reviewer believes to be the right behavior of the population risk as a function of the number of rounds in that case are certainly very interesting; and we would be happy to look at those aspects in a separate work. For now, we would be grateful if the contributions of this paper are considered and evaluated in their context. That is:
> > >
> > > 1) Analysis of Federated Learning from the view point of generalization error *is lacking* in both the general setup that the reviewer considers more practical or the one we assume in this paper). There exist no real bounds on the generalization error of FL that exploit its interactive structure, and of course none that is explicit in the number of rounds and system parameters. From this perspective, our bounds are the first of their kind for FL in the setup that we consider (which the reviewer considers un-practical)
> > >
> > > 2)  The very same setup that we consider was already considered in the context of Online Federated Learning, see, e.g
> > >
> > > * Chen et al. "Asynchronous Online Federated Learning for Edge Devices with Non-IID Data" 2020 IEEE International Conference on Big Data, 2020
> > > * * Mitra et al. "Online Federated Learning" 2021 60th IEEE Conference on Decision and Control (CDC)
> > >
> > > and similar ones in the context of LocalSGD, see, e.g
> > >
> > > * S. Stich, " Local SGD Converges Fast and Communicates Little",   (ICLR 2019) also available at https://arxiv.org/abs/1805.09767
> > > * Haddadpour et al.  Local SGD with Periodic Averaging: Tighter Analysis and Adaptive Synchronization (NeurIPS 2019)
> > > * Gu et al. “Why (and when) does LocalSGD generalize better than SGD ?”, ICLR 2023
> > >
> > > 3) What we claim in this paper are:
> > >
> > > * a bound on the generalization error of Federated SVM that increases with R; and which suggests that the generalization error of FSVM itself does so. This has also been validated experimentally.
> > > * a direct consequence of this is that "the population risk decreases with R less fast than the empirical risk"
> > > * a bound on the generalization error of general FL (under our setup); and experimental results that show that the population risk can even start to increase beyond some value of R.
> > >
> > > [Conclusion of the paper] At this point, we emphasize that, in particular, we here make no claim as for whether there should always be a tradeoff for the population risk that is similar to that that we observed in Figure 3(b), neither for general setups (where the entire dataset is used during each round) nor in our restricted setting. Actually, the example for FSVM shows no such  tradeoff; but in all presented experimental results the population risk decreases less fast with R than the empirical risk (from the experiment of Fig. 3(b), we know that in some cases the population risk may even start to increase start from some value of R). This latter finding was already observed in the context of Local SGD, see Figure 2 of Gu et al. Why (and when) does LocalSGD generalize better than SGD ?”, ICLR 2023.
> > >
> > > One last point regarding whether the behavior reported in Fig(3) should or not be attributed to a decrease of the number of data points used for training locally at each round. In the opinion of the authors, it does not seem so. First, please observe that, for a large range of R, the population risk continues to decrease with R (less fast than ERM) even though the client uses fewer and fewer data points locally (n/R). It is only beyond some proper threshold R* that the change of the slope of population risk is observed. Second, the very same behavior was observed
> > >
> > >  Gu et al. Why (and when) does LocalSGD generalize better than SGD ?”, ICLR 2023.
> > >
> > > where each data point is visited multiple times.
> > >
> > > **Regarding your point 5**: We don't see a straightforward extension of the current analysis used in proof of Theorem 4 to such a case. The reason is that one of the necessary steps in the proof of Theorem 4 is the perturbative analysis of the quantized local model of each client at the end of each round. This perturbative analysis is only valid if it is averaged over the model of other clients (or some similar "fair" aggregation method). Thus, it does not extend (at least trivially) to the case where different local steps of each client are considered separately.

---

> > > > ### Comment · Reviewer_jTnd · 2023-11-23
> > > >
> > > > I thank the authors for their reply. I increase my score to 5.
> > > >
> > > > I still think more study about the effect of the assumption is needed. For example, some experiments where each sample is used in a fixed number of rounds, to study the effect of the number of samples vs the effect of number of rounds. Also, a clear study and clarification of the trade-off, the conclusion and the results in Fig.4 and Fig.8 is needed.

---

### Author Response · Authors · 2023-11-15
**Responses to all reviewers**

We thank all reviewers for their relevant comments and suggestions. Here we clarify some of the take-home messages of our paper and discuss the comments raised on the considered setup. The other comments from each reviewer are addressed separately. We have also uploaded a new version of the paper with the changes highlighted in blue. We believe we have addressed all the reviewers' concerns, and we would be grateful if the reviewers let us know if any concerns remain.


1. **Regarding the conclusion:** There appears to be some misconceptions about the understanding of the main conclusions of this paper, which we reproduce hereafter for clarity:

- 1.i. First, please note that, as we now make it clearer in the abstract even, the conclusions of our paper pertain to this specific setup that we study and should not be taken to hold for general FL. In this setup that we consider (and which was also considered in other works such as in the study of convergence rates of LocalSGD e.g. Stich (ICLR 2019), Haddadpour et al. (NeurIPS 2019)) we wish to study tradeoffs among number of SGD steps performed locally by each client and number of models aggregations. For this reason, we set $n = \tau \times R$. This is useful in practice, e.g., in computationally-constrained training for FL --  see below for other motivations for this setup.

- 1.ii. For this setup, we show that (a bound on) the generalization error of FSVM increases with the number of rounds. This finding is validated experimentally for both FSVM and ResNet-56.

- 1.iii. A consequence is that the population risk decreases with the number of rounds less rapidly than the empirical risk. This is validated experimentally for both FSVM and using ResNet-56.

- 1.iv. Related to the second point, we note that it is possible that the population risk stops decreasing (see FSVM experiment) or even starts increasing (see ResNet-56 experiment) beyond a certain value of rounds. We hasten to mention that in the literature on LocalSGD, experimental results have already been reported where the population risk does start to increase beyond some critical value $R^*$. See, e.g., Figure 2.e. of Gu et al. (ICLR 2023).

- 1.v. We mention that the finding that the population risk may either increase or at least decrease only marginally beyond a certain value of $R$ is important in practice, especially for applications of FL in telecommunication problems. In this case, one may deliberately choose a value of $R$ that gives a sufficiently small empirical risk, because further communication will consume network resources (power and bandwidth) with only a very small return in terms of population risk improvement.

- 1.vi. Finally, in order to avoid confusion, we have adjusted the title to now be "Federated Learning, Lessons from Generalization Study: You May Need to Communicate Less Often" and we reformulated better some sentences in the abstract and body of the material which could have lead to the aforementioned misconceptions.


2. **Regarding the setup:**

- 2.i. First, we mention that there exist already reported works with identical or similar assumptions as ours, e.g., in the context of online learning (see Haddouche and Guedj (Neurips 2022)). Our setup can also be seen as the first epoch of the famous without-replacement sampling, which is used extensively for both LocalSGD and Federated Learning algorithms. In such setups, the whole training dataset does not have to be used at each round, as discussed in Appendix B of Gu et al.

- 2.ii. For the derivation of the theoretical results of this paper, the disjointedness of the datasets is only assumed for simplicity (the bounds read better in this case). In the Appendix~B of the new version, we have provided some of the results for the general case, where each sample may appear in multiple rounds of one client. We have shown how our framework can be easily adapted to derive the results in its general case, which as special case reduces to the existing results. We did this only for the "lossless" version of Theorem 3. Similar bounds can be derived for lossy case as well as other bounds. However, the results involve complicated correlation between each client's dataset at each round with models at different rounds and needs a comprehensive study is needed to rigorously investigate such dependencies to make the bounds more practical. This is left for future work.

---

### Author Response · Authors · 2023-11-20
**Seeking feedback to our rebuttal**

Dear Reviewers,

Once again, we thank you much for comments which helped us clarify better the contributions of our paper.

We would appreciate if you could please take a look at our rebuttal, and let us know as soon as this is possible for you if there are any remaining concerns on your side.

Best of the Regards
Authors

---

### Meta-Review · Area_Chair_fLx6 · 2023-12-19

**Metareview:**

The paper aims to study the generalization error of statistical learning models in a Federated Learning (FL) setting in connection to the number of communication rounds between the clients and the parameter server. The authors establish PAC-Bayes and rate-distortion theoretic bounds on the generalization error that account explicitly for the effect of the number of rounds as well as the number of participating devices and individual datasets size.

I thank the authors for their thorough responses. The reviewers had raised several concerns, some alleviated by the responses, but some others have remained (and there was a reasonable amount of discussion between the authors and the reviewers). I also had a close look at the rebuttal+discussions+the paper. While I find the contributions of the paper to be solid and interesting, I recommend the authors to rewrite some parts of the paper to reflect/incorporate the suggestions given by the reviewers and avoid misunderstandings about the theoretical setup. The reviewers, some of whiich are experts on the topic, have raised some important points. I also encourage the authors to incorporate the comments from the reviewers. I believe that, once the reviewers' comments are fully addressed, the paper will be an excellent contribution to the FL literature.

**Justification For Why Not Higher Score:**

There has been enough discussion on the paper, and the reasonable decision would be to reject based on the discussions and scores.

**Justification For Why Not Lower Score:**

--

---

### Decision · Program_Chairs · 2024-01-16

Reject